# A Decision-Theoretic View of Test-Time Training: When, How Far, and Which Directions to Adapt

**Tomoya Wakayama** [1]

## Abstract

Test-time training (TTT) adapts a pretrained model to each prompt via parameter updates, improving accuracy under pretraining-to-test distribution shifts. Yet, its performance often suffers from instability and sensitivity to hyperparameters such as update steps and subspace. We explain this behavior through a decision-theoretic lens, treating TTT as implicit Bayesian inference in the kernel regime. Under a Gaussian process benchmark, we show that TTT reduces prediction error when updates are spectrally matched to the prompt's signal-to-noise ratio and aligned with query-relevant eigen-directions. This perspective underpins the following results: (1) we show when fixed update steps and subspaces fail under distribution shifts, motivating adaptive strategies; (2) we prove that selecting update steps via prompt evidence admits a PAC-Bayes guarantee against overfitting; and (3) we characterize the Bayes-optimal update subspace under a linear-Gaussian correction model, yielding a scoring rule for selecting Transformer blocks and heads. Our theory helps explain the empirical instability of TTT, taking a step toward principled guidance for when, how far, and which directions to adapt.

## 1. Introduction

Modern foundation models increasingly serve as general-purpose predictors, answering diverse user queries. Yet real deployments frequently deviate from pretraining conditions, in which a fixed (no-update) inference procedure can become brittle. This has kindled interest in test-time training (TTT), which updates model parameters using only the prompt to adapt to a specific test task. Following the

seminal work by Sun et al. (2020), a growing body of work has shown that TTT improves robustness and domain generalization in vision (Wang et al., 2021; 2022; Yuan et al., 2023), extends to high-stakes settings such as medical image segmentation (Valanarasu et al., 2024), and adapts end-to-end automatic speech recognition models to non-stationary acoustic conditions (Kim et al., 2023; Lin et al., 2024). More recently, analogous ideas have also been explored for large language models (Hardt & Sun, 2024; Hübotter et al., 2025a).

While these empirical improvements are noteworthy, TTT often exhibits instability in practice. TTT's performance is sensitive to optimization hyperparameters (e.g., update spaces and step count), and more steps can negatively impact performance (Zhao et al., 2023). This sensitivity is exacerbated by non-stationary shifts and small-batch streaming (Boudiaf et al., 2022; Niu et al., 2022), and analogous issues have been reported for language models (Akyürek et al., 2025). While recent works seek more stable updates and safer parameter choices (Alfarra et al., 2025; Sahoo et al., 2025), we still lack principled guidance for TTT.

Prior theories (Gozeten et al., 2025; Kuwataka & Suzuki, 2025) have started to characterize when TTT can improve in-context learning (ICL, i.e., no parameter updates) in specific regimes (e.g., linearized transformers or structured task models). Complementary to the ICL-centered analyses, Hübotter et al. (2025b) propose a mechanistic explanation for why TTT can help at scale even on nominally in-distribution data. Still, general principles guiding the choice of optimization hyperparameters remain elusive. Our work focuses on the following questions: (i) *when TTT reduces the predictive risk relative to ICL*, (ii) *how the number of test-time update steps affects performance*, and (iii) *which parameter subspace is effective for updating*.

In a kernel (linearized-quadratic) regime, where TTT typically operates, we present a Bayes-risk perspective by interpreting TTT as *implicit Bayesian inference*. Here, the update steps and the update subspace act as hyperparameters of a prompt-induced prior. Our claims are as follows.

**(C1)** Under a Gaussian process benchmark that couples prompt and query, we derive a conditional risk identity showing that query predictions benefit from both filter

[1]RIKEN Center for Advanced Intelligence Project (AIP), Tokyo, Japan. Correspondence to: Tomoya Wakayama <tomoya.wakayama@riken.jp>.

*Proceedings of the 43rd International Conference on Machine Learning*, Seoul, South Korea. PMLR 306, 2026. Copyright 2026 by the author(s).

match (in eigenvalue space) and eigen-alignment with the prompt-query coupling vector (Thm. 5.2). We further show that fixed steps and fixed update subspaces can fail (Propositions 6.1–6.3).

**(C2)** We show that selecting the number of updates by prompt evidence provides a PAC-Bayes guarantee and mitigates overfitting to the prompt (Thm. 7.2).

**(C3)** Under a linear-Gaussian correction prior, we characterize the Bayes-optimal update subspace via an information-capture matrix, yielding an attention block/head scoring rule for prediction (Thm. 8.2); see App. E.6 for a regret bound.

Together, these results give a principled starting point for analyzing few-step, restricted-update TTT. We do not aim to give a complete theory of fully nonlinear TTT. Rather, we use a local Bayesian lens to identify design principles for TTT: how far to update, and which directions to update. The Transformer specializations in Secs. 3.3 and 8.2 make this link to block/head-level TTT explicit.

### 1.1. Related Work

**Test-time training.** TTT is widely used for domain generalization and test-time adaptation (Sun et al., 2020; Niu et al., 2022). Recent work studies parameter-efficient or specialization-oriented updates for foundation models (Hübotter et al., 2025b). Theoretically, Gozeten et al. (2025) relate TTT to ICL in linearized transformer settings, and Kuwataka & Suzuki (2025) show multi-step TTT can enhance ICL on single-index tasks. We cast TTT as Bayesian inference in a kernel regime, analyzing a risk identity and PAC-Bayes theory for steps and optimal update subspaces.

**Bayesian view of test-time behavior.** ICL is often viewed as amortized Bayesian inference learned through pretraining (Reuter et al., 2025; Mittal et al., 2025). Because its inference mapping is fixed, correcting for prior mismatch (Wakayama & Suzuki, 2025) or the amortization gap (Margossian & Blei, 2024) at test time remains challenging. Unlike fixed in-context inference, we view TTT as prompt-induced Bayesian inference, revealing how evidence-tuned steps and subspaces correct prior mismatch.

## 2. Setup: tasks, prompts, and risk

In TTT and ICL, we consider a distribution over tasks: at test time, a latent task is sampled, and then the prompt and query are generated from the corresponding task-specific data distribution. This section introduces a mixture (hierarchical) regression setting as in Wakayama & Suzuki (2025) and formalizes an analytical framework.

**Mixture task model and test-time information** Let $\alpha$ denote a latent task variable drawn from a meta-distribution

$\pi$ over tasks. Conditioned on $\alpha$, data pairs $(X, Y)$ follow a task-specific distribution $\mathcal{P}_\alpha$ over $\mathcal{X} \times \mathcal{Y} \subset \mathbb{R}^{d_x} \times \mathbb{R}$. At test time, we observe an i.i.d. labeled prompt $P_n := \{(X_i, Y_i)\}_{i=1}^n$, followed by a query input $X$ and target $Y$, all generated from the same task

$$\alpha \sim \pi, \quad (X_i, Y_i) \mid \alpha \overset{\text{i.i.d.}}{\sim} \mathcal{P}_\alpha, \quad (X, Y) \mid \alpha \sim \mathcal{P}_\alpha. \quad (1)$$

At test time, only $(P_n, X)$ is observed, while $\alpha$ is not. Throughout, we assume each labeled example corresponds to a single token position.

**Bayes gap.** We use the squared loss $\ell(u, v) = (u - v)^2$, following Gozeten et al. (2025); Kuwataka & Suzuki (2025). To evaluate a measurable map $M : (P_n, X) \mapsto \widehat{Y} \in \mathbb{R}$, we adopt the *Bayes risk* (see Murphy, 2022, §5.3.1.2), i.e., the expected squared prediction error $\mathcal{R}(M) := \mathbb{E}\big[\{M(P_n, X) - Y\}^2\big]$, where the expectation is over the joint distribution in (1).

In this setting, the minimizer of the Bayes risk among all square-integrable measurable maps of $(P_n, X)$ is the conditional mean $M_{\text{Bayes}}(P_n, X) := \mathbb{E}[Y \mid P_n, X]$. See Appendix C for a review of Bayes risk and the Bayes predictor.

**Proposition 2.1** (Orthogonal Bayes-risk decomposition)**.** *Assume $\mathbb{E}[M(P_n, X)^2] < \infty$ and $\mathbb{E}[Y^2] < \infty$. Then*

$$\mathcal{R}(M) = \underbrace{\mathbb{E}\Big[\big(M(P_n, X) - M_{\text{Bayes}}(P_n, X)\big)^2\Big]}_{=:\,\mathcal{G}(M)\ \text{(Bayes gap)}}$$

$$+ \underbrace{\mathbb{E}\Big[\big(M_{\text{Bayes}}(P_n, X) - Y\big)^2\Big]}_{=:\,\mathcal{V}\,=\,\mathbb{E}[\text{Var}(Y|P_n, X)]\ \text{(posterior variance)}}. \quad (2)$$

Here, $\mathcal{V}$ is the irreducible uncertainty about $Y$ after observing $(P_n, X)$ and is independent of predictor $M$. In contrast, the Bayes gap $\mathcal{G}(M)$ is the excess risk relative to $M_{\text{Bayes}}$, measuring how accurately $M$ approximates $M_{\text{Bayes}}$ as a map of $(P_n, X)$. Thus, any improvement of TTT must come from reducing $\mathcal{G}$.

**Correction view.** To analyze the adaptation process, consider the setting where we correct a base predictor $B$ (e.g., ICL). We write any adapted rule as $M = B + C$ with correction $C$. Let $\Delta_B(P_n, X) := \mathbb{E}[Y \mid P_n, X] - B(P_n, X)$ be the Bayes-optimal correction. Under squared loss, the Bayes gap of $M$ simplifies to correction gap $\mathcal{G}(M) = \mathbb{E}\big[\{C(P_n, X) - \Delta_B(P_n, X)\}^2\big]$. Thus, improving over $B$ amounts to learning a prompt-dependent correction.

As the Bayes correction under the general model is typically intractable, in Secs. 5–8, we analyze the Bayes gap relative to a tractable benchmark Bayes correction.

## 3. A Local Linearized Model for TTT

This section builds a tractable local model for few-step TTT. We do not assume that practical TTT is globally linear. Instead, we use the first-order model as a controlled approximation to the nonlinear update path. This approximation is appropriate when the update subspace is restricted and the number of test-time steps is modest; Appendix F.1 gives a finite-step deviation bound between the nonlinear and linearized prompt corrections. Within this model, TTT becomes a linear inverse problem in the update parameters. Since test-time supervision is available only through the labeled prompt, we use leave-one-out residuals as noisy observations of the desired correction at prompt points.

**Residual correction induced by a base predictor.** Let TTT output $M_{\text{TTT}}(P_n, x) = B(P_n, x) + C(P_n, x)$. We form leave-one-out residuals $r \in \mathbb{R}^n$ with $r_i := y_i - B(P_n^{(-i)}, x_i)$, where $P_n^{(-i)} := P_n \setminus \{(x_i, y_i)\}$. Define the (prompt-point) Bayes-optimal correction by $f_{\star,i} := \mathbb{E}[Y_i \mid P_n^{(-i)}, x_i] - B(P_n^{(-i)}, x_i)$. Then, $\mathbb{E}[r_i - f_{\star,i} \mid P_n^{(-i)}, x_i] = 0$, so $r$ is an unbiased noisy observation of $f_\star$. We use $r$ to avoid label leakage: the target $y_i$ is not used in computing either $B(P_n^{(-i)}, x_i)$ or the Jacobian feature, defined later.

### 3.1. Restricted update subspace and Jacobian features

Let $w \in \mathbb{R}^p$ denote the full parameter vector of a model and $w_0$ its pretrained value. Let $f_w(x; P_n) \in \mathbb{R}$ denote the model prediction at query input $x$ given prompt $P_n$ under parameters $w$. Note that $B(P_n, x) = f_{w_0}(x; P_n)$. Since practical TTT typically updates only a small subset of parameters or a low-rank reparameterization (Wang et al., 2021; Niu et al., 2022; Hübotter et al., 2025a;b), we consider the following setting.

**Assumption 3.1** (Restricted update subspace). There exists a $d(< p)$-dimensional linear subspace $\mathcal{U} \subset \mathbb{R}^p$ such that test-time updates satisfy $w \in w_0 + \mathcal{U}$.

Only the subspace $\mathcal{U}$ matters, not a particular basis, that is, any full-column-rank $\widetilde{A}$ with $\text{range}(\widetilde{A}) = \mathcal{U}$ can be replaced, without loss of generality, by an orthonormal basis. Let $A \in \mathbb{R}^{p \times d}$ be such a basis ($A^\top A = I_d$), so that any update can be written as $w(\theta) = w_0 + A\theta$. We denote the orthogonal projector by $\Pi_{\mathcal{U}} := AA^\top$.

Since TTT uses few gradient steps within a low-dimensional update subspace, a first-order expansion in $\theta$ is a natural local surrogate (Jacot et al., 2018; Lee et al., 2019; Chizat et al., 2019). We refer to this few-step approximation as a local linear (kernel/NTK) regime.

**Assumption 3.2** (Linearized surrogate model). We analyze TTT through the linearized surrogate $f_\theta(x; P_n) := f_{w_0}(x; P_n) + \phi(x; P_n)^\top \theta$, with Jacobian features $\phi(x; P_n) := \nabla_\theta f_{w(\theta)}(x; P_n)\big|_{\theta=0}$.

Note that the approximation $f_\theta(x; P_n) \approx f_{w_0}(x; P_n) + \phi(x; P_n)^\top \theta$ is valid in the few-step regime, as in common practice (Akyürek et al., 2025; Sun et al., 2020). See Appendix F.1 for validation of the assumption.

Stack the Jacobian features into the matrix $\Phi := (\phi(x_1; P_n^{(-1)}), \ldots, \phi(x_n; P_n^{(-n)}))^\top \in \mathbb{R}^{n \times d}$. Under Assumption 3.2, the vector of prediction shifts on the prompt is approximately $\Phi\theta$. Hence, fitting the residuals $r$ reduces to a linear inverse problem in $\theta$.

### 3.2. Quadratic prompt loss and the prompt-induced geometry $K(P_n)$

Based on the linearization, we use the quadratic prompt objective $L(\theta) := \frac{1}{2\sigma^2}\|r - \Phi\theta\|_2^2$. For fixed $\sigma^2$, minimizing $L$ is maximum likelihood estimation in the Gaussian regression model $r = \Phi\theta_\star + \varepsilon$ with $\varepsilon \sim \mathcal{N}(0, \sigma^2 I_n)$, where $\theta_\star$ is the (unknown) optimal update parameter that best explains the residuals under the linearization and $\sigma^2$ is an effective noise level that accounts for label noise and linearization errors. In Sec. 4, we study the few-step optimization dynamics of $L$.

In this kernel regime, the optimization dynamics on this objective are governed by the prompt-space Gram structure of the features, captured by $K(P_n) := \Phi\Phi^\top \in \mathbb{R}^{n \times n}$. We refer to $K$ as the *prompt-induced geometry*, as it determines how updates couple the $n$ prompt residuals during optimization. Crucially, this geometry depends on both the prompt instance $P_n$ and the update mechanism $A$ (e.g., which attention heads are adaptable at test time). This dependence distinguishes TTT from classical early stopping with a fixed geometry, where the Gram structure is predetermined.

To relate the prompt geometry $K = \Phi\Phi^\top$ to the full parameter space and the chosen update subspace, collect the $n$ supervised prompt outputs as $s_w(P_n) = (s_1(w), \ldots, s_n(w)) \in \mathbb{R}^n$ with $s_i(w) := f_w(x_i; P_n^{(-i)})$ and targets $y \in \mathbb{R}^n$; then $r = y - s_{w_0}(P_n)$. Let $J_w := \nabla_w s_w(P_n)\big|_{w=w_0} \in \mathbb{R}^{n \times p}$. Since $w(\theta) = w_0 + A\theta$, the Jacobian in update parameters is $\nabla_\theta s_{w(\theta)}(P_n)\big|_{\theta=0} = J_w A$, i.e., $\Phi = J_w A$, and thus $K = \Phi\Phi^\top = (J_w A)(J_w A)^\top = J_w \Pi_{\mathcal{U}} J_w^\top$.

### 3.3. Specializing the Prompt Kernel to Transformers

We specialize $K$ to Transformer attention blocks (Vaswani et al., 2017) under a local frozen-feature regime. Specifically, we evaluate all forward activations (including attention weights) and upstream gradients at $w_0$ and treat them as constants (i.e., we ignore their dependence on the test-time updates), in the spirit of Chizat et al. (2019); Fu et al. (2023). See Appendix H.1 for detailed discussion.

Let $h \in \{1, \ldots, H\}$ index attention heads. We in-

dex the $n$ supervised prompt positions (tokens) by $k \in \{1, \ldots, n\}$ and define the head output at token $k$ by $o_k^{(h)} := W_O^{(h)} u_k^{(h)}$, $u_k^{(h)} := \sum_{s \leq k} \alpha_{ks}^{(h)} W_V^{(h)} z_s$, where $z_s$ is the residual stream and $\alpha_{ks}^{(h)}$ are attention weights (evaluated at $w_0$). Under tokenwise supervision ($i \equiv k$), let $g_k^{(h)} := \nabla_{o_k^{(h)}} s_k(w) \big|_{w_0}$ be the upstream gradient.

Define $m_k^{(h)} := \sum_{s \leq k} \alpha_{ks}^{(h)} z_s$ and $\beta_k^{(h)} := (W_O^{(h)})^\top g_k^{(h)}$, so $u_k^{(h)} = W_V^{(h)} m_k^{(h)}$. Then the Jacobian feature w.r.t. $W_V^{(h)}$ is $m_k^{(h)} \otimes \beta_k^{(h)}$ and $K_{kk'}^{(h,V)} = (m_k^{(h)\top} m_{k'}^{(h)})(\beta_k^{(h)\top} \beta_{k'}^{(h)})$.

Stack $Z = [z_1^\top; \ldots; z_n^\top]$, $\mathcal{A}_{ks}^{(h)} = \alpha_{ks}^{(h)}$, $M^{(h)} := \mathcal{A}^{(h)} Z$, and $\mathbf{B}^{(h)} = [\beta_1^{(h)\top}; \ldots; \beta_n^{(h)\top}]$. Then

$$K^{(h,V)} = \left(M^{(h)} M^{(h)\top}\right) \odot \left(\mathbf{B}^{(h)} \mathbf{B}^{(h)\top}\right), \qquad (3)$$

and if $g_k^{(h)} \equiv g^{(h)}$ (hence $\beta_k^{(h)} \equiv \beta^{(h)}$),

$$K^{(h,V)} = \|\beta^{(h)}\|_2^2 \, \mathcal{A}^{(h)} (Z Z^\top) \mathcal{A}^{(h)\top}. \qquad (4)$$

## 4. Few-step TTT as implicit Bayesian inference

This section shows that the resulting update is a spectral filter and admits an exact auxiliary Bayesian interpretation.

**Gradient descent dynamics.** To analyze the update dynamics under the local linear–quadratic surrogate of Sec. 3, fix a prompt $P_n$ and consider gradient descent on the quadratic prompt loss $L(\theta)$ starting from $\theta_0 = 0$. With a scaled step size $\rho := \eta/\sigma^2$, the parameter update is

$$\theta_{t+1} = \theta_t - \eta \nabla L(\theta_t) = \theta_t + \rho \, \Phi^\top (r - \Phi \theta_t). \quad (5)$$

Multiplying (5) by $\Phi$ yields the dynamics of the in-prompt prediction correction $\hat{f}_t := \Phi \theta_t$ as $\hat{f}_{t+1} = \hat{f}_t + \rho K(r - \hat{f}_t)$, where $K := \Phi \Phi^\top$. This reveals that TTT operates on $r$ solely through $K(P_n)$.

**Spectral filtering.** In this setup, the standard early-stopping identity (Yao et al., 2007; Bauer et al., 2007) gives $\hat{f}_T = G_T r$, where $G_T := I_n - (I_n - \rho K)^T$. If $K = U \operatorname{diag}(\lambda_1, \ldots, \lambda_n) U^\top$ denotes an eigen-decomposition with $\lambda_i \geq 0$, then

$$\hat{f}_T = U \operatorname{diag}\big(g_T(\lambda_1), \ldots, g_T(\lambda_n)\big) U^\top r, \qquad (6)$$

where $g_T(\lambda) := 1 - (1 - \rho \lambda)^T$. Set $0 < \rho < 1/\lambda_{\max}(K)$ to ensure $g_T(\lambda) \in [0, 1)$ for all $\lambda \in [0, \lambda_{\max}(K)]$ and $T < \infty$.

Under the same linearized model, the $T$-step update also induces a query-point correction of the form $\hat{h}_T(x) := \phi(x; P_n)^\top \theta_T = k_x^\top q_T(K) \, r$, where $k_x := \Phi \phi(x; P_n) \in \mathbb{R}^n$ and $q_T(\lambda) := \{1 - (1 - \rho \lambda)^T\}/\lambda$ for $\lambda > 0$ (with $q_T(0) := \rho T$). Thus, to understand when TTT helps, it suffices to compare the induced filter $q_T$ with the Bayes-optimal filter over the spectrum of $K$, with spectral weights

scaled by the alignment of $k_x$ with the eigenvectors of $K$ (Sec. 5).

**Bayesian view.** Recalling that the quadratic prompt objective corresponds to a Gaussian observation model with noise level $\sigma^2$ as in Sec. 3.2, we next show that the smoother $r \mapsto \hat{f}_T$ admits an exact representation as a Gaussian posterior mean. This interprets the algorithm-induced map $G_T$, and we do not assume the true residuals are generated from this prior. This view treats $T$ as a hyperparameter and selects it based on evidence (Sec. 7).

**Proposition 4.1** (TTT equals a Gaussian posterior mean). *Assume $0 < \rho < 1/\lambda_{\max}(K)$. There exists a Gaussian prior $f \sim \mathcal{N}(0, \sigma^2 G_T (I_n - G_T)^{-1})$ such that, under the auxiliary observation model $r = f + \varepsilon$, $\varepsilon \sim \mathcal{N}(0, \sigma^2 I_n)$, the posterior mean satisfies $\mathbb{E}[f \mid r] = \hat{f}_T$.*

The result suggests that TTT corresponds to Bayesian inference under a prompt-induced Gaussian prior, in which the **prior shape** (eigenvectors) and the **prior strength** (eigenvalues via $T$) vary across prompts. In particular, changing which attention blocks/heads are updated changes the eigenmodes of $K$, i.e., it changes the shape of the implicit prior.

Detailed derivations and discussions regarding extensions to other loss functions are provided in the Appendix E.2.

## 5. When does TTT help? Bayes-gap identities

Sec. 4 shows that few-step TTT applies a spectral filter to the prompt residuals through $K(P_n)$. For prediction, however, our interest lies in the induced query-point correction: under the local linear model (Sec. 3), $\hat{h}_q(x) = k_x^\top q(K) r$ with $k_x := \Phi \phi(x; P_n)$. See Prop. E.1 for the derivation.

To keep the presentation simple and make the Bayes-optimal query correction explicit, we adopt the following Gaussian process benchmark (Rasmussen & Williams, 2006). Below, $f_{\text{Bayes}}(x)$ denotes the posterior mean under a benchmark (a Bayes-optimal correction).

**Assumption 5.1** (Gaussian benchmark). Fix prompt inputs $(x_1, \ldots, x_n)$ and a query input $x$. Let $K \succeq 0$ be the prompt geometry and let the prompt–query covariance vector $k_x$ and the query variance $\kappa_x$ satisfy $\begin{pmatrix} K & k_x \\ k_x^\top & \kappa_x \end{pmatrix} \succeq 0$. Assume latent functions $(f_\star, f_x)$ are Gaussian with mean 0 and covariance $\tau^2 \begin{pmatrix} K & k_x \\ k_x^\top & \kappa_x \end{pmatrix}$, and we observe prompt residuals $r = f_\star + \varepsilon$ with $\varepsilon \sim \mathcal{N}(0, \sigma^2 I_n)$ independent. Write $\lambda_\star := \sigma^2/\tau^2$.

We do *not* claim real TTT residuals follow this model; rather, it provides an interpretable baseline for quantifying filter mismatch and motivates extensions beyond Gaussianity. See Appendix F.2 and F.3 for extensions.

**Theorem 5.2** (Conditional Bayes-gap identity). *Grant Assumption 5.1 and let $K = U \operatorname{diag}(\lambda_i) U^\top$ be an eigen-*

decomposition with eigenvectors $\{u_i\}_{i=1}^n$. *The Bayes-optimal estimator (posterior mean) of the query correction is* $f_{\text{Bayes}}(x) = k_x^\top (K + \lambda_\star I_n)^{-1} r$. *Moreover, for any kernel-linear estimator* $\hat{h}_q(x) = k_x^\top q(K)\, r$ *with* $q_\star(\lambda) := (\lambda + \lambda_\star)^{-1}$, *the Bayes gap satisfies* $\mathbb{E}\big[\{\hat{h}_q(x) - f_{\text{Bayes}}(x)\}^2 \mid K, k_x, \kappa_x\big] = \sum_{i=1}^n a_i(x)\big\{q(\lambda_i) - q_\star(\lambda_i)\big\}^2$, *where the mode weights are* $a_i(x) := (\tau^2 \lambda_i + \sigma^2)\,(u_i^\top k_x)^2$.

Thm. 5.2 suggests that achieving a small Bayes gap at the query requires both: (i) **filter match**, meaning that $q_T(\lambda_i)$ is close to $q_\star(\lambda_i) = (\lambda_i + \lambda_\star)^{-1}$ on the modes with large weight $a_i(x)$; and (ii) **eigen-alignment**, since only modes with non-negligible $(u_i^\top k_x)^2$ contribute through $a_i(x)$. This perspective parallels spectral analyses of generalization on kernel regression (Bordelon et al., 2020). Importantly, simply fitting the model to the prompt is not sufficient.

**Specialization to TTT and ICL (Corollary E.5).** TTT uses $q = q_T$, while naive ICL uses $q \equiv 0$. Thus, conditioned on $(K, k_x, \kappa_x)$, TTT improves over ICL in terms of the Bayes gap if and only if $\sum_{i=1}^n a_i(x)\big\{q_T(\lambda_i) - q_\star(\lambda_i)\big\}^2 < \sum_{i=1}^n a_i(x)\, q_\star(\lambda_i)^2$.

> **Takeaway (C1-1): When TTT helps.**
>
> TTT improves upon ICL via **1. Optimal Steps**: the step count tunes shrinkage (so that $q_T$ matches $q_\star$), avoiding underfitting/overfitting; **2. Geometric Alignment**: the update subspace contains directions learned from the prompt that are also aligned with the query prediction.

### 5.1. Geometry and Dynamics of Transformer TTT

**Prompt geometry as token coupling.** In a Transformer, the prompt geometry $K = \Phi\Phi^\top$ is an $n \times n$ token–token Gram matrix: $K_{kk'} = \langle \nabla_\theta s_k, \nabla_\theta s_{k'} \rangle$ measures the extent to which a small update (within the chosen update subspace) changes the outputs at tokens $k$ and $k'$ in similar directions. For value-matrix updates in head $h$, $K^{(h,V)} = (M^{(h)} M^{(h)\top}) \odot (\mathbf{B}^{(h)} \mathbf{B}^{(h)\top})$ (Eq. (3)), so a token pair is emphasized only when it is both similar in attention-mixed representations $M^{(h)}$ and aligned in upstream gradient directions $\mathbf{B}^{(h)}$. In the global-upstream regime, this simplifies to $K^{(h,V)} = \|\beta^{(h)}\|_2^2\, \mathcal{A}^{(h)}(ZZ^\top)\mathcal{A}^{(h)\top}$, an attention-sandwiched token-similarity matrix (symmetric PSD even for causal $\mathcal{A}^{(h)}$). Since head/block contributions sum to $K = \sum_{b \in \mathcal{B}} K^{(b)}$, *choosing which blocks/heads to update determines which eigen-directions in token-space are available for test-time correction.*

**Filtering dynamics and query coupling.** Few-step TTT applies the polynomial filter $\hat{f}_T = \sum_{t=0}^{T-1} \rho K(I - \rho K)^t r$, which can be viewed as diffusion/smoothing of residual information over this attention-induced token geometry; the horizon $T$ sets the diffusion time and thus the amount of

shrinkage across eigenmodes. Finally, Thm. 5.2 implies that query-point improvement comes only from eigen-directions that also align with the prompt-query vector $k_x$ (weights $(u_i^\top k_x)^2$), so *head/block choices matter both through the spectrum of $K$ and through its coupling to the query.*

## 6. Necessity of adapting the update steps and directions

The identity in Theorem 5.2 identifies two requirements for query improvement: the update filter must match the prompt signal-to-noise structure, and the updated eigendirections must align with the query. In this section, we show that fixing the number of update steps $T$ or the update subspace $A$ as in standard optimization can fail. We then argue for the necessity of prompt-dependent step selection and subspace design.

Recall from Sec. 4 that, for a fixed prompt $P_n$, few-step TTT on the quadratic prompt loss produces an in-prompt correction $\hat{f}_T = g_T(K)\, r$ with filter $g_T(\lambda) = 1 - (1 - \rho\lambda)^T$. In this section, we consider the broader class of prompt-only spectral estimators $\hat{f}_g := g(K)\, r$ for measurable $g : [0, \infty) \to \mathbb{R}$ with $g(0) = 0$, and denote by $\mathcal{S}(P_n) = \text{range}(K(P_n))$ the representable subspace in prompt space.

We outline the three propositions below. For the formal statements, see Appendix E.4.

**Proposition 6.1** (Necessity of adapting $T$ under prior shift; informal). *Assume $K$ is a rank-$m$ projector with $\{0, 1\}$-eigenvalues. Under Assumption 5.1, the Bayes-optimal estimator is $f_{\text{Bayes}} = \frac{1}{1+\lambda_\star}\Pi_{\mathcal{S}} r$, which varies with the test-time SNR $\frac{1}{\lambda_\star}$. Hence, no fixed horizon $T$ can be Bayes-optimal across two different priors.*

**Connection to phenomenon.** Prop. 6.1 underpins empirical reports that TTT is sensitive to the number of update steps (Akyürek et al., 2025; Zhao et al., 2023) and that, in particular under test-time condition shifts across scenarios, fixing $T$ can lead to catastrophic failure (Boudiaf et al., 2022).

**Proposition 6.2** ($\infty$-steps TTT can be sub-optimal; informal). *Assume the linearized quadratic prompt loss and $0 < \rho < 2/\lambda_{\max}(K)$. GD initialized at zero satisfies $\hat{f}_T \to \Pi_{\mathcal{S}} r$ as $T \to \infty$. If $\dim(\mathcal{S}) \geq 3$ (with known Gaussian noise level), there exists a shrinkage rule within $\mathcal{S}$ that reduces its squared-error risk.*

**Connection to phenomenon.** Prior experiments have reported that increasing the number of test-time adaptation steps can eventually hurt performance due to overfitting (Zhao et al., 2023; Alfarra et al., 2025). Prop. 6.2 provides a principled explanation of this.

**Proposition 6.3** (Irreducible error of subspace-mismatch;

informal). *For any target $f_\star \in \mathbb{R}^n$, the TTT estimator $\hat{f}_{g_T}$ satisfies* $\inf_{T\geq 0} \|\hat{f}_{g_T} - f_\star\|_2^2 \geq \|\Pi_{\mathcal{S}^\perp} f_\star\|_2^2$.

In the global-upstream regime (4), the representable prompt-space directions satisfy $\mathrm{range}(K^{(h,V)}) = \mathrm{range}(\mathcal{A}^{(h)}Z)$; see Lem. H.3 in the Appendix.

**Connection to phenomenon.** Prop. 6.3 formalizes a representational bottleneck, that is, regardless of hyperparameters, the correction $\hat{f}_g$ lies in $\mathcal{S}$, so components outside $\mathcal{S}$ are unavoidable. Consistent with this view, Zhang et al. (2025) discuss gains from scaling test-time state capacity (expanding $\mathcal{S}$ in our context). However, simply expanding the update space is not sufficient. Choi et al. (2022) report settings where full-parameter updates can degrade performance, motivating careful design of the update mechanism $A$ (and hence the induced $\mathcal{S}$).

> **Takeaway (C1-2): why $T$ and $A$ must adapt**
>
> A fixed $T$ can be Bayes-suboptimal under test-time prior shift; taking $T \to \infty$ can overfit; and any fixed representable subspace induces an irreducible mismatch floor.

In addition, Appendix G explains how the spectrum of $K(P_n)$ governs when a fixed $T$ can (or cannot) match Bayes shrinkage across modes via the gradient flow and random matrix theory (RMT).

## 7. Per-prompt update steps via PAC-Bayes

The first failure mode above concerns how far to adapt. Building on the evidence (marginal likelihood) framework (MacKay, 1992), we interpret $T$ as a hyperparameter in the implicit Bayesian formulation of Section 4 and analyze it via PAC-Bayes (Catoni, 2007; Alquier et al., 2024).

**Evidence family induced by TTT.** Fix a prompt-induced geometry $K \succeq 0$ and $\rho \in (0, 1/\lambda_{\max}(K))$. Motivated by Prop. 4.1, we consider the associated auxiliary evidence family in which $r$ is distributed according to the Gaussian marginal induced by the implicit prior:

$$r \mid (K,T) \sim \mathcal{N}(0, \Sigma_T), \quad \Sigma_T = \sigma^2(I - \rho K)^{-T}.$$

This induces a data-dependent score for $T$, given by the normalized negative log-evidence (ignoring $T$-independent constants) $\ell_T(r) := \frac{1}{2n}\big\{\log\det(\Sigma_T) + r^\top \Sigma_T^{-1} r\big\}$. We will use these scores to form a posterior over $T$ via a Gibbs construction introduced below. To analyze the statistical validity of selecting $T$ via this objective, we formalize the data generating process as follows.

**Assumption 7.1** (Reference model for residuals). Conditional on $K$, the true residual vector is Gaussian: $r \mid K \sim \mathcal{N}(0, \Sigma_\star)$, $\Sigma_\star \succ 0$.

Under this reference model, define the normalized population cross-entropy risk as $\mathcal{L}(T \mid K) := \mathbb{E}[\ell_T(r) \mid K]$. Since $\Sigma_\star$ is unknown, we use the observed prompt residuals $r$ to evaluate $\ell_T(r)$ across candidates and convert these scores into the Gibbs posterior over $T$ introduced next.

**Gibbs posterior over update steps.** Fix a finite candidate set $\mathcal{T} \subset \mathbb{N}_0$ and a prior $\pi_0$ on $\mathcal{T}$ with full support. For $\beta > 0$, define the Gibbs posterior $\nu_\beta(T \mid r) \propto \pi_0(T) \exp\{-\beta n \ell_T(r)\}$. This posterior can be used either to select a point estimate $T_{\mathrm{MAP}} \in \arg\max_T \nu_\beta(T \mid r)$ or to perform model averaging (mixture-of-filters (Dalalyan & Tsybakov, 2008)) $\bar{G} := \mathbb{E}_{T\sim\nu_\beta}[g_T(K)]$. When $\pi_0$ is uniform, $T_{\mathrm{MAP}}$ coincides with evidence minimization (Germain et al., 2016), defined as $T_{\mathrm{MAP}} \in \arg\min_{T\in\mathcal{T}} \ell_T(r)$.

Formally, this Gibbs posterior is defined as a Radon–Nikodym tilt of $\pi_0$; see Remark E.10.

**Theorem 7.2** (PAC-Bayes guarantee for selecting $T$). *Consider a point mass posterior $\nu = \delta_T$ and uniform prior $\pi_0$ on $\mathcal{T}$. Grant Assumption 7.1 and let $\mathcal{T}$ be finite. Define $A_T := \Sigma_\star^{1/2}\Sigma_T^{-1}\Sigma_\star^{1/2} \succeq 0$ and $\bar{v} := \sup_{T\in\mathcal{T}} \frac{1}{n}\mathrm{tr}(A_T^2)$. Then, for any $\delta \in (0,1)$, with probability at least $1 - \delta$ over $r \mid K$, choosing $\beta = 2\sqrt{\{\log|\mathcal{T}| + \log(1/\delta)\}/n\bar{v}}$ yields*

$$\mathcal{L}(T_{\mathrm{MAP}} \mid K) \leq \ell_{T_{\mathrm{MAP}}}(r) + \sqrt{\frac{\bar{v}\big(\log|\mathcal{T}| + \log(1/\delta)\big)}{n}}.$$

Thm. 7.2 controls the population negative log-evidence of the prompt-residual model. It is not an end-to-end theorem that the evidence-minimizing horizon is always query-risk optimal. The link to prediction is through the shared filter: Sec. 5 shows that the query correction uses the same $q_T(K)$, weighted by the prompt–query coupling $k_x$, while Sec. 8 handles query-dependent update directions. Thus, evidence selects the shrinkage level within the TTT filter family; query-aware subspace selection addresses the remaining query dependence. Under the GP reference model, App. E.5.3 further relates evidence closeness to a normalized Bayes gap.

> **Takeaway (C2): per-prompt $T$ selection**
>
> Selecting $T$ by prompt evidence enjoys a PAC-Bayes bound for the out-of-sample negative log-evidence, so tuning $T$ on the prompt does not overfit the prompt residuals.

## 8. Bayes-optimal subspace selection

Evidence selection chooses how far to move within a fixed prompt geometry; it does not by itself choose which geometry is most useful for the query. We next turn from the update horizon to the update subspace. The update

subspace should capture query-aware prompt signal since $\hat{h}_T(x) = k_x^\top q_T(K)\,r$ (Sec. 4) and improvements come primarily from eigenmodes aligned with $k_x$ (Thm. 5.2). We formalize this test-time update subspace as a Bayesian design variable, in the spirit of Attia et al. (2018).

**Assumption 8.1** (Gaussian benchmark of subspace). There exists a latent parameter correction $u_\star \in \mathbb{R}^p$ such that $u_\star \sim \mathcal{N}(0, \Sigma)$, $r = J_w u_\star + \varepsilon$, $\varepsilon \sim \mathcal{N}(0, \sigma^2 I_n)$, and $u_\star \perp \varepsilon$, where $\Sigma \succ 0$ is task-to-task variability and $J_w := \nabla_w s_w(P_n)|_{w=w_0} \in \mathbb{R}^{n \times p}$.

This assumption is the subspace version of Assumption 5.1 (see Lem. E.20). For a query input $x$, denote the linearized query correction by $f_x := j_x^\top u_\star$, where $j_x := \nabla_w f_{w_0}(x; P_n) \in \mathbb{R}^p$.

### 8.1. Bayes-optimal update subspace

To focus on query prediction, we assess an estimator $\widehat{u}(r)$ by the mean squared error of the linearized query correction $\mathbb{E}\big[\{j_X^\top(\widehat{u} - u_\star)\}^2\big]$, where the expectation is over the test-time query $X$ (conditioned on the prompt, unless stated otherwise). Define $Q := \mathbb{E}\big[j_X j_X^\top\big] \succeq 0$. Then $\mathbb{E}[\{j_X^\top(\widehat{u} - u_\star)\}^2] = \mathbb{E}[\|\widehat{u} - u_\star\|_Q^2]$ with $\|v\|_Q^2 := v^\top Q v$.

To align the subspace constraint with the query metric, we reparameterize via $v := Q^{1/2} u$ (restricted to $\mathrm{range}(Q)$ if $Q$ is singular), so that $\|u\|_Q^2 = \|v\|_2^2$. We choose an update subspace of dimension budget $k_{\text{budget}}$ by selecting a rank-$k_{\text{budget}}$ Euclidean projector $\widetilde{\Pi}(= \widetilde{\Pi}^\top = \widetilde{\Pi}^2)$ acting on $v$, and restrict estimators to $\widehat{v}(r) \in \mathrm{range}(\widetilde{\Pi})$.

Let $\bar{u}(r) := \mathbb{E}[u_\star \mid r]$ denote the Bayes estimator under the linear-Gaussian model, and set $\bar{v}(r) := Q^{1/2}\bar{u}(r)$. The key quantity for subspace design is how much recoverable signal the prompt provides about $u_\star$, captured by the covariance of the posterior mean: $\mathcal{I} := \mathrm{Var}(\bar{u}(r)) = \Sigma J_w^\top(J_w \Sigma J_w^\top + \sigma^2 I_n)^{-1} J_w \Sigma$. Its query-whitened version $\mathcal{I}_Q := \mathrm{Var}(\bar{v}(r)) = Q^{1/2}\mathcal{I}Q^{1/2}$ will determine the Bayes-optimal rank-$k_{\text{budget}}$ update subspace.

**Theorem 8.2** (Optimal rank-$k_{\text{budget}}$ update subspace). *Under Assumption 8.1, define the query-weighted Bayes risk*

$$\mathcal{R}(\widetilde{\Pi}) := \inf_{\widehat{v}(\cdot) \in \mathrm{range}(\widetilde{\Pi})} \mathbb{E}\Big[\|\widehat{v}(r) - Q^{1/2}u_\star\|_2^2\Big]$$
$$= \mathrm{tr}(Q\Sigma) - \mathrm{tr}(\widetilde{\Pi}\mathcal{I}_Q).$$

*Then, any minimizer over rank-$k_{\text{budget}}$ projectors $\widetilde{\Pi}$ is the projector onto the top-$k_{\text{budget}}$ eigenspace of $\mathcal{I}_Q$.*

To realize the designed subspace in the original coordinates $u$, map back from $v = Q^{1/2}u$. If $Q \succ 0$, the corresponding $u$-space subspace is $Q^{-1/2}\mathrm{range}(\widetilde{\Pi})$. If $Q$ is singular, interpret $Q^{-1/2}$ as the pseudoinverse on $\mathrm{range}(Q)$.

By the law of total variance, the $Q$-weighted total variability

of the correction $\mathrm{Var}(Q^{1/2}u_\star)$ decomposes into $\mathcal{I}_Q$ (recoverable from the prompt) and $\mathbb{E}\big[Q^{1/2}\mathrm{Var}(u_\star \mid r)\,Q^{1/2}\big]$ (irreducible). Hence, $\mathrm{tr}(\widetilde{\Pi}\mathcal{I}_Q)$ is the query-aligned posterior-mean variability recovered by the update subspace. This is large when the subspace both (i) contains directions whose posterior means are identifiable from the prompt and (ii) aligns with query sensitivities (via $Q$), consistent with the eigen-alignment findings of Thm. 5.2. A global (population-level) update mechanism can be obtained by averaging $\mathcal{I}_Q$ across prompts and applying a plug-in top-eigenspace estimator. Details and a martingale-type regret bound are deferred to Appendix E.6.

### 8.2. Block/head selection for Transformer

**Transformer partitioning.** To obtain a practical approximation to Thm. 8.2, we restrict the update projector to be aligned with a Transformer parameter partition (motivated by previous analyses showing that attention-head contributions are so uneven that many heads are prunable (Michel et al., 2019; Voita et al., 2019)). Specifically, we partition parameters into disjoint blocks (e.g., by layer/head-/matrix type), writing the corresponding Jacobian block columns by $J_w = [J^{(b)}]_{b \in \mathcal{B}_{\text{all}}}$, and restrict updates to a block-diagonal projector $\Pi = \mathrm{diag}\big(\Pi^{(b)}\big)_{b \in \mathcal{B}_{\text{all}}}$, which includes block on/off selection ($\Pi^{(b)} \in \{0, I_{p_b}\}$).

For block-diagonal $\Pi$, let $\mathcal{I}_{bb}$ denote the diagonal block of $\mathcal{I}$ for block $b$. If we specify $Q$ by a single query $x$ (i.e., $Q = j_x j_x^\top$) and perform on/off selection, then the risk reduction attributable to block $b$ is proportional to the score

$$j_x^{(b)\top}\mathcal{I}_{bb}j_x^{(b)} \qquad (\text{QUERY-AWARE}),$$

where $j_x^{(b)}$ is the restriction of $j_x$ to block $b$. We refer to ranking blocks by this score as QUERY-AWARE selection.

Under the isotropic correction prior $\Sigma = I_p$, we have $\mathcal{I} = J_w^\top(K_{\text{full}} + \sigma^2 I_n)^{-1}J_w$ with $K_{\text{full}} := J_w J_w^\top$ (the prompt geometry under full parameter updates). Let $K_{\text{full}}^{(b)} := J^{(b)}J^{(b)\top}$. Then the query-marginal block score becomes

$$\mathrm{tr}\big(\mathcal{I}_{bb}\big) = \mathrm{tr}\Big(K_{\text{full}}^{(b)}(K_{\text{full}} + \sigma^2 I_n)^{-1}\Big). \quad (\text{TRACE-TOPK})$$

Selecting the top-$k_{\text{budget}}$ blocks by this score corresponds to the TRACE-TOPK baseline in Sec. 9. In contrast, for a specific query $x$, the QUERY-AWARE score can be written as $k_x^{(b)\top}(K_{\text{full}} + \sigma^2 I_n)^{-1}k_x^{(b)}$, where $k_x^{(b)} := J^{(b)}j_x^{(b)}$ is the prompt-query coupling induced by block $b$. The factor $(K_{\text{full}} + \sigma^2 I_n)^{-1}$ downweights directions already explained by other blocks (or dominated by noise), making the score explicitly redundancy-aware and query-aligned.

For on/off selection $\Pi^{(b)} \in \{0, I_{p_b}\}$ under a block/parameter budget, we rank blocks by an appropriate score (e.g.,

QUERY-AWARE: $j_x^{(b)\top} \mathcal{I}_{bb} j_x^{(b)}$ when $Q = j_x j_x^\top$), optionally normalized by $p_b$. For a global update mechanism, use the query-averaged $Q = \mathbb{E}_X[j_X j_X^\top]$.

The exact query-specific QUERY-AWARE score requires one backward pass through the query output to obtain $j_x$. This overhead is additive rather than per-block: a single query backward pass gives $j_x$ for all blocks, after which the block-specific vectors $j_x^{(b)}$ are obtained by slicing. The remaining inverses and trace scores are in prompt space (dimension $n$) and can be implemented with Jacobian–vector products without materializing $J_w$. When per-query latency is undesirable, the query-averaged choice $Q = \mathbb{E}_X[j_X j_X^\top]$ gives an amortized global design that removes the per-query backward pass while remaining the Bayes-optimal solution for the averaged query objective; see Appendix H.4 for details.

> **Takeaway (C3): optimal update directions**
>
> For TTT prediction, the Bayes-optimal rank-$k_{budget}$ update subspace is the top-$k_{budget}$ eigenspace of $\mathcal{I}_Q$. For Transformer, this yields query-aware scores for attention block/head selection.

## 9. Experiments

The experiments are performed for mechanism-level checks of the theory rather than a comprehensive benchmark study. Experiment 1 tests whether evidence selects prompt-dependent horizons, and Experiment 2 tests whether query-aligned directions improve query prediction more reliably than prompt-fit directions.

**Experimental setup.** We use a digit-shift task with `distilgpt2` (Wolf et al., 2020).

We generate 2,000 tasks, each sampled from one of two settings: *clean* ($p_{noise} = 0.0$) and *noisy* ($p_{noise} = 0.55$). For each task, we sample a shift $s \sim \text{Unif}(0, \dots, 9)$, draw digits $x \sim \text{Unif}(0, \dots, 9)$, and set labels $y = (x + s) \bmod 10$. Each test instance contains $n = 10$ prompt pairs $(x, y)$ and an independent query digit $x$ from the same task. Prompt labels are corrupted independently with probability $p_{noise}$ by replacing $y$ with a uniformly random digit in $\{0, \dots, 9\}$.

We update only attention Value weights at the (layer, head) level via gradient descent on leave-one-out prompt residuals. Although the model outputs logits over digits, we treat this as a regression problem: labels are normalized to $\tilde{y} := y/9$ and the model's prediction is taken as the expected digit $\hat{y} = \sum_{j=0}^{9} \frac{j}{9} \text{softmax}(\ell)_j$. Full details are in Appendix I.

**(Exp1) Per-prompt $T$ selection.** To demonstrate claim **(C2)**, we compare ICL (no update), TTT with a fixed $T$ ($T = 8$), and evidence-based per-prompt selection ($T_{MAP}$)[1].

| Method | MSE ($\pm$ SE) | Mean $T$ |
|---|---|---|
| TTT (evidence, MLE-$\sigma$) | $0.0719 \pm 0.0029$ | 11.200 |
| TTT (evidence, fixed-$\sigma$) | $0.0722 \pm 0.0030$ | 9.793 |
| TTT ($T$=8) | $0.0734 \pm 0.0030$ | – |
| ICL ($T$=0) | $0.0747 \pm 0.0031$ | – |

*Table 1.* **Evidence-based TTT reduces risk (Exp1).** Comparison of MSE over 2,000 tasks. While fixed TTT yields marginal gains, adaptive $T_{MAP}$ (MLE-$\sigma$) achieves the lowest risk, consistent with Claims **(C1)**-**(C2)**.

We report averaged MSE across 2,000 tasks with SE denoting the standard error of the mean across tasks.

Table 1 shows that evidence-based $T$ selection improves over both ICL and fixed-$T$ TTT. Although the effect sizes are modest, the gains are statistically significant under a task-matched paired analysis. For each task, we compute the paired improvement as the baseline query MSE minus the method's query MSE, so that positive values indicate lower error. The best fixed horizon $T$=8 improves over ICL with a 95% bootstrap confidence interval $[0.000648, 0.001796]$ and paired permutation p-value $p = 5.0 \times 10^{-5}$. Both evidence-based variants further improve over this fixed-$T$ baseline: fixed-$\sigma$ evidence gives CI $[0.000545, 0.002894]$ with $p = 0.00325$, and MLE-$\sigma$ evidence gives CI $[0.000605, 0.003079]$ with $p = 0.00250$.

Figure 1 shows the empirical cumulative distribution function (ECDF) of the selected $T_{MAP}$ under the fixed-$\sigma$ evidence method, revealing substantial variation across prompts rather than a single globally optimal $T$. Moreover, $T_{MAP}$ responds to prompt signal-to-noise: in the noisy regime, evidence tends to select smaller $T$s to mitigate overfitting, whereas, in the clean regime, it favors larger $T$ to maximize adaptation benefits.

**(Exp2) Alignment-aware block selection.** We next investigate update-subspace design under a fixed budget of $k_{budget}$ blocks. We compare QUERY-AWARE, TRACE-TOPK, TRACE-BOTTOMK, and random selection.

Consistent with Claim **(C3)**, Figure 1 illustrates that the QUERY-AWARE selection consistently outperforms random selection. Notably, the TRACE-TOPK baseline, which prioritizes dominant prompt features without considering the query, often fits the prompt well but fails to generalize to the query. This confirms that effective adaptation requires not just capturing prompt variance, but capturing variance aligned with the query, as predicted by **(C1)**.

## 10. Discussion

We have presented a unified Bayes-risk framework for TTT, interpreting it as implicit Bayesian inference. We found

---

[1]We implement two evidence variants: a fixed-$\sigma$ plug-in and an MLE-$\sigma$ variant; details in Appendix.

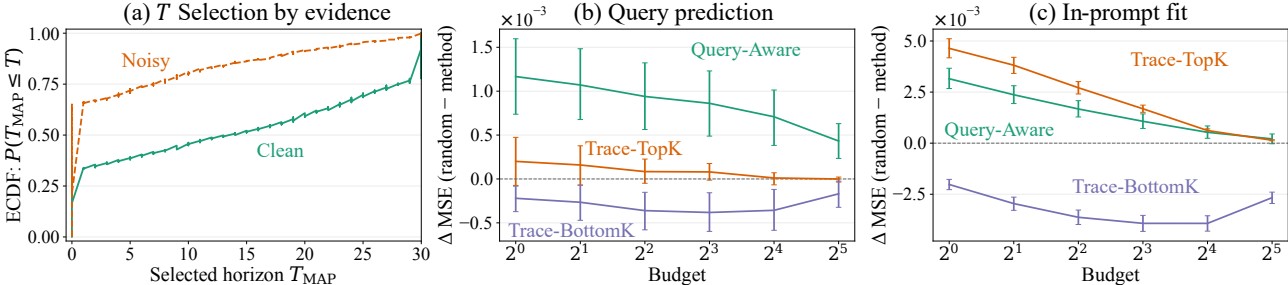

*Figure 1.* **(a) Evidence adaptively selects** $T$ **(Exp1).** ECDF of the selected $T_{\text{MAP}}$. The optimal $T$ is prompt-dependent, so adapting $T$ is preferable to using a fixed $T$ (Claims **(C1)**–**(C2)**). See Table 1 for predictive performance. **(b) Query gains depend on alignment (Exp2).** Query-MSE improvement relative to random selection ($\Delta > 0$ is better). QUERY-AWARE consistently outperforms TRACE-TOPK (Claims **(C1)**, **(C3)**). **(c) Prompt fit is a different objective (Exp2).** The same $\Delta$MSE evaluated using prompt-fit MSE. TRACE-TOPK tends to improve prompt fit, but this does not necessarily generalize to query improvement in (b). Error bars indicate standard error of the mean across tasks.

that predictive improvements depend critically on *filter match* and *eigen-alignment* **(C1)**, explaining why fixed hyperparameters often fail in practice. This Bayesian view also motivates evaluating $T$ through prompt evidence and yields a PAC-Bayes non-overfitting guarantee **(C2)**. Finally, the Bayes-optimal update subspace is characterized by an information-capture matrix, leading to query-aligned block-/head criteria for Transformers **(C3)**. Experiments support these implications, highlighting the roles of adaptive regularization and query-aligned update geometry. More broadly, as the kernel regime analysis (such as NTK) informed neural network engineering, we believe this work represents an important step toward making TTT more reliable in practice.

**Novelty boundary.** The implicit-Bayesian view is a conceptual lens rather than our sole technical contribution. Our novelty lies in the TTT-specific consequences of this lens: query-level Bayes-gap identities, PAC-Bayes step selection, and Bayes-optimal update-subspace design.

**What is expected to transfer.** We expect two lessons to be useful beyond the exact Gaussian-quadratic model. First, the update horizon should depend on the prompt, because prompts can have different signal-to-noise levels and different overfitting risks. Second, update directions should be chosen with the query in mind, because fitting the prompt alone need not improve the query. Our Transformer derivations in Secs. 3.3 and 8.2 show how these ideas can be expressed as attention block/head geometry and query-aware scores.

**Scope and future directions.** Our analysis is local. It uses restricted update parameters, a first-order approximation to the model, and a locally quadratic prompt objective. It is therefore best viewed as a theory of few-step TTT around a fixed pretrained model, rather than as a complete theory of arbitrary test-time adaptation. Appendix F.1 gives a finite-step deviation bound, but the present theory does not yet cover large updates, changing representations along the

adaptation path, full-parameter TTT, or the optimization dynamics of self-supervised TTT losses.

Several extensions are natural. First, one could move from a fixed local model to a trajectory-level analysis, where the tangent features, kernel, and prompt–query coupling are allowed to evolve during adaptation. Second, one could extend the Bayesian view to non-squared and self-supervised losses, such as entropy minimization, masked prediction, consistency losses, or pseudo-label training. This would require modeling pseudo-residuals, local curvature, and the bias introduced by noisy self-supervision. Third, one could develop scalable approximations to query-aware subspace selection, for example through amortized, low-rank, or query-averaged scores for choosing Transformer blocks, heads, or adapters. Finally, broader experiments on language, vision, and tabular tasks are needed to test which parts of the local Bayesian picture transfer to realistic nonlinear TTT.

## Acknowledgments

We thank the anonymous reviewers for their constructive comments. TW was partially supported by JSPS KAKENHI (26K21188), JST ACT-X (JPMJAX23CS) and RIKEN Incentive Research Project.

## Impact Statement

This paper presents work whose goal is to advance the field of Machine Learning. There are many potential societal consequences of our work, none of which we feel must be specifically highlighted here.

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

# A. Roadmap

This appendix provides full proofs and additional analyses supporting the main claims. We begin with a self-contained primer on the relevant decision-theoretic background. We then derive the Bayes risk identities and other theorems stated in the main text. Finally, we describe the experimental setup to facilitate reproducibility.

- **Notation and background.** We list the notation used throughout (App. B), provide a brief primer on decision theory (App. C), and relate our setting to classical early stopping (App. D).
- **Proofs for the main results.** We provide proofs for the decision-theoretic setup (App. E.1), the spectral-filter and implicit-prior equivalence and out-of-sample (query) forms (App. E.2), and Bayes-gap identities under GP benchmarks including the query-coupled form used in the main text (App. E.3). We provide rigorous versions of the three necessity propositions (App. E.4) and full PAC-Bayes details for evidence-based per-prompt horizon selection (App. E.5). Finally, we prove the Bayes-optimal subspace characterization, and include global-design and plug-in regret bounds (App. E.6).
- **Validity and robustness of the local model.** We bound finite-step deviations from linearization (App. F.1), extend filter identities beyond Gaussianity via second-moment arguments (App. F.2), and discuss anisotropic priors and preconditioning (App. F.3).
- **Interpretability: gradient flow and random-matrix theory.** We provide a gradient-flow interpretation of the update horizon, derive deterministic oracle curves via random-matrix theory, and connect spectral properties to horizon selection (App. G).
- **Details specific to Transformers and stability.** We derive Transformer Jacobian and geometry formulas (App. H.1), include a simple step-size stability bound (App. H.2), and characterize representable prompt subspaces for value updates (App. H.3).
- **Experimental details.** We describe task generation, optimization and evidence procedures, and the evaluation protocol (App. I).
- **Supplementary real-data check.** We include a small SST-2 experiment with `distilgpt2`, using 10 labeled prompt examples and one held-out query per task, to test the same two mechanisms—evidence-based horizon selection and query-aware head selection—outside the synthetic digit-shift setting (App. J).

# B. Notation list

This section collects notation for clarity.

**(Sec. 2)**

$\alpha$: Latent task variable, $\alpha \sim \pi$.

$\pi$: Meta-distribution over tasks.

$\mathcal{P}_\alpha$: Task-conditional data distribution on $\mathcal{X} \times \mathcal{Y}$.

$(X_i, Y_i)$: Labeled prompt samples, i.i.d. given $\alpha$.

$(X, Y)$: Query input/target random pair drawn from $\mathcal{P}_\alpha$.

$(x, y)$: A realized value of $(X, Y)$.

$P_n$: Labeled prompt of length $n$: $P_n = \{(X_i, Y_i)\}_{i=1}^n$.

$n$: Number of supervised prompt positions (residual terms), i.e., $r \in \mathbb{R}^n$ and $K \in \mathbb{R}^{n \times n}$. In our simplified setting, each labeled example corresponds to one supervised token position, so $n$ also equals the number of labeled examples.

$d_x$: Input dimension $\mathcal{X} \subset \mathbb{R}^{d_x}$.

$M$: Prediction rule mapping $(P_n, X) \mapsto \widehat{Y}$.

$\mathcal{R}(M)$: Bayes risk $\mathbb{E}[(M(P_n, X) - Y)^2]$.

$M_{\text{Bayes}}$: Bayes predictor $M_{\text{Bayes}}(P_n, X) = \mathbb{E}[Y \mid P_n, X]$.

$\mathcal{G}(M)$: Bayes gap $\mathbb{E}[(M - M_{\text{Bayes}})^2]$.

$\mathcal{V}$: Posterior variance $\mathbb{E}[\text{Var}(Y \mid P_n, X)]$.

$B$: Base predictor (typically the no-update ICL predictor).

$C$: Prompt-dependent correction rule, with $M = B + C$.

$\Delta_B$: Bayes correction relative to $B$: $\Delta_B(P_n, X) = M_{\text{Bayes}}(P_n, X) - B(P_n, X)$.

**(Sec. 3)**

$w, w_0$: Model parameters and their pretrained value.

$p$: Total number of parameters $w \in \mathbb{R}^p$.

$A$: Orthonormal basis of the update subspace, $A \in \mathbb{R}^{p \times d}$ with $A^\top A = I_d$.

$d$: Update-subspace dimension.

$\mathcal{U}$: Update subspace $\mathcal{U} = \text{range}(A)$.

$\Pi_{\mathcal{U}}$: Orthogonal projector onto $\mathcal{U}$, $\Pi_{\mathcal{U}} = AA^\top$.

$\theta$: Update parameters, $w(\theta) = w_0 + A\theta$.

$f_w(x; P_n)$: Prediction at input $x$ given prompt $P_n$ under parameters $w$.

$f_\theta(x; P_n)$: Linearized surrogate model.

$P_n^{(-i)}$: Prompt with the $i$th example removed.

$r$: Prompt residual vector: $r_i = y_i - B(P_n^{(-i)}, x_i)$.

$\phi(x; P_n)$: Jacobian feature $\phi(x; P_n) = \nabla_\theta f_\theta(x; P_n)|_{\theta=0}$.

$\Phi$: Prompt feature matrix in $\mathbb{R}^{n \times d}$, with rows $\phi(x_i; P_n^{(-i)})^\top$.

$L(\theta)$: Quadratic prompt loss $L(\theta) = (2\sigma^2)^{-1} \|r - \Phi\theta\|_2^2$.

$\theta_\star$: Latent update parameter signal in the local regression model $r = \Phi\theta_\star + \varepsilon$.

$\varepsilon$: Noise vector in the local regression model, $\varepsilon \sim \mathcal{N}(0, \sigma^2 I_n)$.

$\sigma^2$: Effective noise or temperature parameter.

$K$: Prompt-induced geometry (kernel) $K = K(P_n) = \Phi\Phi^\top \in \mathbb{R}^{n \times n}$.

$s_w(P_n)$: Vector of prompt supervision outputs: $s_w(P_n) = (s_1(w), \ldots, s_n(w))$.

$J_w$: Jacobian of prompt outputs $J_w = \nabla_w s_w(P_n)|_{w=w_0} \in \mathbb{R}^{n \times p}$.

$K_{\text{full}}$: Full (unrestricted) kernel: $K_{\text{full}} = J_w J_w^\top$.

$\mathcal{S}$: Representable prompt subspace: $\mathcal{S} = \text{range}(K)$.

$k$: Token index / supervised-position index, $k \in \{1, \ldots, n\}$.

$H$: Number of attention heads.

$h$: Attention head index.

$\alpha_{ks}^{(h)}$: Attention weight from token $s$ to token $k$ in head $h$.

$z_s$: Residual-stream representation of token $s$.

$Z$: Stack of residual-stream vectors: $Z = [z_1^\top; \ldots; z_n^\top]$.

$W_V^{(h)}, W_O^{(h)}$: Value and output matrices of head $h$.

$g_k^{(h)}$: Upstream gradient into head-$h$ output at token $k$ $g_k^{(h)} := \nabla_{o_k^{(h)}} s_k(w)|_{w_0}$.

$m_k^{(h)}$: Attention-mixed representation $m_k^{(h)} = \sum_{s \leq k} \alpha_{ks}^{(h)} z_s$.

$M^{(h)}$: Stack of attention-mixed representations: $M^{(h)} := \mathcal{A}^{(h)} Z$.

$\beta_k^{(h)}$: Projected gradient $\beta_k^{(h)} = (W_O^{(h)})^\top g_k^{(h)}$.

$\odot$: Hadamard (elementwise) product.

$\mathbf{B}^{(h)}$: Stack of projected gradients $\mathbf{B}^{(h)} = [\beta_1^{(h)\top}; \ldots; \beta_n^{(h)\top}]$.

**(Sec. 4)**

$t$: GD iteration index ($t = 0, 1, 2, \ldots$).

$T$: Update horizon (number of steps).

$\eta$: GD step size (in parameter/update space).

$\rho$: Normalized step size: $\rho = \eta/\sigma^2$.

$\theta_t$: GD iterate, initialized at $\theta_0 = 0$.

$\hat{f}_t$: Prompt-space correction: $\hat{f}_t = \Phi\theta_t$.

$G_T$: TTT smoother: $G_T = I_n - (I_n - \rho K)^T$.

$g_T(\lambda)$: GD spectral filter: $g_T(\lambda) = 1 - (1 - \rho\lambda)^T$.

$\hat{f}_T$: $T$-step prompt correction: $\hat{f}_T = G_T r$.

$U, \{\lambda_i\}$: Eigen-decomposition of $K$: $K = U\text{diag}(\lambda_1, \ldots, \lambda_n)U^\top$.

$u_i$: Eigenvector of $K$ corresponding to the eigenvalue $\lambda_i$ (i.e., the $i$-th column of $U$).

$g(K)$: Functional calculus of $g$; namely, $g(K) = U\text{diag}(g(\lambda_i))U^\top$.

$A \succeq 0$, $A \preceq B$: For symmetric matrices, $A \succeq 0$ means that $A$ is positive semidefinite, and $A \preceq B$ means that $B - A$ is positive semidefinite.

$\lambda_{\max}(A)$: The largest eigenvalue of a symmetric matrix $A$.

$\|A\|_{\text{op}}$: The operator norm of $A$ (equal to the largest singular value; for symmetric $A$, it equals $\max_i |\lambda_i(A)|$).

$A^\dagger$: The Moore–Penrose pseudoinverse.

$\log \det^*(A)$: For a positive semidefinite matrix $A$ with eigenvalues $\{\mu_i\}$, $\log \det^*(A) := \sum_{i:\mu_i>0} \log \mu_i$.

**(Sec. 5)**

$f_\star$: Latent true prompt-level correction.

$\tau^2$: Prior variance / signal strength.

$\lambda_\star$: test-time noise-to-signal ratio $\lambda_\star = \sigma^2/\tau^2$.

$g_\star(\lambda)$: Bayes-optimal shrinkage: $g_\star(\lambda) = \lambda/(\lambda + \lambda_\star)$.

$\hat{h}_T(x)$: Query-point correction induced by $T$-step TTT: $\hat{h}_T(x) = \phi(x; P_n)^\top \theta_T = k_x^\top q_T(K) r$.

$f_{\text{Bayes}}$: Bayes posterior mean.

$k_x$: Prompt–query kernel vector: $k_x = \Phi \phi(x; P_n) \in \mathbb{R}^n$.

$q_T(\lambda)$: Query-side TTT filter: $q_T(\lambda) = \frac{1-(1-\rho\lambda)^T}{\lambda}$ for $\lambda > 0$ (and $q_T(0) = \rho T$).

$q_\star(\lambda)$**:** Bayes-optimal resolvent: $q_\star(\lambda) = (\lambda + \lambda_\star)^{-1}$.

$\kappa_x$**:** Query-point kernel scalar in the GP benchmark, so that $\begin{pmatrix} K & k_x \\ k_x^\top & \kappa_x \end{pmatrix} \succeq 0$ and $\mathrm{Var}(f_x \mid K, k_x, \kappa_x) = \tau^2 \kappa_x$.

$\alpha_{ks}^{(h)}$**:** Attention weight (head $h$) from token $s$ to token $k$.

$\mathcal{A}^{(h)}$**:** Attention matrix with entries $\mathcal{A}_{ks}^{(h)} = \alpha_{ks}^{(h)}$.

**(Sec. 7)**

$\mathcal{T}$**:** Finite candidate set of horizons ($\subset \mathbb{N}_0$).

$\pi_0$**:** Prior distribution over $\mathcal{T}$.

$\Sigma_T$**:** Evidence covariance: $\Sigma_T = \sigma^2(I - \rho K)^{-T}$.

$\ell_T(r)$**:** Normalized negative log-evidence for horizon $T$.

$\mathcal{L}(T \mid K)$**:** Population cross-entropy risk: $\mathcal{L}(T \mid K) = \mathbb{E}[\ell_T(r) \mid K]$.

$\nu_\beta(T \mid r)$**:** Gibbs posterior: $\nu_\beta(T \mid r) \propto \pi_0(T) \exp\{-\beta n\, \ell_T(r)\}$.

$\beta$**:** Gibbs temperature parameter in the PAC-Bayes / evidence selection procedure.

$\Sigma_\star$**:** True/reference covariance of $r$ conditional on $K$ in Assumption 7.1.

$A_T$**:** Information matrix: $A_T := \Sigma_\star^{1/2} \Sigma_T^{-1} \Sigma_\star^{1/2} \succeq 0$.

$v_T$**:** Variance proxy: $v_T := \frac{1}{n}\mathrm{tr}(A_T^2)$.

$\bar{v}$**:** Max variance proxy: $\bar{v} := \sup_{T \in \mathcal{T}} v_T$.

$T_{\mathrm{MAP}}$**:** Evidence/MAP-selected horizon: $T_{\mathrm{MAP}} \in \arg\min_{T \in \mathcal{T}} \ell_T(r)$ (uniform $\pi_0$).

**(Sec. 8)**

$u_\star$**:** Latent parameter correction, $u_\star \sim \mathcal{N}(0, \Sigma)$.

$\Sigma$**:** Prior covariance of parameter corrections.

$\varepsilon$**:** Observation noise, $\varepsilon \sim \mathcal{N}(0, \sigma^2 I_n)$.

$\bar{u}(r)$**:** Posterior mean: $\bar{u}(r) = \mathbb{E}[u_\star \mid r]$.

$\Pi$**:** Rank-$d$ orthogonal projector defining an update subspace.

$\widehat{u}_\Pi(r)$**:** Constrained Bayes estimator: $\widehat{u}_\Pi(r) = \Pi\, \bar{u}(r)$.

$\mathcal{I}$**:** Information-capture matrix: $\mathcal{I} = \mathrm{Var}(\bar{u}(r)) = \Sigma J_w^\top (J_w \Sigma J_w^\top + \sigma^2 I_n)^{-1} J_w \Sigma$.

$j_X$**:** Query Jacobian: $j_X = \nabla_w f_{w_0}(X; P_n) \in \mathbb{R}^p$.

$j_x$**:** Query Jacobian at a specific query input $x$: $j_x := \nabla_w f_{w_0}(x; P_n) \in \mathbb{R}^p$.

$Q$**:** Query sensitivity (semi)metric: $Q := \mathbb{E}[j_X j_X^\top] \succeq 0$ (conditioning on the prompt if desired).

$v$**:** $Q$-whitened parameter coordinates: $v := Q^{1/2} u$ (restricted to $\mathrm{range}(Q)$ if $Q$ is singular).

$\widetilde{\Pi}$**:** Rank-$k_{\text{budget}}$ Euclidean orthogonal projector acting on $v$-space (defines the update subspace in the $Q$-whitened coordinates).

$k_{\text{budget}}$**:** Rank / budget of the update subspace (number of adaptable directions/blocks).

$\mathcal{I}_Q$**:** Query-whitened information-capture matrix: $\mathcal{I}_Q := \mathrm{Var}(\bar{v}(r)) = Q^{1/2} \mathcal{I} Q^{1/2} \succeq 0$.

# C. Primer on Decision Theory and Bayesian Estimation

To render our theoretical framework self-contained and accessible to readers less familiar with statistical decision theory (Lehmann & Casella, 1998; Robert, 2007; Murphy, 2022), we briefly review predictive risk, Bayes risk under our mixture task model, and the Bayes posterior (risk-minimizing) estimator. Throughout this primer, we use the squared loss and assume square-integrability (e.g., $\mathbb{E}[Y^2] < \infty$) so that conditional expectations under squared loss are well-defined. We highlight the minimal assumptions needed for each identity and indicate where they are relaxed in later appendices.

## C.1. Risk and Bayes-optimal prediction for a fixed distribution

Let $(\Omega, \mathcal{F}, \mathbb{P})$ be a probability space and let $(X, Y) : \Omega \to \mathcal{X} \times \mathbb{R}$ be a random pair with joint law $\mathcal{P}$. A (deterministic) predictor is a measurable map $M : \mathcal{X} \to \mathbb{R}$, and the induced prediction is the random variable $\widehat{Y} := M(X)$.

**Predictive risk.** Assume $\mathbb{E}[M(X)^2] < \infty$. The (squared-loss) risk (Bishop, 2006; Murphy, 2022) of $M$ under $\mathcal{P}$ is

$$R_\mathcal{P}(M) := \mathbb{E}\big[(M(X) - Y)^2\big].$$

Even the best possible predictor incurs nonzero risk when $Y$ is noisy given $X$.

**Bayes predictor under squared loss.** Any minimizer of $R_\mathcal{P}(M)$ over measurable $M$ with $\mathbb{E}[M(X)^2] < \infty$ is given (uniquely $\mathcal{P}_X$-a.s.) by the conditional expectation:

$$M_{\mathrm{Bayes},\mathcal{P}}(X) := \mathbb{E}[Y \mid X].$$

Equivalently, any two minimizers agree $\mathcal{P}_X$-almost surely.

## C.2. Bayes risk under the mixture task model

In the context of TTT and ICL, we do not face a single static distribution. Instead, we face a distribution of tasks (e.g., a mixture of different functions or domains (Wakayama & Suzuki, 2025)).

As in (1), let $\alpha \sim \pi$ and, conditional on $\alpha$, let $(X_i, Y_i)_{i=1}^n$ and $(X, Y)$ be conditionally i.i.d. with law $\mathcal{P}_\alpha$. Let $P_n := \{(X_i, Y_i)\}_{i=1}^n$ and let the available information be $\sigma$-field $\sigma(P_n, X)$. A predictor is any $\sigma(P_n, X)$-measurable map:

$$M : (P_n, X) \mapsto \widehat{Y} \in \mathbb{R},$$

with $\mathbb{E}[\widehat{Y}^2] < \infty$.

**Bayes risk (Robert, 2007; Murphy, 2022).** Assume $\mathbb{E}[M(P_n, X)^2] < \infty$. We evaluate $M$ by the task-averaged (Bayes) risk

$$\mathcal{R}(M) := \mathbb{E}_{\alpha \sim \pi, P_n \sim \mathcal{P}_\alpha^{\otimes n}, (X,Y) \sim \mathcal{P}_\alpha} \left[ (M(P_n, X) - Y)^2 \right],$$

where the expectation is taken with respect to the joint distribution of $(\alpha, P_n, X, Y)$ induced by the mixture model in (1). Here, $\mathcal{P}_\alpha^{\otimes n}$ denotes n-fold product (tensor) measure of $\mathcal{P}_\alpha$.

**Bayes predictor under the mixture model.** Under squared loss, any minimizer of $\mathcal{R}(M)$ over measurable maps $M(P_n, X)$ (with finite second moment) is given (uniquely a.s.) by the conditional expectation

$$M_{\text{Bayes}}(P_n, X) := \mathbb{E}[Y \mid P_n, X].$$

**Bayes-risk decomposition (Prop. 2.1).** For any predictor $M$ with finite second moment, the Bayes risk decomposes as

$$\mathcal{R}(M) = \mathcal{G}(M) \text{ (Bayes gap)} + \mathcal{V} \text{ (posterior variance).}$$

- The **posterior variance** ($\mathcal{V}$) represents the inherent uncertainty in the targets $Y$ that cannot be resolved even with infinite compute and perfect knowledge of the prior given $\sigma(P_n, X)$. It is constant with respect to the model $M$.

- The **Bayes Gap** ($\mathcal{G}$) measures how closely the model $M$ approximates the optimal Bayes predictor. In other words, this is the excess risk to the Bayes predictor.

To improve performance via TTT, we must reduce the Bayes Gap. This means our adapted model must essentially perform a better approximation of the conditional expectation $\mathbb{E}[Y \mid P_n, X]$ than the base model.

**Existence and regular conditional distributions** Let $\mathcal{G} \subseteq \mathcal{F}$ be a sub-$\sigma$-field and let $Y \in L^1(\mathbb{P})$. A conditional expectation $\mathbb{E}[Y \mid \mathcal{G}]$ is any $\mathcal{G}$-measurable random variable $Z$ satisfying $\int_A Z \, d\mathbb{P} = \int_A Y \, d\mathbb{P}$ for all $A \in \mathcal{G}$. Existence follows from the Radon–Nikodym theorem and $Z$ is unique $\mathbb{P}$-a.s.; throughout we fix a $\mathcal{G}$-measurable version when writing $\mathbb{E}[Y \mid P_n, X]$.

In our setting $(P_n, X, Y)$ take values in standard Borel spaces, so there exists a regular conditional distribution $\mathbb{P}(dy \mid P_n, X)$ and, whenever $Y$ is integrable, $\mathbb{E}[Y \mid P_n, X] = \int y \, \mathbb{P}(dy \mid P_n, X)$ holds $\sigma(P_n, X)$-a.s.

# D. Relation to classical optimization theory

Classical early stopping and iterative regularization analyzes gradient descent on a fixed quadratic problem, yielding the spectral filter form $\hat{f}_T = g_T(K)r$ with $g_T(\lambda) = 1 - (1 - \rho\lambda)^T$ Yao et al. (e.g., 2007); Engl et al. (e.g., 2000). We reuse this identity, but our TTT setting differs in three critical ways.

**(1) The geometry $K$ is not fixed.** Under our local linearization, the prompt-space geometry is $K(P_n) = \Phi\Phi^\top = J_w \Pi_\mathcal{U} J_w^\top$. Unlike classical analyses where $K$ (or the design) is exogenous, here $K$ varies across prompts and changes with the update mechanism $\Pi_\mathcal{U}$ (which blocks/heads/directions are adaptable). Thus, TTT behavior is inherently prompt- and mechanism-dependent.

**(2) The target is query prediction, not only in-sample fitting.** Beyond the prompt correction $\hat{f}_T$, prediction relies on the query-point correction $\hat{h}_T(x) = k_x^\top q_T(K)r$ with $k_x = \Phi\,\phi(x; P_n)$. Hence, success depends not only on filter choice (via $q_T$) but also on prompt–query coupling: only eigenmodes aligned with $k_x$ affect the query (Thm. 5.2).

**(3) $T$ and update directions are on-the-fly objects.** In TTT, a fixed $T$ can be Bayes-suboptimal under test-time prior shift, and $T \to \infty$ can overfit (Sec. 6), motivating per-prompt $T$ selection (Sec. 7). Moreover, no choice of $T$ can overcome subspace mismatch: corrections lie in $\mathcal{S}(P_n) = \text{range}(K(P_n))$, so update-direction design is essential.

Together, these distinctions explain why step sensitivity and update-direction choices are intrinsic to TTT, rather than secondary hyperparameter details.

# E. Proofs (by main-text order)

## E.1. Proofs for Sec. 2

This appendix provides proofs for the decision-theoretic statements in Sec. 2. Throughout, expectations are with respect to the joint distribution induced by the mixture model

(1).

### E.1.1. PROOF OF PROP. 2.1

*Proof.* Let $M_{\text{Bayes}}(P_n, X) := \mathbb{E}[Y \mid P_n, X]$, and $\xi := Y - M_{\text{Bayes}}(P_n, X)$. Then $\mathbb{E}[\xi \mid P_n, X] = 0$ and $Y = M_{\text{Bayes}}(P_n, X) + \xi$. For any predictor $M(P_n, X)$ with $\mathbb{E}[M(P_n, X)^2] < \infty$,

$$\begin{aligned}
&\big(M(P_n, X) - Y\big)^2 \\
&= \big(M(P_n, X) - M_{\text{Bayes}}(P_n, X) - \xi\big)^2 \\
&= \big(M(P_n, X) - M_{\text{Bayes}}(P_n, X)\big)^2 + \xi^2 \\
&\quad - 2\xi\big(M(P_n, X) - M_{\text{Bayes}}(P_n, X)\big).
\end{aligned}$$

Taking expectation and using the tower property,

$$\begin{aligned}
\mathbb{E}\big[\xi\big(M - M_{\text{Bayes}}\big)\big] &= \mathbb{E}\big[\mathbb{E}\big[\xi\big(M - M_{\text{Bayes}}\big) \mid P_n, X\big]\big] \\
&= \mathbb{E}\big[(M - M_{\text{Bayes}}) \, \mathbb{E}[\xi \mid P_n, X]\big] \\
&= 0.
\end{aligned}$$

Therefore,

$$\mathcal{R}(M) = \mathbb{E}\Big[\big(M - M_{\text{Bayes}}\big)^2\Big] + \mathbb{E}\big[\xi^2\big].$$

Finally, $\mathbb{E}[\xi^2] = \mathbb{E}[\text{Var}(Y \mid P_n, X)]$ by definition of conditional variance, and it does not depend on $M$. This yields (2). □

## E.2. Proofs and additional details for Sec. 4

This appendix derives the spectral-filter representation of TTT and proves the equivalence to Gaussian posterior means stated in Sec. 4. Before proceeding, several points should be noted.

**Beyond squared loss.** Beyond squared loss, an analogous one-step update can be written by replacing the residual $r$ with the scaled pseudo-residual $\tilde{r} := -\sigma^2 \nabla_s \ell(s_{w_0}(P_n), y)$ evaluated at $\theta = 0$. Extending the exact multi-step spectral-filter identities to general losses requires additional local curvature approximations (e.g., Gauss–Newton/Hessian freezing), which we leave as future work.

**Well-posedness of the prior.** Since $g_T(0) = 0$, the operator $G_T = g_T(K)$ vanishes on $\text{null}(K)$ and hence $\hat{f}_T \in \text{range}(K)$ for all $T$. The implicit prior covariance $C_T$ is supported on $\text{range}(K)$. It has eigenvalue 0 on $\text{null}(K)$ and positive eigenvalues only on $\text{range}(K)$.

### E.2.1. DERIVATION OF (6)

*Proof.* Recall the GD update in update parameters (Sec. 4):

$$\theta_{t+1} = \theta_t + \rho\, \Phi^\top (r - \Phi\theta_t), \quad \rho = \eta/\sigma^2, \quad \theta_0 = 0.$$

Define $\hat{f}_t := \Phi\theta_t \in \mathbb{R}^n$. Multiplying both sides by $\Phi$ gives the prompt-space recursion

$$\hat{f}_{t+1} = \hat{f}_t + \rho\, \Phi\Phi^\top (r - \Phi\theta_t) = \hat{f}_t + \rho K(r - \hat{f}_t).$$

Rearranging yields

$$r - \hat{f}_{t+1} = (I_n - \rho K)(r - \hat{f}_t).$$

Since $\hat{f}_0 = 0$, we have $r - \hat{f}_0 = r$, and iterating gives $r - \hat{f}_T = (I_n - \rho K)^T r$, hence $\hat{f}_T = \big(I_n - (I_n - \rho K)^T\big)r$.

Diagonalizing $K = U\text{diag}(\lambda_i)U^\top$ implies

$$\begin{aligned}
G_T &= U\,\text{diag}\big(1 - (1 - \rho\lambda_1)^T, \dots, 1 - (1 - \rho\lambda_n)^T\big)\,U^\top \\
&= U\,\text{diag}\big(g_T(\lambda_1), \dots, g_T(\lambda_n)\big)\,U^\top,
\end{aligned}$$

where $g_T(\lambda) = 1 - (1 - \rho\lambda)^T$, which yields the statement.

Finally, if $0 < \rho < 2/\lambda_{\max}(K)$, then all eigenvalues of $I_n - \rho K$ lie in $(-1, 1]$, and if moreover $0 < \rho < 1/\lambda_{\max}(K)$, then $0 \le 1 - \rho\lambda_i < 1$ for all $i$, so $g_T(\lambda_i) \in [0, 1)$ for all finite $T$. □

### E.2.2. FULL PROOF OF PROP. 4.1

*Proof.* Recall $0 < \rho < 1/\lambda_{\max}(K)$. Let $K = U\text{diag}(\lambda_1, \dots, \lambda_n)U^\top$. Then

$$G_T = U\,\text{diag}\big(g_T(\lambda_1), \dots, g_T(\lambda_n)\big)\,U^\top,$$

$0 \le g_T(\lambda_i) < 1$ for all $i$.

Define the implicit prior covariance $C_T$:

$$C_T := \sigma^2\, G_T\, (I_n - G_T)^{-1}.$$

In the eigenbasis of $K$ (equivalently of $G_T$), the matrix $(I_n - G_T)^{-1}$ is diagonal with entries

$$(1 - g_T(\lambda_i))^{-1} = \begin{cases} \frac{1}{1 - g_T(\lambda_i)}, & \lambda_i > 0, \\ 1, & \lambda_i = 0, \end{cases}$$

but since $g_T(0) = 0$, the product $g_T(0)(1 - g_T(0))^{-1} = 0$. Hence $C_T$ has the form of $C_T = U\text{diag}(c_{T,1}, \dots, c_{T,n})U^\top$, where

$$c_{T,i} = \begin{cases} \sigma^2\, \dfrac{g_T(\lambda_i)}{1 - g_T(\lambda_i)}, & \lambda_i > 0, \\ 0, & \lambda_i = 0. \end{cases}$$

Now consider the Gaussian latent-variable model $r = f + \varepsilon$ with $f \sim \mathcal{N}(0, C_T)$ and $\varepsilon \sim \mathcal{N}(0, \sigma^2 I_n)$ independent. Working in the eigenbasis of $C_T$, we have coordinate-wise for $\tilde{r} := U^\top r$, $\tilde{f} := U^\top f$, and $\tilde{\varepsilon} := U^\top \varepsilon$:

$$\tilde{r}_i = \tilde{f}_i + \tilde{\varepsilon}_i, \quad \tilde{f}_i \sim \mathcal{N}(0, c_{T,i}), \quad \tilde{\varepsilon}_i \sim \mathcal{N}(0, \sigma^2).$$

For each coordinate, the posterior mean is given by the scalar Gaussian conditioning formula (Rasmussen & Williams, 2006; Bishop, 2006):

$$\mathbb{E}[\tilde{f}_i \mid \tilde{r}_i] = \frac{c_{T,i}}{c_{T,i} + \sigma^2}\, \tilde{r}_i.$$

If $\lambda_i > 0$, then by the definition of $c_{T,i}$,

$$\frac{c_{T,i}}{c_{T,i} + \sigma^2} = \frac{\sigma^2 \frac{g_T(\lambda_i)}{1-g_T(\lambda_i)}}{\sigma^2 \frac{g_T(\lambda_i)}{1-g_T(\lambda_i)} + \sigma^2} = g_T(\lambda_i).$$

If $\lambda_i = 0$, then $c_{T,i} = 0$ and also $g_T(0) = 0$, so the identity still holds. Therefore,

$$\mathbb{E}[\tilde{f}_i \mid \tilde{r}_i] = g_T(\lambda_i)\tilde{r}_i \quad \text{for all } i,$$

and transforming back gives

$$\mathbb{E}[f \mid r] = U\mathrm{diag}\big(g_T(\lambda_1),\ldots,g_T(\lambda_n)\big)U^\top r = \hat{f}_T,$$

where the last equality uses (6). $\qquad\square$

### E.2.3. OUT-OF-SAMPLE (QUERY) FORM OF $T$-STEP GD UNDER THE PROMPT GEOMETRY

We record the kernel-linear form of the $T$-step prediction at a query input $x$ under the linearized model (Assumption 3.2). This form is used by the predictive Bayes-gap identity in Appendix E.3.2.

**Proposition E.1** (Out-of-sample form of TTT). *Work under Assumptions 3.1–3.2 and the quadratic prompt loss. Let $K = \Phi\Phi^\top \in \mathbb{R}^{n\times n}$ and let $r \in \mathbb{R}^n$ be the prompt residual vector. For a query input $x$, define the prompt–query kernel vector*

$$k_x := \Phi\, \phi(x; P_n) \in \mathbb{R}^n,$$

*so $(k_x)_i = \langle \phi(x_i; P_n^{(-i)}), \phi(x; P_n)\rangle$. Let $\{\theta_t\}_{t\geq 0}$ be the GD iterates (5) with $\theta_0 = 0$, and define the query correction $\hat{h}_T(x) := \phi(x; P_n)^\top \theta_T$. Then for every $T \geq 1$,*

$$\hat{h}_T(x) = k_x^\top q_T(K)\, r, \quad q_T(K) := \rho \sum_{t=0}^{T-1}(I_n - \rho K)^t, \quad (7)$$

*where $\rho = \eta/\sigma^2$. Equivalently, if $K = U\mathrm{diag}(\lambda_1,\ldots,\lambda_n)U^\top$, then $\hat{h}_T(x) = \sum_{i=1}^n (u_i^\top k_x)\, q_T(\lambda_i)\, (u_i^\top r)$, where*

$$q_T(\lambda) = \begin{cases} \dfrac{1-(1-\rho\lambda)^T}{\lambda}, & \lambda > 0, \\[2mm] \rho T, & \lambda = 0. \end{cases} \quad (8)$$

*Moreover, on the prompt points one recovers $\hat{f}_T = g_T(K)r$ as $\hat{f}_T = Kq_T(K)r$.*

*Proof.* Unrolling the GD recursion (5) gives

$$\theta_T = \rho \sum_{t=0}^{T-1}(I_d - \rho\Phi^\top\Phi)^t\, \Phi^\top r.$$

Therefore,

$$\hat{h}_T(x) = \phi(x)^\top \theta_T = \rho \sum_{t=0}^{T-1}\phi(x)^\top(I_d - \rho\Phi^\top\Phi)^t\, \Phi^\top r.$$

Using the identity $(I_n - \rho K)\Phi = \Phi(I_d - \rho\Phi^\top\Phi)$ and induction, one has $(I_n - \rho K)^t\Phi = \Phi(I_d - \rho\Phi^\top\Phi)^t$. Hence

$$\begin{aligned}
\phi(x)^\top(I_d - \rho\Phi^\top\Phi)^t\,\Phi^\top r &= \big(\Phi(I_d - \rho\Phi^\top\Phi)^t\phi(x)\big)^\top r \\
&= \big((I_n - \rho K)^t k_x\big)^\top r \\
&= k_x^\top(I_n - \rho K)^t r.
\end{aligned}$$

Substituting back yields (7). Diagonalizing $K$ gives the scalar form (8). Finally, for $\lambda > 0$ we have $\lambda q_T(\lambda) = 1 - (1 - \rho\lambda)^T = g_T(\lambda)$, and $g_T(0) = 0$, so $Kq_T(K) = g_T(K)$ on $\mathrm{range}(K)$, implying $\hat{f}_T = Kq_T(K)r$ on the prompt points. $\qquad\square$

### E.3. Proofs and additional details for Sec. 5

This appendix proves the spectral Bayes-gap identity (Thm. E.4 (prompt values)) and records its query-point analogue under a standard GP extension.

### E.3.1. PROMPT-VALUE (IN-SAMPLE) IDENTITIES: BAYES-OPTIMAL SHRINKAGE AND FILTER-MISMATCH GAP

This subsection identifies the Bayes-optimal predictor and shows that the Bayes gap of any spectral estimator is a weighted squared filter mismatch.

Fix a prompt $P_n$ and the associated geometry $K \succeq 0$. Recall the prompt residuals $r \in \mathbb{R}^n$ from Sec. 3. We model the (unknown) latent prompt-level correction as a random vector $f_\star \in \mathbb{R}^n$ and adopt the following benchmark prior.

**Assumption E.2** (Aligned GP benchmark on prompt values). Conditioned on $K$, the latent prompt-level correction and observed residuals satisfy $f_\star \mid K \sim \mathcal{N}(0, \tau^2 K)$, $r = f_\star + \varepsilon$, $\varepsilon \sim \mathcal{N}(0, \sigma^2 I_n)$, and $f_\star \perp \varepsilon$, for some $\tau^2 > 0$ and effective noise level $\sigma^2 > 0$.

The key structural assumption is alignment: $\mathrm{Cov}(f_\star \mid K)$ shares eigenvectors with $K$ (here $\mathrm{Cov}(f_\star \mid K) = \tau^2 K$). This is what makes the Bayes-optimal estimator a spectral shrinkage rule on $K$ and enables the exact filter-mismatch identities below.

Within this benchmark, we model test-time prior shift as changes in the effective SNR $\tau^2/\sigma^2$ (equivalently $\lambda_\star$), which shifts the Bayes-optimal shrinkage $g_\star$.

**Proposition E.3** (Bayes-optimal estimator on prompt values). *Under Assumption E.2 with $K = U\mathrm{diag}(\lambda_1, \ldots, \lambda_n)U^\top$, conditioned on $(K, r)$, the Bayes-optimal estimator of $f_\star$ under squared loss is the posterior mean $f_{\mathrm{Bayes}} = U\,\mathrm{diag}\big(g_\star(\lambda_1), \ldots, g_\star(\lambda_n)\big)\,U^\top r$, where $g_\star(\lambda) := \lambda/(\lambda + \lambda_\star)$.*

The Bayes-optimal estimator is a spectral shrinkage rule: each eigenmode with eigenvalue $\lambda$ is scaled by $g_\star(\lambda) \in [0, 1]$, where $\lambda_\star$ encodes the test-time noise-to-signal ratio. This provides a precise target for adaptation: a procedure helps when it moves its spectral shrinkage closer to $g_\star$ on the eigenvalues of $K(P_n)$.

We next quantify the Bayes gap of any spectral estimator under the GP benchmark.

Let $K = U\mathrm{diag}(\lambda_1, \ldots, \lambda_n)U^\top$. Given a measurable map $g : [0, \infty) \to \mathbb{R}$ with $g(0) = 0$, define the spectral estimator $\hat{f}_g := g(K)r$, where $g(K) := U\,\mathrm{diag}\big(g(\lambda_1), \ldots, g(\lambda_n)\big)U^\top$.

**Theorem E.4** (Bayes risk and Bayes gap in the spectral domain). *Assume Assumption E.2 and condition on $K = U\mathrm{diag}(\lambda_i)U^\top$. Let $\hat{f}_g = g(K)r$ be any spectral estimator. Then*

$$\mathbb{E}\Big[\|\hat{f}_g - f_\star\|_2^2 \,\Big|\, K\Big] = \sum_{i=1}^{n} \big[\tau^2\lambda_i(1 - g(\lambda_i))^2 + \sigma^2 g(\lambda_i)^2\big]. \tag{9}$$

*Moreover, the unique minimizer over $g(\lambda_i)$ is $g_\star(\lambda_i) = \lambda_i/(\lambda_i + \lambda_\star)$, and the excess Bayes risk (equivalently, Bayes gap to $f_{\mathrm{Bayes}}$) satisfies the exact identity*

$$\mathbb{E}\Big[\|\hat{f}_g - f_\star\|_2^2 \,\Big|\, K\Big] - \mathbb{E}\big[\|f_{\mathrm{Bayes}} - f_\star\|_2^2 \,\big|\, K\big]$$
$$= \sum_{i=1}^{n}(\tau^2\lambda_i + \sigma^2)\big(g(\lambda_i) - g_\star(\lambda_i)\big)^2. \tag{10}$$

*Proof.* Work under Assumption E.2 and condition on $K = U\mathrm{diag}(\lambda_1, \ldots, \lambda_n)U^\top$. Let $\tilde{f}_\star := U^\top f_\star$, $\tilde{\varepsilon} := U^\top \varepsilon$, and $\tilde{r} := U^\top r$. Then, coordinate-wise for $i = 1, \ldots, n$,

$$\tilde{f}_{\star,i} \sim \mathcal{N}(0, \tau^2\lambda_i), \quad \tilde{\varepsilon}_i \sim \mathcal{N}(0, \sigma^2), \quad \tilde{r}_i = \tilde{f}_{\star,i} + \tilde{\varepsilon}_i,$$

and $(\tilde{f}_{\star,i}, \tilde{\varepsilon}_i)$ are independent across $i$.

For a spectral estimator $\hat{f}_g = g(K)r$, we have in the same eigenbasis

$$\widetilde{\hat{f}}_{g,i} := (U^\top \hat{f}_g)_i = g(\lambda_i)\tilde{r}_i.$$

Hence

$$\widetilde{\hat{f}}_{g,i} - \tilde{f}_{\star,i} = \big(g(\lambda_i) - 1\big)\tilde{f}_{\star,i} + g(\lambda_i)\tilde{\varepsilon}_i.$$

By independence and zero means,

$$\mathbb{E}\Big[(\widetilde{\hat{f}}_{g,i} - \tilde{f}_{\star,i})^2 \,\Big|\, K\Big] = \tau^2\lambda_i(1 - g(\lambda_i))^2 + \sigma^2 g(\lambda_i)^2.$$

Summing over $i$ yields (9).

To obtain the excess-risk identity, fix $\lambda \geq 0$ and define the scalar risk

$$r_\lambda(g) := \tau^2\lambda(1 - g)^2 + \sigma^2 g^2.$$

This is a strictly convex quadratic in $g$ with derivative

$$\frac{d}{dg}r_\lambda(g) = -2\tau^2\lambda(1 - g) + 2\sigma^2 g = 2\big((\tau^2\lambda + \sigma^2)g - \tau^2\lambda\big),$$

so the unique minimizer is

$$g_\star(\lambda) = \frac{\tau^2\lambda}{\tau^2\lambda + \sigma^2} = \frac{\lambda}{\lambda + \lambda_\star}, \quad \lambda_\star = \sigma^2/\tau^2.$$

Completing the square gives

$$r_\lambda(g) - r_\lambda(g_\star) = (\tau^2\lambda + \sigma^2)(g - g_\star)^2.$$

Applying this identity with $\lambda = \lambda_i$ and summing over $i$ yields (10). $\square$

Thm. E.4 shows that, conditional on $K(P_n)$, each prompt-only algorithm corresponds to a spectral shrinkage rule $g$, and its suboptimality is the weighted squared distance to the Bayes-optimal rule $g_\star$ on the spectrum of $K(P_n)$. We interpret prior mismatch as this filter-level distance.

We now compare the two procedures of interest within the prompt-level correction model:

- **Naive ICL (no update).** No correction is fit on the prompt: $\hat{f}_{\mathrm{ICL}} = 0$, equivalently $g_{\mathrm{ICL}}(\lambda) \equiv 0$.
- **$T$-step TTT.** TTT yields $\hat{f}_T = g_T(K)r$ with $g_T(\lambda) = 1 - (1 - \rho\lambda)^T$ (Eq. (6)).

**Corollary E.5** (Conditional Bayes gaps for ICL and TTT). *Under Assumption E.2 and conditioned on $K = U\mathrm{diag}(\lambda_i)U^\top$,*

$$\mathbb{E}\Big[\|\hat{f}_{\mathrm{ICL}} - f_{\mathrm{Bayes}}\|_2^2\Big|K\Big] = \sum_{i=1}^{n}(\tau^2\lambda_i + \sigma^2)g_\star(\lambda_i)^2, \text{ and}$$

$$\mathbb{E}\Big[\|\hat{f}_T - f_{\mathrm{Bayes}}\|_2^2\Big|K\Big] = \sum_{i=1}^{n}(\tau^2\lambda_i + \sigma^2)\big(g_T(\lambda_i) - g_\star(\lambda_i)\big)^2.$$

*In particular, within this benchmark, $T$-step TTT improves over naive ICL (in Bayes gap on prompt values) if and only if the weighted squared filter mismatch decreases:*

$$\sum_{i=1}^{n}(\tau^2\lambda_i + \sigma^2)\big(g_T(\lambda_i) - g_\star(\lambda_i)\big)^2 < \sum_{i=1}^{n}(\tau^2\lambda_i + \sigma^2)g_\star(\lambda_i)^2.$$

E.3.2. PROOF OF THM. 5.2

*Proof.* Condition on $(K, k_x, \kappa_x)$ and work under Assumption 5.1. By construction, $(f_\star, f_x)$ is jointly Gaussian with mean 0, and $\mathrm{Cov}\begin{pmatrix} f_\star \\ f_x \end{pmatrix} = \tau^2 \begin{pmatrix} K & k_x \\ k_x^\top & \kappa_x \end{pmatrix}$, $r = f_\star + \varepsilon$,

$\varepsilon \sim \mathcal{N}(0, \sigma^2 I_n)$, and $\varepsilon \perp (f_\star, f_x)$. Hence

$$\text{Cov}(r) = \tau^2 K + \sigma^2 I_n = \tau^2(K + \lambda_\star I_n),$$
$$\text{Cov}(f_x, r) = \tau^2 k_x^\top, \qquad \lambda_\star := \sigma^2/\tau^2.$$

Gaussian conditioning yields the Bayes (posterior-mean) estimator

$$\begin{aligned} f_{\text{Bayes}}(x) &= \mathbb{E}[f_x \mid r, K, k_x, \kappa_x] \\ &= \text{Cov}(f_x, r) \text{Cov}(r)^{-1} r \\ &= k_x^\top (K + \lambda_\star I_n)^{-1} r. \end{aligned}$$

Let $\hat{h}_q(x) = k_x^\top q(K) r$ be any kernel-linear estimator and define $\Delta q(K) := q(K) - q_\star(K)$ with $q_\star(\lambda) := (\lambda + \lambda_\star)^{-1}$. Then $\hat{h}_q(x) - f_{\text{Bayes}}(x) = k_x^\top \Delta q(K) r$, and conditioning on $(K, k_x, \kappa_x)$,

$$\begin{aligned} \mathbb{E}&\Big[ \{\hat{h}_q(x) - f_{\text{Bayes}}(x)\}^2 \mid K, k_x, \kappa_x \Big] \\ &= k_x^\top \Delta q(K) \text{Cov}(r) \Delta q(K) k_x \\ &= k_x^\top \Delta q(K) (\tau^2 K + \sigma^2 I_n) \Delta q(K) k_x. \end{aligned}$$

Diagonalize $K = U \text{diag}(\lambda_1, \ldots, \lambda_n) U^\top$ with columns $\{u_i\}_{i=1}^n$. Since $q(K)$ and $q_\star(K)$ are functions of $K$, they share eigenvectors with $K$, so expanding in this basis gives

$$\begin{aligned} \mathbb{E}&\Big[ \{\hat{h}_q(x) - f_{\text{Bayes}}(x)\}^2 \mid K, k_x, \kappa_x \Big] \\ &= \sum_{i=1}^n (\tau^2 \lambda_i + \sigma^2)(u_i^\top k_x)^2 \{q(\lambda_i) - q_\star(\lambda_i)\}^2, \end{aligned}$$

which matches Thm. 5.2 with $a_i(x) := (\tau^2 \lambda_i + \sigma^2)(u_i^\top k_x)^2$. $\qquad \square$

### E.4. Formal statements and proofs for Sec. 6

E.4.1. PRIOR-SHIFT SEPARATION FOR FIXED SPECTRAL BASELINES

Here, we use the term prior shift to denote test-time distribution shift that changes the task prior (equivalently, the effective SNR parameter $\lambda_\star = \sigma^2/\tau^2$) in the GP benchmark, thereby altering the Bayes-optimal shrinkage level.

**Proposition E.6** (Prior-shift separation on a projector kernel). *Work under Assumption 5.1 and focus on the prompt latent correction $f_\star \in \mathbb{R}^n$ observed through $r = f_\star + \varepsilon$, where (marginally) $f_\star \mid K \sim \mathcal{N}(0, \tau^2 K)$ and $\varepsilon \sim \mathcal{N}(0, \sigma^2 I_n)$ is independent. Assume $K \in \mathbb{R}^{n \times n}$ is an orthogonal projector of rank $m$ (so $K^2 = K$ and $\text{tr}(K) = m$), and recall the SNR parameter $\lambda = \sigma^2/\tau^2 > 0$. Then, the Bayes estimator of $f_\star$ given $(K, r)$ is*

$$f_{\text{Bayes}}^{(\lambda)} = \frac{1}{1 + \lambda} K r.$$

*Now fix any spectral estimator $\hat{f}_0 = g_0(K) r$ with $g_0(0) = 0$, and write $a := g_0(1)$ (so $\hat{f}_0 = aKr$). Assume $a$ is fixed (i.e., it does not depend on $\lambda$). Then for any two distinct prior parameters $\lambda_1 \neq \lambda_2$, letting*

$$s_j := \frac{1}{1 + \lambda_j}, \qquad \tau_j^2 := \frac{\sigma^2}{\lambda_j} \quad (j \in \{1, 2\}),$$

*one has the separation bound*

$$\begin{aligned} \max_{j \in \{1,2\}} &\mathbb{E}_{\lambda_j} \Big[ \| \hat{f}_0 - f_{\text{Bayes}}^{(\lambda_j)} \|_2^2 \,\Big|\, K \Big] \\ &\geq m \min_{j \in \{1,2\}} (\tau_j^2 + \sigma^2) \left( \frac{|s_1 - s_2|}{2} \right)^2, \end{aligned}$$

*where $\mathbb{E}_{\lambda_j}$ denotes expectation under the model with prior parameter $\lambda_j$.*

*Proof.* Under the stated marginal prompt model,

$$\text{Cov}(f_\star \mid K) = \tau^2 K, \qquad \text{Cov}(r \mid K) = \tau^2 K + \sigma^2 I_n.$$

By standard Gaussian conditioning for $r = f_\star + \varepsilon$,

$$\begin{aligned} f_{\text{Bayes}}^{(\lambda)} &= \mathbb{E}[f_\star \mid r, K] \\ &= \tau^2 K (\tau^2 K + \sigma^2 I_n)^{-1} r \\ &= K(K + \lambda I_n)^{-1} r, \end{aligned}$$

where $\lambda = \sigma^2/\tau^2$. Since $K$ is a projector, it has only eigenvalues 0 and 1, hence $K(K + \lambda I_n)^{-1} = \frac{1}{1+\lambda} K$, which yields $f_{\text{Bayes}}^{(\lambda)} = \frac{1}{1+\lambda} Kr$.

Diagonalize $K = U \text{diag}(I_m, 0) U^\top$. Because $g_0(0) = 0$, the functional calculus gives

$$g_0(K) = U \text{diag}(g_0(1) I_m, 0) U^\top = aK, \quad a := g_0(1),$$

so $\hat{f}_0 = g_0(K) r = aKr$.

Fix $j \in \{1, 2\}$. Under prior $\lambda_j$ (equivalently $\tau_j^2 = \sigma^2/\lambda_j$), we have $f_{\text{Bayes}}^{(\lambda_j)} = s_j Kr$, and $s_j := \frac{1}{1+\lambda_j}$. Therefore, $\hat{f}_0 - f_{\text{Bayes}}^{(\lambda_j)} = (a - s_j) Kr \Rightarrow \| \hat{f}_0 - f_{\text{Bayes}}^{(\lambda_j)} \|_2^2 = (a - s_j)^2 \| Kr \|_2^2$. Taking conditional expectation given $K$ (under prior $j$),

$$\mathbb{E}_{\lambda_j} \Big[ \| \hat{f}_0 - f_{\text{Bayes}}^{(\lambda_j)} \|_2^2 \,\Big|\, K \Big] = (a - s_j)^2 \mathbb{E}_{\lambda_j}[\| Kr \|_2^2 \mid K].$$

Since $\mathbb{E}_{\lambda_j}[r \mid K] = 0$ and $\text{Cov}_{\lambda_j}(r \mid K) = \tau_j^2 K + \sigma^2 I_n$,

$$\begin{aligned} \mathbb{E}_{\lambda_j}[\| Kr \|_2^2 \mid K] &= \text{tr}\big(\text{Cov}_{\lambda_j}(Kr \mid K)\big) \\ &= \text{tr}\big(K(\tau_j^2 K + \sigma^2 I_n) K\big) \\ &= (\tau_j^2 + \sigma^2) \text{tr}(K) \\ &= m(\tau_j^2 + \sigma^2). \end{aligned}$$

Hence, for each $j \in \{1, 2\}$,

$$\mathbb{E}_{\lambda_j}\left[\|\hat{f}_0 - f_{\text{Bayes}}^{(\lambda_j)}\|_2^2 \,\Big|\, K\right] = m(\tau_j^2 + \sigma^2)(a - s_j)^2. \quad (11)$$

Because $s_1 \neq s_2$, for any scalar $a$ at least one of the distances $|a - s_1|$ or $|a - s_2|$ is at least $\frac{1}{2}|s_1 - s_2|$ (e.g., by considering the midpoint $(s_1 + s_2)/2$). Thus

$$\max_{j \in \{1,2\}} (a - s_j)^2 \geq \left(\frac{|s_1 - s_2|}{2}\right)^2.$$

Combine this with (11) and lower bound $(\tau_j^2 + \sigma^2)$ by $\min_{j \in \{1,2\}}(\tau_j^2 + \sigma^2)$ to obtain the stated separation bound. The final claim follows because a single fixed $a$ cannot equal both $s_1$ and $s_2$. $\quad\square$

### E.4.2. FULL LEAST-SQUARES FITTING CAN BE DOMINATED

The following proposition is a corollary of the Stein phenomenon (James & Stein, 1961; Shao & Strawderman, 1994).

**Proposition E.7** (Inadmissibility of projection on a $p$-dimensional subspace). *Let* $r = f_\star + \varepsilon$ *with* $\varepsilon \sim \mathcal{N}(0, \sigma^2 I_n)$ *(known $\sigma^2$), and assume* $f_\star \in \mathcal{S}$ *where* $\mathcal{S} \subseteq \mathbb{R}^n$ *is a fixed subspace of dimension* $d_{\text{st}} := \dim(\mathcal{S}) \geq 3$. *Let* $\Pi_\mathcal{S}$ *be the orthogonal projector onto* $\mathcal{S}$. *Then the full-fit estimator* $\hat{f}_\infty := \Pi_\mathcal{S} r$ *is inadmissible under squared loss* $\|\hat{f} - f_\star\|_2^2$; *namely, it is dominated by the (subspace) James–Stein shrinkage estimator*

$$\hat{f}_{\text{JS}} = \begin{cases} 0, & \|\Pi_\mathcal{S} r\|_2 = 0, \\ \left(1 - \dfrac{(d_{\text{st}} - 2)\sigma^2}{\|\Pi_\mathcal{S} r\|_2^2}\right) \Pi_\mathcal{S} r, & \text{otherwise.} \end{cases}$$

*Equivalently, for all* $f_\star \in \mathcal{S}$,

$$\mathbb{E}_{f_\star}\|\hat{f}_{\text{JS}} - f_\star\|_2^2 \leq \mathbb{E}_{f_\star}\|\hat{f}_\infty - f_\star\|_2^2,$$

*with strict inequality for some* $f_\star \in \mathcal{S}$.

*Remark E.8.* Although Prop. E.7 is stated under $f_\star \in \mathcal{S}$, the conclusion can be read as a statement about improving the estimation of the in-subspace component $\Pi_\mathcal{S} f_\star$ from $\Pi_\mathcal{S} r$. For general $f_\star \in \mathbb{R}^n$, the orthogonal component contributes a fixed additive error by the Pythagorean decomposition (Prop. E.9).

### E.4.3. ADAPTING UPDATE STEPS CANNOT OVERCOME SUBSPACE MISMATCH

Let $K \succeq 0$ be symmetric and define the representable subspace $\mathcal{S} := \text{range}(K) \subseteq \mathbb{R}^n$. Let $\Pi_\mathcal{S}$ and $\Pi_{\mathcal{S}^\perp}$ be Euclidean orthogonal projectors.

**Proposition E.9** (Irreducible error outside $\text{range}(K)$). *For any target vector* $f_\star \in \mathbb{R}^n$ *and any estimate* $\hat{f} \in \mathcal{S}$,

$$\|\hat{f} - f_\star\|_2^2 = \|\hat{f} - \Pi_\mathcal{S} f_\star\|_2^2 + \|\Pi_{\mathcal{S}^\perp} f_\star\|_2^2.$$

*In particular,* $\|\hat{f} - f_\star\|_2^2 \geq \|\Pi_{\mathcal{S}^\perp} f_\star\|_2^2$ *and* $\inf_{\hat{f} \in \mathcal{S}} \|\hat{f} - f_\star\|_2^2 = \|\Pi_{\mathcal{S}^\perp} f_\star\|_2^2$.

*Moreover, any spectral estimator* $\hat{f}_g = g(K)r$ *with* $g(0) = 0$ *satisfies* $\hat{f}_g \in \mathcal{S}$. *In particular, TTT for any $T$ lies in $\mathcal{S}$.*

*Proof.* Decompose $f_\star = \Pi_\mathcal{S} f_\star + \Pi_{\mathcal{S}^\perp} f_\star$. If $\hat{f} \in \mathcal{S}$, then $\hat{f} - \Pi_\mathcal{S} f_\star \in \mathcal{S}$, while $\Pi_{\mathcal{S}^\perp} f_\star \in \mathcal{S}^\perp$, so the cross term vanishes:

$$\|\hat{f} - f_\star\|_2^2 = \|\hat{f} - \Pi_\mathcal{S} f_\star\|_2^2 + \|\Pi_{\mathcal{S}^\perp} f_\star\|_2^2.$$

The lower bound and the infimum follow immediately.

For the final claim, diagonalize $K = U \text{diag}(\lambda_1, \ldots, \lambda_n) U^\top$. If $g(0) = 0$, then $g(K) = U \text{diag}(g(\lambda_i)) U^\top$ annihilates $\ker(K)$ and therefore maps $\mathbb{R}^n$ into $\text{range}(K) = \mathcal{S}$. Hence $\hat{f}_g = g(K)r \in \mathcal{S}$. $\quad\square$

### E.5. PAC-Bayes details for Sec. 7

This appendix proves the normalized PAC-Bayes generalization bound stated in Sec. 7. Throughout, we condition on the prompt-induced geometry $K \succeq 0$ and treat all matrices as deterministic given $K$.

*Remark* E.10 (Gibbs posteriors as Radon–Nikodym derivatives; measurability and conditional PAC-Bayes). We use a Gibbs construction to select (or average over) the update horizon $T$. This can be formulated cleanly in measure-theoretic terms, which clarifies measurability and why our PAC-Bayes guarantees are naturally stated *conditionally on the prompt geometry $K$*.

**RN definition (general form).** Let $(\mathcal{H}, \mathcal{A})$ be a hypothesis space (here $\mathcal{H} = \mathcal{T}$) and let $\pi_0$ be a prior probability measure on $\mathcal{H}$. Given data $D$ (here $D = (r, K)$ or just $r$ when $K$ is fixed) and a measurable empirical score/loss $\ell : \mathcal{H} \times \mathsf{D} \to \mathbb{R}$, define the normalizer

$$Z_\beta(D) := \int_\mathcal{H} \exp\{-\beta n\, \ell(h, D)\}\, \pi_0(dh),$$

assumed finite and strictly positive. The Gibbs posterior is the probability measure $\nu_\beta(\cdot \mid D)$ on $\mathcal{H}$ defined by the Radon–Nikodym derivative

$$\frac{d\nu_\beta(\cdot \mid D)}{d\pi_0}(h) = \frac{\exp\{-\beta n\, \ell(h, D)\}}{Z_\beta(D)}.$$

In our setting with finite $\mathcal{T}$, this reduces to the familiar normalized weights $\nu_\beta(T \mid r) \propto \pi_0(T) \exp\{-\beta n \ell_T(r)\}$.

**Measurability (posterior as a kernel).** If $\ell(h, D)$ is measurable in $(h, D)$, then $Z_\beta(D)$ is measurable in $D$ and the map $D \mapsto \nu_\beta(\cdot \mid D)$ is a measurable probability kernel from the data space to $(\mathcal{H}, \mathcal{A})$. Consequently, posterior expectations (model averaging) and measurable selectors such as an argmin/argmax (e.g., $T_{\mathrm{MAP}}$ with a deterministic tie-breaking rule) are measurable functions of the observed data.

**Why the PAC-Bayes statement is naturally conditional on $K$.** In Sec. 7 and App. E.5, we condition on the prompt-induced geometry $K$ and analyze the randomness of $r \mid K$. This is not merely a convenience: the Gibbs posterior $\nu_\beta(\cdot \mid r, K)$ is defined as an RN tilt of $\pi_0$ given the realized data, so all PAC-Bayes change-of-measure steps (e.g., Donsker–Varadhan / exponential-moment arguments) can be applied *pathwise in $K$* to the conditional law of $r \mid K$. The resulting bounds therefore hold with high probability over $r \mid K$ for each fixed $K$. If desired, one can then integrate over the distribution of $K$ to obtain unconditional statements.

**Nullspace contributions.** If $K$ is singular, then on $\mathrm{null}(K)$ we have $\Sigma_T = \sigma^2 I$ for all $T$, so both $\log \det(\Sigma_T)$ and $r^\top \Sigma_T^{-1} r$ contribute only $T$-independent constants from $\mathrm{null}(K)$. Equivalently, after ignoring $T$-independent constants, all determinants/inverses may be interpreted on $\mathrm{range}(K)$.

Let $\mathcal{T}$ be a finite set of candidates of $T$ and let $\pi_0$ be a prior on $\mathcal{T}$ with full support. Write $\Delta(\mathcal{T})$ for the simplex of distributions over $\mathcal{T}$.

**Lemma E.11** (PAC-Bayes with exponential-moment). *Let $\{X_T(r)\}_{T \in \mathcal{T}}$ be real-valued random variables and fix $\beta > 0$. Assume there exist deterministic (given $K$) numbers $\psi_T(\beta)$ such that*

$$\mathbb{E}\big[\exp\big(\beta X_T(r)\big) \mid K\big] \leq \exp\{\psi_T(\beta)\} \qquad \forall T \in \mathcal{T}.$$

*Then for any $\delta \in (0, 1)$, with probability at least $1 - \delta$ over $r \mid K$, simultaneously for all $\nu \in \Delta(\mathcal{T})$,*

$$\mathbb{E}_{T \sim \nu}[X_T(r)] \leq \frac{1}{\beta}\Big\{\mathrm{KL}(\nu\|\pi_0) + \log\tfrac{1}{\delta}\Big\} + \frac{1}{\beta}\mathbb{E}_{T \sim \nu}[\psi_T(\beta)].$$

This is a standard PAC-Bayes result based on Donsker-Varadhan's variational formula. See Lem. 2.2 and the proof of Thm. 2.1 in Alquier et al. (2024).

**Lemma E.12** (Exact MGF and a trace bound). *Under Assumption 7.1, for every $T \in \mathcal{T}$ and $\beta > 0$,*

$$\mathbb{E}\left[\exp\left(\frac{\beta}{2}\big(\mathrm{tr}(A_T) - Z^\top A_T Z\big)\right)\right]$$
$$= \exp\left\{\frac{\beta}{2}\mathrm{tr}(A_T)\right\} \det(I + \beta A_T)^{-1/2}, \quad Z \sim \mathcal{N}(0, I_n).$$
$$(12)$$

*Moreover,*

$$\log \mathbb{E}\left[\exp\left(\frac{\beta}{2}\big(\mathrm{tr}(A_T) - Z^\top A_T Z\big)\right)\right] \leq \frac{\beta^2}{4}\mathrm{tr}(A_T^2).$$
$$(13)$$

*Proof.* Diagonalize $A_T = U\mathrm{diag}(a_i)U^\top$ with $a_i \geq 0$ and set $\widetilde{Z} = U^\top Z \sim \mathcal{N}(0, I)$. Then $Z^\top A_T Z = \sum_i a_i \widetilde{Z}_i^2$ and $\mathbb{E}[\exp(-\frac{\beta}{2}Z^\top A_T Z)] = \prod_i (1 + \beta a_i)^{-1/2} = \det(I + \beta A_T)^{-1/2}$, which gives (12). For (13), use $\log(1 + u) \geq u - u^2/2$ for $u \geq 0$ to obtain $\log \det(I + \beta A_T) \geq \beta\mathrm{tr}(A_T) - \frac{\beta^2}{2}\mathrm{tr}(A_T^2)$, and substitute into log of (12). $\square$

### E.5.1. PROOF OF THE NORMALIZED PAC-BAYES BOUND

**Lemma E.13** (PAC-Bayes bound for update steps). *Grant Assumption 7.1 and let $\mathcal{T}$ be finite. Fix a prior $\pi_0$ on $\mathcal{T}$ with full support and $\beta > 0$. Then for any $\delta \in (0, 1)$, with probability at least $1 - \delta$ over $r \mid K$, simultaneously for all $\nu \in \Delta(\mathcal{T})$,*

$$\mathbb{E}_{T \sim \nu}\big[\mathcal{L}(T \mid K)\big]$$
$$\leq \mathbb{E}_{T \sim \nu}\big[\ell_T(r)\big] + \frac{\mathrm{KL}(\nu\|\pi_0) + \log(1/\delta)}{\beta n} + \frac{\beta}{4}\bar{v}.$$
$$(14)$$

*Proof.* Write $r = \Sigma_\star^{1/2} Z$ with $Z \sim \mathcal{N}(0, I_n)$. For fixed $T$, the log-determinant term $\log \det(\Sigma_T)$ is deterministic, hence

$$\mathcal{L}(T \mid K) - \ell_T(r) = \frac{1}{2n}\Big\{\mathrm{tr}(\Sigma_T^{-1}\Sigma_\star) - r^\top \Sigma_T^{-1} r\Big\}$$
$$= \frac{1}{2n}\Big\{\mathrm{tr}(A_T) - Z^\top A_T Z\Big\}.$$

Define

$$X_T(r) := n\big(\mathcal{L}(T \mid K) - \ell_T(r)\big) = \frac{1}{2}\big(\mathrm{tr}(A_T) - Z^\top A_T Z\big).$$

By Lem. E.12,

$$\log \mathbb{E}[\exp(\beta X_T(r)) \mid K] \leq \frac{\beta^2}{4}\mathrm{tr}(A_T^2)$$
$$\Rightarrow \mathbb{E}[\exp(\beta X_T(r)) \mid K] \leq \exp\left\{\frac{\beta^2}{4}\mathrm{tr}(A_T^2)\right\}.$$

Apply Lem. E.11 with certificate $\psi_T(\beta) = \frac{\beta^2}{4}\mathrm{tr}(A_T^2)$ to obtain, with probability $\geq 1 - \delta$,

$$\mathbb{E}_{T \sim \nu}[X_T(r)]$$
$$\leq \frac{\mathrm{KL}(\nu\|\pi_0) + \log(1/\delta)}{\beta} + \frac{1}{\beta}\mathbb{E}_{T \sim \nu}\left[\frac{\beta^2}{4}\mathrm{tr}(A_T^2)\right].$$

Divide by $n$ and use $v_T = \mathrm{tr}(A_T^2)/n$ to conclude (14). $\square$

### E.5.2. OPTIMIZING THE TEMPERATURE $\beta$ (RATE FORM)

**Corollary E.14** (Rate form via $\beta$-optimization). *On the event of Lem. E.13, for any $\nu \in \Delta(\mathcal{T})$,*

$$\mathbb{E}_{T\sim\nu}\big[\mathcal{L}(T \mid K)\big]$$
$$\leq \mathbb{E}_{T\sim\nu}\big[\ell_T(r)\big] + \sqrt{\frac{\bar{v}\big(\mathrm{KL}(\nu\|\pi_0) + \log(1/\delta)\big)}{n}}. \quad (15)$$

*Proof.* Starting from the last bound in (14), set $A := \mathrm{KL}(\nu\|\pi_0) + \log(1/\delta)$ and minimize $f(\beta) = A/(\beta n) + (\beta/4)\bar{v}$ over $\beta > 0$. The minimizer is $\beta^\star = 2\sqrt{A/(n\bar{v})}$, yielding $f(\beta^\star) = \sqrt{\bar{v} A/n}$, which proves (15). $\quad\square$

If $\pi_0$ is uniform on $\mathcal{T}$ and $\nu = \delta_T$, then $\mathrm{KL}(\delta_T\|\pi_0) = \log|\mathcal{T}|$ and Cor. E.14 gives

$$\mathcal{L}(T \mid K) \leq \ell_T(r) + \sqrt{\frac{\bar{v}\big(\log|\mathcal{T}| + \log(1/\delta)\big)}{n}}.$$

### E.5.3. FROM EVIDENCE MINIMIZATION TO VANISHING NORMALIZED BAYES GAP

This subsection formalizes how evidence-based $T$ selection controls (and, in well-specified regimes, drives to zero) the normalized Bayes gap under the GP benchmark of Assumption E.2.

We adopt the notation from Sec. 7 under the GP benchmark (Assumption E.2). Let $\Sigma_\star := \tau^2 K + \sigma^2 I_n$ be the marginal residual covariance under the prior. We define the oracle baseline risk $\mathcal{L}_{\mathrm{true}}(K)$ and the normalized Bayes gap $\mathrm{Gap}_n(T \mid K)$ as follows:

$$\mathcal{L}_{\mathrm{true}}(K) := \frac{1}{2n}\Big\{\log\det(\Sigma_\star) + n\Big\},$$
$$\mathrm{Gap}_n(T \mid K) := \frac{1}{n}\,\mathbb{E}\Big[\|\hat{f}_T - f_{\mathrm{Bayes}}\|_2^2 \,\Big|\, K\Big],$$

where $\hat{f}_T$ is the $T$-step TTT estimator and $f_{\mathrm{Bayes}}$ is the Bayes posterior mean.

### E.5.4. A TRANSFER INEQUALITY: KL CONTROLS THE BAYES GAP

**Lemma E.15** (A scalar quadratic lower bound). *For every $t > 0$,*

$$t - 1 - \log t \geq \frac{(t-1)^2}{2\max\{t,1\}}. \quad (16)$$

*Proof.* If $0 < t \leq 1$, define $h(t) := t - 1 - \log t - \frac{(t-1)^2}{2}$. Then $h(1) = h'(1) = 0$ and $h''(t) = t^{-2} - 1 \geq 0$, so $h(t) \geq 0$ on $(0,1]$. If $t \geq 1$, define $h(t) := t - 1 - \log t - \frac{(t-1)^2}{2t}$. Then $h(1) = 0$ and

$$h'(t) = 1 - \frac{1}{t} - \frac{1}{2}\Big(1 - \frac{1}{t^2}\Big) = \frac{(t-1)^2}{2t^2} \geq 0,$$

so $h(t) \geq 0$ for $t \geq 1$. This yields (16). $\quad\square$

**Lemma E.16** (Evidence-to-Bayes-gap transfer). *Under Assumption E.2 and conditioned on $K$, for every $T \in \mathbb{N}_0$,*

$$\mathbb{E}\Big[\|\hat{f}_T - f_{\mathrm{Bayes}}\|_2^2 \,\Big|\, K\Big] \leq 4\sigma^2\,\mathrm{KL}(\mathcal{N}(0,\Sigma_\star) \,\|\, \mathcal{N}(0,\Sigma_T)). \quad (17)$$

*Moreover,*

$$\mathrm{KL}(\mathcal{N}(0,\Sigma_\star) \,\|\, \mathcal{N}(0,\Sigma_T)) = n\Big(\mathcal{L}(T \mid K) - \mathcal{L}_{\mathrm{true}}(K)\Big). \quad (18)$$

*Proof.* Condition on $K = U\mathrm{diag}(\lambda_i)U^\top$. In the $K$-eigenbasis, the Bayes shrinkage is $g_\star(\lambda) = \lambda/(\lambda + \lambda_\star)$ and the $T$-step GD shrinkage is $g_T(\lambda) = 1 - (1 - \rho\lambda)^T$. Since $r \mid K \sim \mathcal{N}(0,\Sigma_\star)$ with $\Sigma_\star = \tau^2 K + \sigma^2 I$, the mode-wise Bayes gap is

$$\mathbb{E}\Big[\|\hat{f}_T - f_{\mathrm{Bayes}}\|_2^2 \,\Big|\, K\Big] = \sum_{i=1}^n (\tau^2\lambda_i+\sigma^2)\big(g_T(\lambda_i)-g_\star(\lambda_i)\big)^2.$$

Define the dimensionless eigenvalues $m_{\star,i} := \frac{\tau^2\lambda_i+\sigma^2}{\sigma^2} = 1 + \frac{\lambda_i}{\lambda_\star} \geq 1$, $m_{T,i} := \frac{\mathrm{eig}(\Sigma_T)_i}{\sigma^2} = (1 - \rho\lambda_i)^{-T} \geq 1$, and $r_i := \frac{m_{\star,i}}{m_{T,i}} > 0$. Then $g_\star(\lambda_i) = 1 - 1/m_{\star,i}$ and $g_T(\lambda_i) = 1 - 1/m_{T,i}$, hence $g_T(\lambda_i) - g_\star(\lambda_i) = m_{\star,i}^{-1} - m_{T,i}^{-1} = (r_i - 1)/m_{\star,i}$ and therefore

$$(\tau^2\lambda_i + \sigma^2)\big(g_T(\lambda_i) - g_\star(\lambda_i)\big)^2 = \sigma^2\frac{(r_i - 1)^2}{m_{\star,i}}.$$

Since $m_{T,i} \geq 1$, one has $r_i = m_{\star,i}/m_{T,i} \leq m_{\star,i}$, and also $m_{\star,i} \geq 1$, so $m_{\star,i} \geq \max\{r_i, 1\}$. Hence

$$(\tau^2\lambda_i + \sigma^2)\big(g_T(\lambda_i) - g_\star(\lambda_i)\big)^2 \leq \sigma^2\frac{(r_i - 1)^2}{\max\{r_i, 1\}}. \quad (19)$$

On the other hand, since $\Sigma_\star$ and $\Sigma_T$ commute (both are maps of $K$), the Gaussian KL decomposes mode-wise:

$$\mathrm{KL}(\mathcal{N}(0,\Sigma_\star) \,\|\, \mathcal{N}(0,\Sigma_T)) = \frac{1}{2}\sum_{i=1}^n\Big(r_i - 1 - \log r_i\Big).$$

Applying Lem. E.15 to each $r_i$ yields $\frac{(r_i-1)^2}{\max\{r_i,1\}} \leq 2(r_i - 1 - \log r_i)$, and combining with (19) gives

$$(\tau^2\lambda_i + \sigma^2)\big(g_T(\lambda_i) - g_\star(\lambda_i)\big)^2 \leq 2\sigma^2(r_i - 1 - \log r_i)$$
$$= 4\sigma^2\frac{1}{2}(r_i - 1 - \log r_i).$$

Summing over $i$ proves (17).

Finally, for zero-mean Gaussians,

$$\mathrm{KL}(\mathcal{N}(0,\Sigma_\star) \,\|\, \mathcal{N}(0,\Sigma_T))$$
$$= \frac{1}{2}\Big\{\log\det(\Sigma_T) - \log\det(\Sigma_\star) + \mathrm{tr}(\Sigma_T^{-1}\Sigma_\star) - n\Big\}.$$

Using the definitions of $\mathcal{L}(T \mid K)$ and $\mathcal{L}_{\mathrm{true}}(K)$ yields (18). $\quad\square$

### E.5.5. A PAC-BAYES HIGH-PROBABILITY BOUND FOR THE EB/MAP $T$

Let $\mathcal{T} \subset \mathbb{N}_0$ be a finite candidate set and define the EB/MAP $T$

$$\widehat{T} \in \arg\min_{T \in \mathcal{T}} \ell_T(r). \tag{20}$$

**Lemma E.17** (High-probability Bayes-gap bound for evidence minimization)**.** *Let $\mathcal{T}$ be finite. For any $\delta \in (0,1)$, with probability at least $1 - \delta$ over $r \mid K$, $\mathrm{Gap}_n(\widehat{T} \mid K) \leq 4\sigma^2\Big(\min_{T \in \mathcal{T}} \ell_T(r) - \mathcal{L}_{\mathrm{true}}(K) + \sqrt{\frac{\bar{v}(\log|\mathcal{T}| + \log(1/\delta))}{n}}\Big).$*

*Proof.* On the $1 - \delta$ event of Thm. 7.2 (point selection, uniform prior), $\mathcal{L}(T \mid K) \leq \ell_T(r) + \sqrt{\bar{v}(\log|\mathcal{T}| + \log(1/\delta))/n}$ holds for all $T \in \mathcal{T}$. Using (18) yields, for all $T \in \mathcal{T}$ on the same event, $\mathrm{KL}(\mathcal{N}(0, \Sigma_\star) \| \mathcal{N}(0, \Sigma_T)) \leq n\Big(\ell_T(r) - \mathcal{L}_{\mathrm{true}}(K) + \sqrt{\frac{\bar{v}(\log|\mathcal{T}| + \log(1/\delta))}{n}}\Big).$ Apply Thm. E.16 and divide by $n$, then use $\ell_{\widehat{T}}(r) = \min_{T \in \mathcal{T}} \ell_T(r)$ by definition of $\widehat{T}$. $\square$

### E.5.6. ASYMPTOTIC CONSEQUENCE: VANISHING NORMALIZED BAYES GAP

**Lemma E.18** (One-sided concentration of $\ell_T$ around $\mathcal{L}(T \mid K)$)**.** *Fix $T \in \mathbb{N}_0$. Let $A_T = \Sigma_\star^{1/2} \Sigma_T^{-1} \Sigma_\star^{1/2}$ and $v_T = \frac{1}{n}\mathrm{tr}(A_T^2)$. Then for any $\delta \in (0,1)$, with probability at least $1 - \delta$ over $r \mid K$,*

$$\ell_T(r) \leq \mathcal{L}(T \mid K) + \sqrt{\frac{v_T \log(1/\delta)}{n}} + \frac{\|A_T\|_{\mathrm{op}} \log(1/\delta)}{n}. \tag{21}$$

*Proof.* Write $r = \Sigma_\star^{1/2} Z$ with $Z \sim \mathcal{N}(0, I_n)$. Then

$$\ell_T(r) - \mathcal{L}(T \mid K) = \frac{1}{2n}\Big(Z^\top A_T Z - \mathrm{tr}(A_T)\Big).$$

Diagonalize $A_T = U \mathrm{diag}(a_i) U^\top$ with $a_i \geq 0$ and $\widetilde{Z} = U^\top Z \sim \mathcal{N}(0, I)$. Then $Z^\top A_T Z = \sum_i a_i \widetilde{Z}_i^2$. A standard Chernoff bound for Gaussian quadratic forms (Vershynin, 2018) gives: for any $t > 0$, $\mathbb{P}\Big(Z^\top A_T Z - \mathrm{tr}(A_T) \geq 2\sqrt{\mathrm{tr}(A_T^2) t} + 2\|A_T\|_{\mathrm{op}} t\Big) \leq e^{-t}$. Setting $t = \log(1/\delta)$ and dividing by $2n$ yields (21) using $\mathrm{tr}(A_T^2) = nv_T$. $\square$

**Theorem E.19** (Vanishing normalized Bayes gap)**.** *Let $\{\mathcal{T}_n\}_{n \geq 1}$ be a sequence of finite candidate sets and let $\widehat{T}_n$ be the EB selector (20). Assume:*

*(A1)* *(Approximation / well-specification.) There exists a (possibly $n$-dependent) comparator $T_n^\star \in \mathcal{T}_n$ such that*

$$\alpha_n := \frac{1}{n} \mathrm{KL}\big(\mathcal{N}(0, \Sigma_\star) \| \mathcal{N}(0, \Sigma_{T_n^\star})\big) \to 0.$$

*(In particular, $\alpha_n \equiv 0$ in the well-specified case $\Sigma_\star = \Sigma_{T_n^\star}$.)*

*(A2)* *(Complexity.) $\bar{v}_n := \sup_{T \in \mathcal{T}_n} v_T = O(1)$ and $\log|\mathcal{T}_n| = o(n)$.*

*(A3)* *(Boundedness.) $\|A_{T_n^\star}\|_{\mathrm{op}} = O(1)$.*

*Then, for any fixed $\delta \in (0,1)$, with probability at least $1 - 2\delta$ (over $r \mid K$),*

$$\mathrm{Gap}_n(\widehat{T}_n \mid K)$$
$$\leq 4\sigma^2\Big(\alpha_n + \sqrt{\frac{\bar{v}_n(\log|\mathcal{T}_n| + \log(1/\delta))}{n}}$$
$$+ \sqrt{\frac{v_{T_n^\star} \log(1/\delta)}{n}} + \frac{\|A_{T_n^\star}\|_{\mathrm{op}} \log(1/\delta)}{n}\Big), \quad (22)$$

*and in particular $\mathrm{Gap}_n(\widehat{T}_n \mid K) \to 0$ in probability (and along the above high-probability event).*

*Proof.* Apply Lem. E.17 with confidence $\delta$ to get, with probability $\geq 1 - \delta$, $\mathrm{Gap}_n(\widehat{T}_n \mid K) \leq 4\sigma^2\Big(\ell_{T_n^\star}(r) - \mathcal{L}_{\mathrm{true}}(K) + \sqrt{\frac{\bar{v}_n(\log|\mathcal{T}_n| + \log(1/\delta))}{n}}\Big)$, since $\ell_{\widehat{T}_n}(r) \leq \ell_{T_n^\star}(r)$. Next, apply Lem. E.18 to $T = T_n^\star$ with confidence $\delta$ to obtain, with probability $\geq 1 - \delta$,

$$\ell_{T_n^\star}(r) \leq \mathcal{L}(T_n^\star \mid K) + \sqrt{\frac{v_{T_n^\star} \log(1/\delta)}{n}} + \frac{\|A_{T_n^\star}\|_{\mathrm{op}} \log(1/\delta)}{n}.$$

Intersect the two events (probability $\geq 1 - 2\delta$) and combine with $\mathcal{L}(T_n^\star \mid K) - \mathcal{L}_{\mathrm{true}}(K) = \alpha_n$ by (18). This yields (22). The stated convergence follows from assumptions (i)–(iii). $\square$

---

**Summary: Evidence tuning does more than "avoid overfit"**

- **KL controls Bayes gap:** evidence closeness $\Rightarrow$ prediction gap shrinks (App. E.5.3).
- **EB/MAP comes with guarantees:** it yields high-probability Bayes-gap bounds, not just heuristics.

---

### E.6. Bayes-optimal update subspaces: proofs and extensions

This appendix provides proofs and extensions for Sec. 8. We work under the linear-Gaussian correction model (Assumption 8.1).

Note that Assumption 8.1 implies Assumption 5.1.

**Lemma E.20** (Assumption 8.1 induces the Gaussian benchmark)**.** *Under Assumption 8.1, condition on $(J_w, j_x)$ and define $f_\star := J_w u_\star \in \mathbb{R}^n$ and $f_x := j_x^\top u_\star \in \mathbb{R}$. Then $(f_\star, f_x)$ is jointly Gaussian with $\mathrm{Cov}(f_\star) = J_w \Sigma J_w^\top, \mathrm{Cov}(f_\star, f_x) = J_w \Sigma j_x$, and $\mathrm{Var}(f_x) = j_x^\top \Sigma j_x$.*

*In particular, taking $\Sigma = \Pi_{\mathcal{U}}$ yields the benchmark in Assumption 5.1 with $K = J_w \Pi_{\mathcal{U}} J_w^\top$ and $k_x = J_w \Pi_{\mathcal{U}} j_x$ (and $\kappa_x = j_x^\top \Pi_{\mathcal{U}} j_x$).*

**Remark on singular $Q$ (whitened coordinates).** When the query metric $Q \succeq 0$ is singular (e.g., $Q = j_x j_x^\top$ in the QUERY-AWARE specialization), we interpret the whitening map $v := Q^{1/2} u$ as mapping into $\mathrm{range}(Q)$ and restrict all $v$-space objects (including any projector $\widetilde{\Pi}$) to $\mathrm{range}(Q)$ (equivalently, quotient out $\mathrm{null}(Q)$). When mapping back to $u$, we understand $Q^{-1/2}$ as the Moore–Penrose pseudoinverse on $\mathrm{range}(Q)$ (cf. the implementation note in Sec. 8).

**Lemma E.21** (Posterior mean and covariance). *Under Assumption 8.1, the posterior is Gaussian $u_\star \mid r \sim \mathcal{N}(\bar{u}, \Sigma_{\mathrm{post}})$ with $\bar{u} = \mathbb{E}[u_\star \mid r] = \Sigma J_w^\top (J_w \Sigma J_w^\top + \sigma^2 I_n)^{-1} r$, and $\Sigma_{\mathrm{post}} = \Sigma - \Sigma J_w^\top (J_w \Sigma J_w^\top + \sigma^2 I_n)^{-1} J_w \Sigma$.*

*Proof.* This is the standard Gaussian conditioning formula for linear observations (see Bishop, 2006, §2.3). $\qquad\square$

**Lemma E.22** (Constrained Bayes estimator equals projection of the posterior mean). *Fix an orthogonal projector $\Pi$ of rank $d$ in $\mathbb{R}^p$. Among all measurable estimators $\widehat{u} = \widehat{u}(r)$ satisfying $\widehat{u}(r) \in \mathrm{range}(\Pi)$ almost surely, the unique minimizer of $\mathbb{E}[\|\widehat{u} - u_\star\|_2^2 \mid r]$ is*

$$\widehat{u}_\Pi(r) = \Pi\, \bar{u}(r) = \Pi\, \mathbb{E}[u_\star \mid r].$$

*Remark.* Lemma E.22 is purely Euclidean and applies verbatim to the whitened coordinates $v = Q^{1/2} u$: replacing $(u_\star, \bar{u}, \Pi)$ with $(v_\star, \bar{v}, \widetilde{\Pi})$ yields the constrained estimator $\widehat{v}_{\widetilde{\Pi}}(r) = \widetilde{\Pi}\, \bar{v}(r)$.

*Proof.* Condition on $r$ and decompose $u_\star = \Pi u_\star + (I - \Pi) u_\star$ into orthogonal components. For any $\widehat{u} \in \mathrm{range}(\Pi)$, $\|\widehat{u} - u_\star\|_2^2 = \|\widehat{u} - \Pi u_\star\|_2^2 + \|(I - \Pi) u_\star\|_2^2$. The second term is independent of $\widehat{u}$, so the conditional risk is minimized by $\widehat{u} = \mathbb{E}[\Pi u_\star \mid r] = \Pi \bar{u}$. Uniqueness follows from strict convexity. $\qquad\square$

### E.6.1. PROOF OF THM. 8.2

*Proof.* Work in the $Q$-whitened coordinates $v := Q^{1/2} u$ (restricted to $\mathrm{range}(Q)$ if $Q$ is singular). Define $v_\star := Q^{1/2} u_\star$, $\bar{v}(r) := Q^{1/2} \bar{u}(r)$, and $\mathcal{I}_Q := \mathrm{Var}(\bar{v}(r)) = Q^{1/2} \mathcal{I} Q^{1/2} \succeq 0$. Fix a rank-$k_{\mathrm{budget}}$ Euclidean projector $\widetilde{\Pi}$ acting on $v$-space and restrict estimators to $\widehat{v}(r) \in \mathrm{range}(\widetilde{\Pi})$ almost surely. By Lem. E.22 (applied to $(v_\star, \bar{v})$), the unique Bayes-optimal constrained estimator is $\widehat{v}_{\widetilde{\Pi}}(r) = \widetilde{\Pi}\, \bar{v}(r)$.

Write $v_\star = \bar{v} + z$ with $z := v_\star - \bar{v}$. Then $\mathbb{E}[z \mid r] = 0$ and $\mathrm{Var}(v_\star) = \mathrm{Var}(\bar{v}) + \mathbb{E}[\mathrm{Var}(v_\star \mid r)] = \mathcal{I}_Q + \mathbb{E}[\mathrm{Var}(v_\star \mid r)]$.

Moreover, $\bar{v}$ and $z$ are uncorrelated (and jointly Gaussian), so the cross term vanishes:

$$\begin{aligned}
\mathcal{R}(\widetilde{\Pi}) = \mathbb{E}[\|\widehat{v}_{\widetilde{\Pi}}(r) - v_\star\|_2^2] &= \mathbb{E}[\|\widetilde{\Pi}\bar{v} - (\bar{v} + z)\|_2^2] \\
&= \mathbb{E}[\|(I - \widetilde{\Pi})\bar{v}\|_2^2] + \mathbb{E}[\|z\|_2^2] \\
&= \mathrm{tr}((I - \widetilde{\Pi})\mathcal{I}_Q) + \mathrm{tr}(\mathbb{E}[\mathrm{Var}(v_\star \mid r)]) \\
&= \mathrm{tr}(\mathrm{Var}(v_\star)) - \mathrm{tr}(\widetilde{\Pi}\mathcal{I}_Q) \\
&= \mathrm{tr}(Q^{1/2} \Sigma Q^{1/2}) - \mathrm{tr}(\widetilde{\Pi}\mathcal{I}_Q) \\
&= \mathrm{tr}(Q\Sigma) - \mathrm{tr}(\widetilde{\Pi}\mathcal{I}_Q).
\end{aligned}$$

Therefore, minimizing $\mathcal{R}(\widetilde{\Pi})$ over rank-$k_{\mathrm{budget}}$ projectors is equivalent to maximizing $\mathrm{tr}(\widetilde{\Pi}\mathcal{I}_Q)$. Let $\mathcal{I}_Q = \sum_{i=1}^{p} \lambda_i v_i v_i^\top$ with $\lambda_1 \geq \cdots \geq \lambda_p \geq 0$. The maximum of $\mathrm{tr}(\widetilde{\Pi}\mathcal{I}_Q)$ over rank-$k_{\mathrm{budget}}$ projectors is achieved by taking $\widetilde{\Pi}$ to project onto $\mathrm{span}\{v_1, \ldots, v_{k_{\mathrm{budget}}}\}$. $\qquad\square$

Let $J := J_w(P_n)$ denote the prompt Jacobian (random via random prompts). In this subsection, we take the query metric $Q \succeq 0$ to be global across prompts (e.g., $Q = \mathbb{E}[j_X j_X^\top]$ under the test-time query distribution). [2] Define $\mathcal{I}(J) := \Sigma J^\top (J \Sigma J^\top + \sigma^2 I_n)^{-1} J \Sigma$, $\mathcal{I}_Q(J) := Q^{1/2} \mathcal{I}(J) Q^{1/2}$, and $\mathcal{I}_{Q, \mathrm{glob}} := \mathbb{E}[\mathcal{I}_Q(J)]$, where the expectation is over the prompt distribution (equivalently over $J$).

**Corollary E.23** (Global optimal subspace). *Any rank-$k_{\mathrm{budget}}$ projector $\widetilde{\Pi}^\star$ onto the top-$k_{\mathrm{budget}}$ eigenspace of $\mathcal{I}_{Q, \mathrm{glob}}$ maximizes $\mathrm{tr}(\widetilde{\Pi}\, \mathcal{I}_{Q, \mathrm{glob}})$ and hence minimizes $\mathbb{E}[\mathcal{R}_{J,Q}(\widetilde{\Pi})]$ over rank-$k_{\mathrm{budget}}$ projectors, where $\mathcal{R}_{J,Q}(\widetilde{\Pi})$ is the conditional risk in Thm. 8.2.*

*Proof.* By Thm. 8.2 applied conditionally on $(J, Q)$, $\mathcal{R}_{J,Q}(\widetilde{\Pi}) = \mathrm{tr}(Q\Sigma) - \mathrm{tr}(\widetilde{\Pi}\, \mathcal{I}_Q(J))$. Taking expectation over random prompts (equivalently over $J$) yields $\mathbb{E}[\mathcal{R}_{J,Q}(\widetilde{\Pi})] = \mathbb{E}[\mathrm{tr}(Q\Sigma)] - \mathrm{tr}(\widetilde{\Pi}\, \mathcal{I}_{Q, \mathrm{glob}})$. Thus minimizing $\mathbb{E}[\mathcal{R}_{J,Q}(\widetilde{\Pi})]$ is equivalent to maximizing $\mathrm{tr}(\widetilde{\Pi}\, \mathcal{I}_{Q, \mathrm{glob}})$, achieved by the top-$k_{\mathrm{budget}}$ eigenspace of $\mathcal{I}_{Q, \mathrm{glob}}$. $\qquad\square$

### E.6.2. FINITE-SAMPLE PLUG-IN SUBSPACE SELECTION AND A REGRET BOUND

Suppose we estimate $\mathcal{I}_{Q, \mathrm{glob}}$ from $m$ prompts, yielding $\mathcal{I}_Q(J_1), \ldots, \mathcal{I}_Q(J_m)$, and define

$$\widehat{\mathcal{I}}_Q := \frac{1}{m} \sum_{j=1}^{m} \mathcal{I}_Q(J_j).$$

Let $\widehat{\widetilde{\Pi}}$ be the rank-$k_{\mathrm{budget}}$ projector onto the top-$k_{\mathrm{budget}}$ eigenspace of $\widehat{\mathcal{I}}_Q$, and let $\widetilde{\Pi}^\star$ be the analogous projector for $\mathcal{I}_{Q, \mathrm{glob}}$.

---

[2] If one instead lets $Q$ depend on the prompt, the same derivation holds conditionally, but mapping back to an $u$-space update subspace involves $Q^{-1/2}(P_n)$ and can become prompt-dependent.

**Assumption E.24** (Matrix-martingale concentration conditions for $\widehat{\mathcal{I}}_Q$). Let $\mathcal{F}_j := \sigma(J_1, \ldots, J_j)$ and define the centered increments $X_j := \mathcal{I}_Q(J_j) - \mathcal{I}_{Q,\text{glob}} \in \mathbb{R}^{d_Q \times d_Q}$. Assume:

**(A1)** (Martingale-difference mean.) $\mathbb{E}[X_j \mid \mathcal{F}_{j-1}] = 0$ a.s. for all $j$. (In particular, this holds when $J_j$ are i.i.d.)

**(A2)** (Uniform boundedness.) There exists $R > 0$ such that $\|X_j\|_{\text{op}} \leq R$ a.s. for all $j$.

**(A3)** (Predictable quadratic variation bound.) Let $V_m := \sum_{j=1}^m \mathbb{E}[X_j^2 \mid \mathcal{F}_{j-1}]$. Assume $\|V_m\|_{\text{op}} \leq \nu$ a.s. for some $\nu > 0$.

**Theorem E.25** (Subspace selection regret via matrix Bernstein/Freedman). *Under Assump. E.24, for any $\delta \in (0,1)$, with probability at least $1 - \delta$ over the $m$ prompts used to form $\widehat{\mathcal{I}}_Q$,*

$$\|\widehat{\mathcal{I}}_Q - \mathcal{I}_{Q,\text{glob}}\|_{\text{op}} \leq \varepsilon_{m,\delta}, \tag{23}$$

*where $\varepsilon_{m,\delta} := \frac{1}{m}\left(\sqrt{2\nu \log \frac{2d_Q}{\delta}} + \frac{2R}{3}\log\frac{2d_Q}{\delta}\right)$. Consequently, on the same event,*

$$\mathbb{E}[\mathcal{R}_{J,Q}(\widehat{\widetilde{\Pi}})] - \mathbb{E}[\mathcal{R}_{J,Q}(\widetilde{\Pi}^\star)] \leq 2k_{budget}\,\varepsilon_{m,\delta}. \tag{24}$$

*Proof.* Let $S_m := \sum_{j=1}^m X_j = m(\widehat{\mathcal{I}}_Q - \mathcal{I}_{Q,\text{glob}})$. Under Assump. E.24 (A1)–(A3), the self-adjoint matrix Freedman inequality (Tropp, 2012) states that for all $t \geq 0$,

$$\mathbb{P}(\|S_m\|_{\text{op}} \geq t) \leq 2d_Q \cdot \exp\left(-\frac{t^2}{2\nu + \frac{2Rt}{3}}\right).$$

Setting $t = \sqrt{2\nu \log \frac{2d_Q}{\delta}} + \frac{2R}{3}\log\frac{2d_Q}{\delta}$ yields $\mathbb{P}(\|S_m\|_{\text{op}} \geq t) \leq \delta$. Dividing by $m$ gives (23).

On the event $\|\widehat{\mathcal{I}}_Q - \mathcal{I}_{Q,\text{glob}}\|_{\text{op}} \leq \varepsilon_{m,\delta}$, write $\widehat{\mathcal{I}}_Q = \mathcal{I}_{Q,\text{glob}} + E$ with $\|E\|_{\text{op}} \leq \varepsilon_{m,\delta}$. Since $\widehat{\widetilde{\Pi}}$ maximizes $\text{tr}(\widetilde{\Pi}\widehat{\mathcal{I}}_Q)$ over rank-$k_{budget}$ projectors,

$$\text{tr}(\widehat{\widetilde{\Pi}}\,\widehat{\mathcal{I}}_Q) \geq \text{tr}(\widetilde{\Pi}^\star\widehat{\mathcal{I}}_Q).$$

Expanding $\widehat{\mathcal{I}}_Q = \mathcal{I}_{Q,\text{glob}} + E$ gives

$$\text{tr}(\widetilde{\Pi}^\star\mathcal{I}_{Q,\text{glob}}) - \text{tr}(\widehat{\widetilde{\Pi}}\,\mathcal{I}_{Q,\text{glob}}) \leq \text{tr}(\widetilde{\Pi}^\star E) - \text{tr}(\widehat{\widetilde{\Pi}}E).$$

For any rank-$k_{budget}$ projector $\widetilde{\Pi}$, $|\text{tr}(\widetilde{\Pi}E)| \leq k_{budget}\|E\|_{\text{op}}$, so the RHS is at most $2k_{budget}\varepsilon_{m,\delta}$. Finally, by the global-risk identity $\mathbb{E}[\mathcal{R}_{J,Q}(\widetilde{\Pi})] = \text{const} - \text{tr}(\widetilde{\Pi}\,\mathcal{I}_{Q,\text{glob}})$ (cf. Cor. E.23), this implies (24). $\square$

*Remark* E.26 (i.i.d. prompts). If $J_1, \ldots, J_m$ are i.i.d., then (A1) holds. One may take $\nu = m\|\mathbb{E}[X_1^2]\|_{\text{op}}$ and (23) becomes

$$\|\widehat{\mathcal{I}}_Q - \mathcal{I}_{Q,\text{glob}}\|_{\text{op}} \leq \sqrt{\frac{2\|\mathbb{E}[X_1^2]\|_{\text{op}}\log\frac{2d_Q}{\delta}}{m}} + \frac{2R}{3m}\log\frac{2d_Q}{\delta}.$$

---

> **Summary: Subspace design can be learned globally**
>
> - **Global mechanism:** average $\mathcal{I}_Q$ over prompts, take top eigenspace (App. E.6).
> - **Finite-sample stability:** plug-in top-eigenspace has a concentration and regret bound.

## F. Validity & Robustness

### F.1. Finite-step validity of the local linearization

In the main text, starting in Sec. 3, we suppose the locally linearized (kernel/NTK) regime. This appendix analyzes how accurately the linearized model represents the actual nonlinear gradient updates used in prompt-only TTT. We show that this linear model is a reliable approximation in Theorem F.5. As long as the local conditions are smooth and the step size is moderate, the difference between the true nonlinear updates and the linear version remains small for a few steps ($T$).

#### F.1.1. NONLINEAR PROMPT DYNAMICS AND LINEARIZED DYNAMICS

Given a prompt $P_n$, we define $s_w(P_n) \in \mathbb{R}^n$ as the vector of predictions produced by the model at weights $w$ for the $n$ supervision points. Let $w_0$ be pretrained weights and $r := y - s_{w_0}(P_n) \in \mathbb{R}^n$ the base residuals. We restrict test-time updates to a $d$-dimensional subspace via $w(\theta) = w_0 + A\theta$ (Assumption 3.1).

Define the in-prompt prediction shift $s(\theta) := s_{w(\theta)}(P_n) - s_{w_0}(P_n) \in \mathbb{R}^n$ with $s(0) = 0$. Consider the squared prompt loss

$$L(\theta) := \frac{1}{2\sigma^2}\|r - s(\theta)\|_2^2. \tag{25}$$

Let the Jacobian in update parameters be $J(\theta) := \nabla_\theta s(\theta) \in \mathbb{R}^{n \times d}$ and the fixed Jacobian be $\Phi := J(0) \in \mathbb{R}^{n \times d}$.

**Nonlinear TTT (true GD).** Gradient descent with step size $\eta > 0$ on (25) is

$$\theta_{t+1} = \theta_t - \eta\nabla L(\theta_t) = \theta_t + \rho\,J(\theta_t)^\top(r - s(\theta_t)), \tag{26}$$

where $\theta_0 = 0$ and $\rho := \eta/\sigma^2$. Define the induced prompt-space correction and residual $f_t := s(\theta_t) \in \mathbb{R}^n$, and $e_t := r - f_t = r - s(\theta_t) \in \mathbb{R}^n$.

**Linearized TTT (fixed Jacobian).** The linearized model replaces $s(\theta)$ by $\Phi\theta$. The corresponding GD iterates satisfy

$$\theta_{t+1}^{\text{lin}} = \theta_t^{\text{lin}} + \rho\,\Phi^\top(r - \Phi\theta_t^{\text{lin}}), \quad \theta_0^{\text{lin}} = 0, \tag{27}$$

with prompt-space correction $f_t^{\text{lin}} := \Phi\theta_t^{\text{lin}} \in \mathbb{R}^n$ and residual $e_t^{\text{lin}} := r - f_t^{\text{lin}}$. Let the (prompt-dependent) restricted

kernel at initialization be $K_0 := \Phi\Phi^\top \succeq 0$. Then, $f_t^{\text{lin}}$ satisfies the closed recursion

$$f_{t+1}^{\text{lin}} = f_t^{\text{lin}} + \rho K_0 \left(r - f_t^{\text{lin}}\right) = f_t^{\text{lin}} + \rho K_0\, e_t^{\text{lin}}, \quad (28)$$

and hence $f_T^{\text{lin}} = \left(I_n - (I_n - \rho K_0)^T\right)r$.

### F.1.2. FINITE-STEP DEVIATION BOUND

We impose standard regularity assumptions in a neighborhood of $\theta = 0$.

**Assumption F.1** (Bounded and Lipschitz Jacobian). *There exist $R > 0$ and constants $B, L > 0$ such that for all $\theta, \theta' \in \mathbb{R}^d$ with $\|\theta\|_2 \le R$ and $\|\theta'\|_2 \le R$,*

$$\|J(\theta)\|_{\text{op}} \le B, \quad (29)$$
$$\|J(\theta) - J(\theta')\|_{\text{op}} \le L\,\|\theta - \theta'\|_2. \quad (30)$$

Assumption F.1 implies a second-order Taylor remainder bound.

**Lemma F.2** (Second-order remainder bound). *Under Assumption F.1, for any $\theta, u \in \mathbb{R}^d$ such that $\|\theta\|_2 \le R$ and $\|\theta + u\|_2 \le R$, $\|s(\theta + u) - s(\theta) - J(\theta)u\|_2 \le \frac{L}{2}\|u\|_2^2$ holds.*

*Proof.* By the integral form of Taylor's theorem, $s(\theta + u) - s(\theta) = \int_0^1 J(\theta + tu)\, u\, dt$. Hence, $s(\theta+u) - s(\theta) - J(\theta)u = \int_0^1 (J(\theta+tu) - J(\theta))u\, dt$, and (30) yields $\|s(\theta+u) - s(\theta) - J(\theta)u\|_2 \le \int_0^1 Lt\|u\|_2^2\, dt = \frac{L}{2}\|u\|_2^2$. $\square$

Suppose the step-size regime

$$0 < \rho \le \frac{1}{B^2}, \quad (31)$$

which ensures $\|I_n - \rho K_t\|_{\text{op}} \le 1$ for all $t$ as long as $\|J(\theta_t)\|_{\text{op}} \le B$, where $K_t := J(\theta_t)J(\theta_t)^\top$.

**Assumption F.3** (Iterate stability). *For a given $T \in \mathbb{N}$, the nonlinear GD iterates (26) satisfy $\|\theta_t\|_2 \le R$ for all $t = 0, 1, \dots, T$.*

Define the time-varying kernel and the linearized kernel $K_t := J(\theta_t)J(\theta_t)^\top \succeq 0$, and $K_0 := J(0)J(0)^\top = \Phi\Phi^\top$. We first derive a function-space recursion for the nonlinear dynamics.

**Lemma F.4** (Nonlinear prompt-space recursion). *Under Assumption F.1, if $\|\theta_t\|_2 \le R$ and $\|\theta_{t+1}\|_2 \le R$, then $f_{t+1} = f_t + \rho K_t e_t + \xi_t$, where the remainder term $\xi_t \in \mathbb{R}^n$ satisfies $\|\xi_t\|_2 \le \frac{L}{2}\|\theta_{t+1} - \theta_t\|_2^2$.*

*Proof.* Let $\Delta\theta_t := \theta_{t+1} - \theta_t = \rho J(\theta_t)^\top e_t$ by (26). Apply Lem. F.2 with $\theta = \theta_t$ and $u = \Delta\theta_t$:

$$f_{t+1} - f_t = s(\theta_t + \Delta\theta_t) - s(\theta_t) = J(\theta_t)\Delta\theta_t + \xi_t,$$

where $\|\xi_t\|_2 \le \frac{L}{2}\|\Delta\theta_t\|_2^2$. Since $J(\theta_t)\Delta\theta_t = \rho J(\theta_t)J(\theta_t)^\top e_t = \rho K_t e_t$, we obtain the statement. $\square$

We now bound the deviation $f_T - f_T^{\text{lin}}$.

**Theorem F.5** (Finite-step deviation between nonlinear and linearized prompt corrections). *Fix a $T \in \mathbb{N}$. Assume Assumptions F.1 and F.3, and the step-size condition (31). Let $f_T = s(\theta_T)$ be the nonlinear prompt correction after $T$ steps of (26), and let $f_T^{\text{lin}} = \Phi\theta_T^{\text{lin}}$ be the linearized prompt correction after $T$ steps of (27). Define the residual scale inside the stability ball $E_R := \sup_{\|\theta\|_2 \le R} \|r - s(\theta)\|_2$. Then*

$$\|f_T - f_T^{\text{lin}}\|_2 \le \frac{3}{2}\, L\, B^2\, \rho^2\, T^2 \left(\max\{E_R, \|r\|_2\}\right)^2$$
$$\le \frac{3}{2}\, L\, B^2\, \rho^2\, T^2\, E_R^2. \quad (32)$$

*In particular, for fixed local regularity constants $(B, L)$ and residual scale $E_R$, the discrepancy is $O\left((\rho T)^2\right)$.*

*Proof.* Define the discrepancy $\delta_t := f_t - f_t^{\text{lin}} \in \mathbb{R}^n$. By Lem. F.4 and (28),

$$\begin{aligned}
\delta_{t+1} &= \left(f_t + \rho K_t e_t + \xi_t\right) - \left(f_t^{\text{lin}} + \rho K_0 e_t^{\text{lin}}\right) \\
&= \delta_t + \rho K_t(r - f_t) - \rho K_0(r - f_t^{\text{lin}}) + \xi_t \\
&= \delta_t + \rho K_t(e_t^{\text{lin}} - \delta_t) - \rho K_0 e_t^{\text{lin}} + \xi_t \\
&= (I_n - \rho K_t)\delta_t + \rho(K_t - K_0)e_t^{\text{lin}} + \xi_t.
\end{aligned}$$

Taking norms and using $\|I_n - \rho K_t\|_{\text{op}} \le 1$ (from (31) and $\|J(\theta_t)\|_{\text{op}} \le B$),

$$\|\delta_{t+1}\|_2 \le \|\delta_t\|_2 + \rho\|K_t - K_0\|_{\text{op}}\|e_t^{\text{lin}}\|_2 + \|\xi_t\|_2. \quad (33)$$

Under (31), $0 \preceq I_n - \rho K_0 \preceq I_n$, so $e_t^{\text{lin}} = (I_n - \rho K_0)^t r$ implies $\|e_t^{\text{lin}}\|_2 \le \|r\|_2$ for all $t$.

Write $J_t := J(\theta_t)$. Then, $K_t - K_0 = J_t J_t^\top - J_0 J_0^\top = (J_t - J_0)J_t^\top + J_0(J_t - J_0)^\top$, so $\|K_t - K_0\|_{\text{op}} \le \|J_t - J_0\|_{\text{op}}\|J_t\|_{\text{op}} + \|J_0\|_{\text{op}}\|J_t - J_0\|_{\text{op}} \le 2B\,\|J_t - J_0\|_{\text{op}}$. By (30), $\|J_t - J_0\|_{\text{op}} \le L\|\theta_t\|_2$. Moreover, from (26) and (29), we have

$$\begin{aligned}
\|\theta_{t+1} - \theta_t\|_2 &= \rho\|J(\theta_t)^\top e_t\|_2 \\
&\le \rho\,\|J(\theta_t)\|_{\text{op}}\,\|e_t\|_2 \\
&\le \rho\, B\, E_R,
\end{aligned}$$

since $\|e_t\|_2 = \|r - s(\theta_t)\|_2 \le E_R$ for $\|\theta_t\|_2 \le R$. Thus $\|\theta_t\|_2 \le t\,\rho B E_R$, and hence

$$\|K_t - K_0\|_{\text{op}} \le 2BL\|\theta_t\|_2 \le 2LB^2\,\rho\, t\, E_R. \quad (34)$$

From Lem. F.4 and the bound on $\|\theta_{t+1} - \theta_t\|_2$,

$$\|\xi_t\|_2 \le \frac{L}{2}\left(\rho B E_R\right)^2 = \frac{L}{2}\,B^2\rho^2 E_R^2. \quad (35)$$

Plug (34), $\|e_t^{\text{lin}}\|_2 \le \|r\|_2$, and (35) into (33):

$$\|\delta_{t+1}\|_2 \le \|\delta_t\|_2 + \rho(2LB^2\rho t E_R)\|r\|_2 + \frac{L}{2}B^2\rho^2 E_R^2.$$

Let $E := \max\{E_R, \|r\|_2\}$. Then $\|r\|_2 \leq E$ and $E_R \leq E$, so

$$\|\delta_{t+1}\|_2 \leq \|\delta_t\|_2 + 2LB^2\rho^2\, t\, E^2 + \frac{L}{2}B^2\rho^2 E^2.$$

Since $\delta_0 = 0$, summing from $t = 0$ to $T-1$ gives

$$\|\delta_T\|_2 \leq 2LB^2\rho^2 E^2 \sum_{t=0}^{T-1} t + \frac{L}{2}B^2\rho^2 E^2 \sum_{t=0}^{T-1} 1$$

$$= LB^2\rho^2 E^2 T(T-1) + \frac{L}{2}B^2\rho^2 E^2 T$$

$$\leq \frac{3}{2}LB^2\rho^2 E^2 T^2,$$

which is (32). $\qquad\square$

**Corollary F.6** (Kernel drift along nonlinear TTT)**.** *Under the assumptions of Thm. F.5, for every $t \leq T$,*

$$\|K_t - K_0\|_{\mathrm{op}} \leq 2LB^2\, \rho\, t\, E_R.$$

**Deviation bound for query predictions**  Thm. F.5 controls discrepancies on the prompt supervision values. A similar bound can be derived for query-point predictions under additional smoothness assumptions. Concretely, suppose the query prediction shift

$$\Delta(x; \theta) := f_{w(\theta)}(x; P_n) - f_{w_0}(x; P_n)$$

is twice differentiable in $\theta$ on the same ball $\{\|\theta\|_2 \leq R\}$, with Jacobian $\nabla_\theta \Delta(x; \theta)$ bounded and Lipschitz uniformly in $\theta$. Then, one can repeat the argument above to show that the gap between the nonlinear query correction $\Delta(x; \theta_T)$ and its linearized form $\phi(x; P_n)^\top \theta_T^{\mathrm{lin}}$ is $O((\rho T)^2)$. We omit the algebraic details to avoid repetition: the proof follows the same contraction, kernel-drift, Taylor-remainder decomposition as in Thm. F.5, merely replacing the prompt map with the query map.

### F.2. Generalizing Filter Mismatch to Non-Gaussian Priors

In Sec. 5, we analyzed the Bayes gap assuming Gaussianity. This appendix discusses the robustness of the mode-wise MSE decomposition and the squared mismatch identity. The statements require only independence and finite second moments.

**Proposition F.7** (Mode-wise MSE and squared mismatch without Gaussianity)**.** *Fix $K \succeq 0$ and suppose that $r = f_\star + \varepsilon$, $\mathbb{E}[f_\star] = 0$, $\mathrm{Cov}(f_\star) = C_\star$  $\mathbb{E}[\varepsilon] = 0$, $\mathrm{Cov}(\varepsilon) = \sigma^2 I_n$, $f_\star \perp \varepsilon$. Let $S$ be any symmetric matrix such that $SK = KS$. Then, there exists an orthogonal matrix $U$ such that $K$ and $S$ are simultaneously diagonalized as $K = U\mathrm{diag}(\lambda_1, \ldots, \lambda_n)\, U^\top$, $S = U\mathrm{diag}(s_1, \ldots, s_n)\, U^\top$. Define $c_i := u_i^\top C_\star u_i = \mathrm{Var}(u_i^\top f_\star) \geq 0$, where $u_i$ is the $i$th*

*column of $U$. Then, the MSE of $\hat{f}_S := Sr$ satisfies*

$$\mathbb{E}\Big[\|\hat{f}_S - f_\star\|_2^2 \,\Big|\, K, S\Big] = \sum_{i=1}^n \Big\{c_i(1 - s_i)^2 + \sigma^2 s_i^2\Big\}. \quad (36)$$

*Moreover, for each coordinate $i$, the unique minimizer over $s_i \in \mathbb{R}$ is $s_i^\star = \frac{c_i}{c_i + \sigma^2}$ and the excess risk admits the exact squared-mismatch identity*

$$\mathbb{E}\Big[\|\hat{f}_S - f_\star\|_2^2 \,\Big|\, K, S\Big] - \mathbb{E}\Big[\|\hat{f}_{S^\star} - f_\star\|_2^2 \,\Big|\, K, S\Big]$$

$$= \sum_{i=1}^n (c_i + \sigma^2)\,(s_i - s_i^\star)^2, \quad (37)$$

*where $S^\star := U\mathrm{diag}(s_1^\star, \ldots, s_n^\star)\, U^\top$.*

This proposition suggests that, relying only on second-moment conditions (no Gaussianity), the estimation error decomposes into independent mode-wise shrinkage problems in the eigenbasis of $K$. The optimal shrinkage is given by $s_i^\star = \frac{c_i}{c_i + \sigma^2}$, and the excess risk admits an exact identity as the weighted sum of squared mismatches $(s_i - s_i^\star)^2$.

*Proof.* Let $\tilde{r} := U^\top r$, $\tilde{f} := U^\top f_\star$, and $\tilde{\varepsilon} := U^\top \varepsilon$. Since $U$ is orthogonal, $\|\hat{f}_S - f_\star\|_2^2 = \|\widetilde{\hat{f}} - \tilde{f}\|_2^2$, where $\widetilde{\hat{f}} = \mathrm{diag}(s_i)\tilde{r}$ and $\tilde{r} = \tilde{f} + \tilde{\varepsilon}$. Thus,

$$\|\widetilde{\hat{f}} - \tilde{f}\|_2^2 = \sum_{i=1}^n (s_i \tilde{r}_i - \tilde{f}_i)^2 = \sum_{i=1}^n \big((s_i - 1)\tilde{f}_i + s_i\tilde{\varepsilon}_i\big)^2.$$

Taking expectations and using $f_\star \perp \varepsilon$ (hence $\mathbb{E}[\tilde{f}_i\tilde{\varepsilon}_i] = 0$), $\mathbb{E}[\tilde{f}_i^2] = c_i$, and $\mathbb{E}[\tilde{\varepsilon}_i^2] = \sigma^2$ yields (36).

For fixed $i$, the scalar risk is a strictly convex quadratic $c_i(1 - s_i)^2 + \sigma^2 s_i^2$, whose unique minimizer is $s_i^\star = \frac{c_i}{c_i + \sigma^2}$. Completing the square gives $c_i(1 - s)^2 + \sigma^2 s^2 = c_i(1 - s^\star)^2 + \sigma^2(s^\star)^2 + (c_i + \sigma^2)(s - s^\star)^2$, which implies (37) after summing over $i$. $\qquad\square$

*Remark* F.8 (Connection to the GP benchmark in Sec. 5)*.* If $C_\star = \tau^2 K$ for some $\tau^2 > 0$ (covariance alignment), then $c_i = \tau^2 \lambda_i$ and $s_i^\star = \lambda_i/(\lambda_i + \lambda_\star)$ with $\lambda_\star = \sigma^2/\tau^2$, recovering the shrinkage $g_\star(\lambda)$ used in the main text. If, in addition, $(f_\star, \varepsilon)$ are jointly Gaussian, then $\hat{f}_{S^\star}$ coincides with the Bayes posterior mean for that GP model.

### F.3. Extension to anisotropic priors

This subsection extends the GP benchmark by relaxing the aligned covariance assumption (covariance proportional to $K$) in Assumption 5.1, allowing anisotropic task variability in the update parameters. Specifically, we show that appropriate preconditioning yields Bayes-consistent filtering under the induced anisotropic geometry.

**Assumption F.9** (Anisotropic Gaussian prior in update parameters). Let $\Phi \in \mathbb{R}^{n \times d}$ be the prompt feature matrix and consider the linearized residual model $r = \Phi \theta_\star + \varepsilon$, $\varepsilon \sim \mathcal{N}(0, \sigma^2 I_n)$. Assume an anisotropic prior on the latent update parameter signal $\theta_\star \sim \mathcal{N}(0, \tau^2 \Sigma)$, with $\Sigma \in \mathbb{R}^{d \times d}$ and $\lambda_\star := \sigma^2 / \tau^2$.

**Proposition F.10** (Bayes-optimal predictor on prompt values under anisotropic priors). *Under Assumption F.9, the latent prompt-level signal $f_\star := \Phi \theta_\star$ satisfies $f_\star \sim \mathcal{N}(0, \tau^2 \widetilde{K})$, $\widetilde{K} := \Phi \Sigma \Phi^\top$, and the Bayes-optimal estimator of $f_\star$ given $r$ under squared loss is*

$$f_{\text{Bayes}} = \mathbb{E}[f_\star \mid r] = \widetilde{K}(\widetilde{K} + \lambda_\star I_n)^{-1} r. \quad (38)$$

*Proof.* Since $\theta_\star \sim \mathcal{N}(0, \tau^2 \Sigma)$ and $f_\star = \Phi \theta_\star$ is linear in $\theta_\star$, we have $f_\star \sim \mathcal{N}(0, \tau^2 \Phi \Sigma \Phi^\top) = \mathcal{N}(0, \tau^2 \widetilde{K})$. With $r = f_\star + \varepsilon$ and $\varepsilon \sim \mathcal{N}(0, \sigma^2 I_n)$, Gaussian conditioning yields the posterior mean (38). $\square$

**Theorem F.11** (Preconditioned GD induces spectral filtering under $\widetilde{K}$). *Under Assumption F.9, consider preconditioned GD in $\theta$-space*

$$\theta_{t+1} = \theta_t + \rho \, \Sigma \, \Phi^\top (r - \Phi \theta_t), \quad (39)$$

*where $\theta_0 = 0$ and $\rho := \eta / \sigma^2$. Let $\hat{f}_t := \Phi \theta_t$. Then, $\hat{f}_{t+1} = \hat{f}_t + \rho \widetilde{K}(r - \hat{f}_t)$ holds, where $\widetilde{K} := \Phi \Sigma \Phi^\top$ and hence, for any $T \geq 0$, $\hat{f}_T = \widetilde{G}_T r$ holds, where $\widetilde{G}_T := I_n - (I_n - \rho \widetilde{K})^T$.*

Consequently, all spectral-filter and Bayes-gap results for TTT apply verbatim after replacing $K$ by $\widetilde{K}$: TTT selects an implicit prior under the geometry induced by $\widetilde{K}$, and the Bayes-optimal shrinkage on prompt values is $\widetilde{K}(\widetilde{K} + \lambda_\star I_n)^{-1}$.

*Proof.* Multiply (39) by $\Phi$ to get

$$\hat{f}_{t+1} = \Phi \theta_{t+1} = \Phi \theta_t + \rho \Phi \Sigma \Phi^\top (r - \Phi \theta_t)$$
$$= \hat{f}_t + \rho \widetilde{K}(r - \hat{f}_t).$$

Solving the linear recursion concludes the proof. $\square$

*Remark* F.12 (Interpretation: whitening and Bayes-consistency). Under the anisotropic prior, the Bayes-optimal predictor depends on $\widetilde{K} = \Phi \Sigma \Phi^\top$ rather than $K = \Phi \Phi^\top$. Preconditioning by $\Sigma$ reshapes the effective geometry so that the induced GD filter family can be viewed as Bayes-consistent shrinkage after whitening the update parameter prior.

**Summary: robustness of the spectral theory**

- **Even non-Gaussian:** excess risk is still a squared spectral shrinkage mismatch under 2nd moments (App. F.2).
- **Anisotropic tasks:** preconditioning replaces $K$ by $\widetilde{K} = \Phi \Sigma \Phi^\top$ (App. F.3).

# G. Gradient flow and random-matrix theory view of the update horizon $T$

This appendix explains how the update horizon is governed by the spectrum of the prompt geometry $K(P_n)$. We recast discrete $T$-step updates via the continuous diffusion time $t \approx \rho T$, under which the estimator becomes an explicit spectral filter $g_t(\lambda) = 1 - e^{-t\lambda}$, making "how far to adapt" a one-parameter shrinkage rule over eigenmodes. We then use this view to define an oracle (risk-optimal) horizon and to motivate deterministic proxies from random-matrix concentration, clarifying why the best $T$ is inherently prompt- and subspace-dependent.

## G.1. Gradient flow as the canonical horizon parameter

Fix a prompt of length $n$. Let $K \in \mathbb{R}^{n \times n}$ be the prompt geometry, and $r \in \mathbb{R}^n$ be the prompt residual vector. Write an eigendecomposition $K = U \operatorname{diag}(\lambda_1, \ldots, \lambda_n) U^\top$ with $\lambda_i \geq 0$.

**Proposition G.1** (Gradient flow solution and spectral filter). *Consider the gradient-flow ODE $\dot{f}(t) = K(r - f(t))$ with $f(0) = 0$. Then*

$$f(t) = (I - e^{-tK}) r = U \operatorname{diag}(g_t(\lambda_1), \ldots, g_t(\lambda_n)) U^\top r,$$

*where $g_t(\lambda) := 1 - e^{-t\lambda}$.*

*Proof.* Let $e(t) := r - f(t)$. Then $\dot{e}(t) = -Ke(t)$ and $e(0) = r$, so $e(t) = e^{-tK} r$, hence $f(t) = r - e(t)$. Diagonalization yields the scalar filter. $\square$

Each eigenmode $u_i$ is learned at rate $\lambda_i$, so large-$\lambda$ modes fit quickly and small-$\lambda$ modes fit slowly.

Note that our discrete GD recursion is $f_{t+1} = f_t + \rho K(r - f_t)$. The gradient flow is written in the rescaled time that absorbs $\rho$, and thus the canonical horizon parameter is the diffusion time $t \simeq \rho T$.

Recall the discrete $T$-step GD filter (main text)

$$g_T(\lambda) = 1 - (1 - \rho \lambda)^T, \qquad 0 < \rho < 1/\lambda_{\max}(K).$$

**Lemma G.2** (Discrete GD $\approx$ gradient flow). *Fix $\lambda \geq 0$ and*

*assume $\rho\lambda < 1$. Let $t := \rho T$. Then*

$$0 \le g_T(\lambda) - g_t(\lambda) = e^{-t\lambda} - (1 - \rho\lambda)^T \le \frac{T(\rho\lambda)^2}{2(1 - \rho\lambda)} e^{-t\lambda}.$$

*Proof.* Let $x = \rho\lambda \in [0, 1)$. Since $1 - x \le e^{-x}$, $(1 - x)^T \le e^{-xT}$, giving $g_T \ge g_t$. Also $\log(1 - x) = -x - \sum_{k\ge 2} x^k/k \ge -x - x^2/(2(1 - x))$, hence $(1 - x)^T \ge e^{-xT} \exp\{-Tx^2/(2(1 - x))\}$. Rearranging and using $1 - e^{-y} \le y$ concludes the proof. $\square$

### G.2. Bayes gap for a fixed prompt: oracle diffusion time and mixture

We evaluate horizons under the aligned GP benchmark (used throughout the main text for filter-match analysis).

**Assumption G.3** (Aligned GP benchmark)**.** *Conditioned on $K$, $f_\star \sim \mathcal{N}(0, \tau^2 K)$ and $r = f_\star + \varepsilon$ with $\varepsilon \sim \mathcal{N}(0, \sigma^2 I_n)$ independent. Define $\lambda_\star := \sigma^2/\tau^2$ and $g_\star(\lambda) := \lambda/(\lambda + \lambda_\star)$.*

**Proposition G.4** (Mode-wise risk and mismatch identity (recall))**.** *Under Assump. G.3 and conditioned on $K = U \mathrm{diag}(\lambda_i) U^\top$, for any measurable $g$ with $g(0) = 0$, the spectral estimator $\hat{f}_g = g(K)r$ satisfies*

$$\mathbb{E}\Big[\|\hat{f}_g - f_\star\|_2^2 \mid K\Big] = \sum_{i=1}^n \Big[\tau^2\lambda_i(1 - g(\lambda_i))^2 + \sigma^2 g(\lambda_i)^2\Big].$$

*Moreover, $g_\star(\lambda) = \lambda/(\lambda + \lambda_\star)$ uniquely minimizes the bracketed term, and the excess Bayes risk is*

$$\mathbb{E}\Big[\|\hat{f}_g - f_\star\|_2^2 \mid K\Big] - \mathbb{E}\big[\|g_\star(K)r - f_\star\|_2^2 \mid K\big]$$
$$= \sum_{i=1}^n a_i\big(g(\lambda_i) - g_\star(\lambda_i)\big)^2,$$

*where $a_i := \tau^2\lambda_i + \sigma^2$.*

Specializing to gradient flow $\hat{f}(t) = g_t(K)r$ with $g_t(\lambda) = 1 - e^{-t\lambda}$ gives a one-parameter family of risks. Define the oracle time

$$t^\star(K) \in \arg\min_{t \ge 0} \mathbb{E}\big[\|g_t(K)r - f_\star\|_2^2 \mid K\big].$$

#### G.2.1. MODE-WISE IDEAL TIME

For a single eigenvalue $\lambda > 0$, matching $g_t(\lambda) = g_\star(\lambda)$ yields an explicit ideal time.

**Lemma G.5** (Mode-wise Bayes-matching time)**.** *For $\lambda > 0$, the unique $t$ satisfying $g_t(\lambda) = g_\star(\lambda)$ is*

$$t_\lambda^\star = \frac{1}{\lambda}\log\Big(1 + \frac{\lambda}{\lambda_\star}\Big). \qquad (40)$$

*Proof.* $g_t(\lambda) = 1 - e^{-t\lambda}$ equals $g_\star(\lambda) = \lambda/(\lambda + \lambda_\star)$ iff $e^{-t\lambda} = \lambda_\star/(\lambda + \lambda_\star)$, i.e. (40). $\square$

**Why a single time is a compromise.** The Bayes-matching time $t_\lambda^\star$ depends on $\lambda$, so unless the spectrum is nearly degenerate, there is no single $t$ that simultaneously matches $g_t(\lambda_i) = g_\star(\lambda_i)$ for all modes. Consequently, any fixed horizon must trade off over-fitting fast modes (large $\lambda$) against under-fitting slow modes (small $\lambda$). This motivates selecting a representative eigenvalue (e.g. $\mu(K)$) or using a time-mixture.

A simple near-oracle proxy matches at a representative eigenvalue. Define the weighted mean

$$\mu(K) := \frac{\sum_{i=1}^n a_i\lambda_i}{\sum_{i=1}^n a_i}, \qquad a_i = \tau^2\lambda_i + \sigma^2. \qquad (41)$$

Set the match-at-$\mu$ time

$$t_\mu := \frac{1}{\mu(K)}\log\Big(1 + \frac{\mu(K)}{\lambda_\star}\Big) \Rightarrow g_{t_\mu}(\mu(K)) = g_\star(\mu(K)). \qquad (42)$$

**Proposition G.6** (Bayes as an exponential mixture of diffusion times)**.** *For $\lambda_\star > 0$ and all $\lambda \ge 0$,*

$$g_\star(\lambda) = \int_0^\infty \lambda_\star e^{-\lambda_\star s}\big(1 - e^{-s\lambda}\big)\, ds.$$

*The same identity holds at the matrix level $g_\star(K) = \int_0^\infty \lambda_\star e^{-\lambda_\star s}(I - e^{-sK})ds$.*

*Proof.* Compute the scalar integral: $\int_0^\infty \lambda_\star e^{-\lambda_\star s}ds = 1$ and $\int_0^\infty \lambda_\star e^{-(\lambda_\star + \lambda)s}ds = \lambda_\star/(\lambda_\star + \lambda)$. Subtract to obtain $g_\star(\lambda)$. Apply spectral calculus for the matrix form. $\square$

**Interpretation (random stopping time).** Let $S \sim \mathrm{Exp}(\lambda_\star)$ be an exponential random time. Then, Prop. G.6 is equivalent to $g_\star(K)r = \mathbb{E}[(I - e^{-SK})r]$. In words, the Bayes shrinkage equals the expected gradient-flow predictor evaluated at a random diffusion time. This highlights why broad spectra are problematic for a single deterministic $t$ and Bayes effectively averages over a range of times.

#### G.2.2. A CONTINUOUS-TIME SPECTRAL CONCENTRATION BOUND

Define the mismatch $d_t(\lambda) := g_t(\lambda) - g_\star(\lambda)$.

**Proposition G.7** (Spectral concentration of mismatch (gradient flow))**.** *For any $t \ge 0$, $\sum_{i=1}^n a_i\, d_t(\lambda_i)^2 \le 2\Big(\sum_{i=1}^n a_i\Big) d_t(\mu)^2 + 2\Big(t + \frac{1}{\lambda_\star}\Big)^2 \sum_{i=1}^n a_i(\lambda_i - \mu)^2$, where $a_i = \tau^2\lambda_i + \sigma^2$ and $\mu = \frac{\sum_i a_i\lambda_i}{\sum_i a_i}$.*

*Proof.* $g_t'(\lambda) = te^{-t\lambda} \le t$ so $g_t$ is $t$-Lipschitz. Also $g_\star'(\lambda) = \frac{\lambda_\star}{(\lambda + \lambda_\star)^2} \le 1/\lambda_\star$. Hence $d_t$ is $(t + 1/\lambda_\star)$-Lipschitz. For each $i$, $|d_t(\lambda_i)| \le |d_t(\mu)| + (t + 1/\lambda_\star)|\lambda_i - \mu|$. Square and sum with weights $a_i$ to obtain the inequality. $\square$

**Interpretation (horizon instability).** The risk identity decomposes the error into two meaningful components: $d_t(\mu)^2$ represents the mismatch at a representative eigenvalue, while $\sum_i a_i(\lambda_i - \mu)^2$ penalizes spectral spread via weighted variance. Notably, the Lipschitz factor $(t + 1/\lambda_\star)$ grows linearly with $t$, suggesting that longer diffusion times heighten the procedure's sensitivity to spectral heterogeneity. This scaling relationship provides a principled explanation for horizon instability, particularly why performance degrades when the prompt geometry $K$ possesses a broad spectrum.

> **Summary: Why $T$ is prompt-dependent (diffusion intuition)**
>
> - **Instability comes from spectral heterogeneity:** wide spectra make a single $t$ overfit fast modes and underfit slow ones, so performance becomes sensitive to $T$.
> - **Bayes behaves like mixing times:** the Bayes shrinkage equals a mixture over stopping times.

### G.3. RMT: deterministic oracle curves from asymptotic spectra

Define the per-token risk under gradient flow

$$\overline{\mathcal{R}}_{\text{in},n}(t \mid K) := \frac{1}{n}\sum_{i=1}^n\Big[\tau^2\lambda_i e^{-2t\lambda_i} + \sigma^2(1 - e^{-t\lambda_i})^2\Big].$$

**Spectral Concentration and Horizon Selection.** Although the prompt kernel $K$ is random, in high dimensions, its spectrum is highly predictable. Specifically, averages over eigenvalues $\frac{1}{n}\sum_{i=1}^n h(\lambda_i(K))$ fluctuate little and converge to integrals under a limiting law (e.g., the Marchenko–Pastur distribution). Since our normalized risk can be expressed as such an eigenvalue-average for each $t$ (i.e., $\overline{\mathcal{R}}_{\text{in},n}(t \mid K) = \frac{1}{n}\text{tr}\,\psi_t(K)$ for an appropriate function $\psi_t$), the entire risk curve $t \mapsto \overline{\mathcal{R}}_{\text{in},n}(t \mid K)$ and its minimizer admit deterministic approximations. This provides a rigorous foundation for using simple spectral moments, such as $\mu(K)$, as stable and robust horizon predictors.

**Assumption G.8** (Wishart kernel model). Let $\Phi = \frac{1}{\sqrt{d}}X \in \mathbb{R}^{n\times d}$ with i.i.d. mean-zero, variance-one subgaussian entries, and $n, d \to \infty$ with $n/d \to \gamma \in (0,\infty)$. Set $K = \Phi\Phi^\top = \frac{1}{d}XX^\top$.

**Proposition G.9** (Asymptotic risk under a Marchenko–Pastur geometry). *Let $\Phi \in \mathbb{R}^{n\times d}$ have i.i.d. entries $\Phi_{ij} \sim \mathcal{N}(0, 1/d)$ and set $K := \Phi\Phi^\top$. Assume $n, d \to \infty$ with $n/d \to \gamma \in (0,\infty)$. Under the aligned GP benchmark (Assump. G.3), consider the gradient-flow estimator*

$\hat{f}(t) = (I - e^{-tK})r$. *Then, almost surely,*

$$\frac{1}{n}\,\mathbb{E}\Big[\|\hat{f}(t) - f_\star\|_2^2 \,\Big|\, K\Big] \longrightarrow \int \psi_t(\lambda)\,\mu_\gamma(d\lambda),$$

*where $\psi_t(\lambda) := \tau^2\lambda e^{-2t\lambda} + \sigma^2(1 - e^{-t\lambda})^2$ and $\mu_\gamma$ is the Marchenko–Pastur law with aspect ratio $\gamma$, i.e. $\mu_\gamma(d\lambda) = \left(1 - \frac{1}{\gamma}\right)_+ \delta_0(d\lambda) + \frac{\sqrt{(\lambda_+ - \lambda)(\lambda - \lambda_-)}}{2\pi\gamma\lambda}\mathbf{1}_{[\lambda_-,\lambda_+]}(\lambda)\,d\lambda$, with $\lambda_\pm = (1 \pm \sqrt{\gamma})^2$.*

*Proof.* By Prop. G.4 with $g = g_t(\lambda) = 1 - e^{-t\lambda}$, conditioning on $K = U\text{diag}(\lambda_i)U^\top$ gives

$$\frac{1}{n}\,\mathbb{E}\Big[\|\hat{f}(t) - f_\star\|_2^2 \,\Big|\, K\Big] = \frac{1}{n}\sum_{i=1}^n\psi_t(\lambda_i),$$

where $\psi_t(\lambda) = \tau^2\lambda e^{-2t\lambda} + \sigma^2(1 - e^{-t\lambda})^2$. Let $\mu_n := \frac{1}{n}\sum_{i=1}^n \delta_{\lambda_i}$ be the empirical spectral measure of $K$. For $K = \Phi\Phi^\top = (1/d)XX^\top$ with $X_{ij} \sim \mathcal{N}(0,1)$, the Marchenko-Pastur theorem yields $\mu_n \Rightarrow \mu_\gamma$ almost surely as $n, d \to \infty$ with $n/d \to \gamma$ (Marčenko & Pastur, 1967; Bai & Silverstein, 2010). Since $\psi_t$ is bounded and continuous on $[0,\infty)$, we have $\int \psi_t\,d\mu_n \to \int \psi_t\,d\mu_\gamma$ almost surely. $\square$

#### G.3.1. MOMENT PROXY: DETERMINISTIC "REPRESENTATIVE EIGENVALUE"

The weighted mean eigenvalue $\mu(K)$ in (41) admits a deterministic limit under MP moments. Since $m_1(\gamma) := \int \lambda\,\nu_\gamma(d\lambda) = 1$ and $m_2(\gamma) := \int \lambda^2\,\nu_\gamma(d\lambda) = 1 + \gamma$,

**Proposition G.10** (Deterministic equivalent of $\mu(K)$). *Under Assump. G.8,*

$$\mu(K) \xrightarrow{\text{a.s.}} \mu_\infty := \frac{\tau^2 m_2(\gamma) + \sigma^2 m_1(\gamma)}{\tau^2 m_1(\gamma) + \sigma^2} = \frac{(1+\gamma) + \lambda_\star}{1 + \lambda_\star}. \tag{43}$$

*Thus the match-at-$\mu$ time concentrates around*

$$t_{\mu_\infty} = \frac{1}{\mu_\infty}\log\Big(1 + \frac{\mu_\infty}{\lambda_\star}\Big).$$

*Proof.* Write $\mu(K) = \frac{\tau^2\frac{1}{n}\sum\lambda_i^2 + \sigma^2\frac{1}{n}\sum\lambda_i}{\tau^2\frac{1}{n}\sum\lambda_i + \sigma^2}$ and use the Marchenko-Pastur theorem. $\square$

**Concentration replacement (what is random).** Under Assump. G.8, $K$ (hence $\{\lambda_i\}$, $\mu(K)$, and $\overline{\mathcal{R}}_{\text{in},n}$) is random. The limits $\nu_\gamma$, $\mu_\infty$, and $\mathcal{R}_\infty$ are deterministic. Standard concentration for linear spectral statistics implies $\mu(K)$ and $\overline{\mathcal{R}}_{\text{in},n}(t \mid K)$ concentrate around these limits (proof sketch omitted; see standard subgaussian random matrix concentration).

### G.4. Transformer head example: head choice ⇒ spectrum ⇒ horizon

We specialize to Value-matrix updates in Transformer head $h$ under the global-upstream regime. With attention matrix $\mathcal{A}^{(h)}$ (entries $\alpha_{ks}^{(h)}$), residual stream stack $Z \in \mathbb{R}^{n \times d_z}$, and upstream direction $\beta^{(h)}$, the head-induced prompt kernel is

$$K^{(h,V)} \approx \|\beta^{(h)}\|_2^2 \, \mathcal{A}^{(h)}(ZZ^\top)\mathcal{A}^{(h)\top}, \quad K = \sum_{h \in \mathcal{H}} K^{(h,V)}.$$

Thus head selection $\mathcal{H}$ changes the eigenvalues $\{\lambda_i(K)\}$ and therefore changes the near-oracle time (e.g. $t_\mu$ in (42)).

#### G.4.1. TWO REGIMES AND A TOY CALCULATION

**(i) Spiky / low-rank head.** Suppose $\mathcal{A}^{(h)}Z$ is approximately rank-one: $\mathcal{A}^{(h)}Z \approx \sqrt{\kappa}\, vu^\top$ with $\|v\|_2 = 1$. Then $K^{(h,V)} \approx \lambda_{\mathrm{sp}}\, vv^\top$ with spike

$$\lambda_{\mathrm{sp}} := \|\beta^{(h)}\|_2^2\, \kappa.$$

Toy horizon. If $K \approx \lambda_{\mathrm{sp}}\, vv^\top$ exactly, matching $g_t(\lambda_{\mathrm{sp}}) = g_\star(\lambda_{\mathrm{sp}})$ yields

$$t_{\mathrm{sp}} = \frac{1}{\lambda_{\mathrm{sp}}} \log\left(1 + \frac{\lambda_{\mathrm{sp}}}{\lambda_\star}\right), \qquad T_{\mathrm{sp}} \approx t_{\mathrm{sp}}/\rho.$$

**Intuition (spike ⇒ step-sensitivity).** A dominant spike makes the effective dynamics essentially one-dimensional and very fast. The residual energy along the spike direction decays on the scale $t \sim 1/\lambda_{\mathrm{sp}}$. Hence small changes in $T$ (or $\rho$) can move the predictor from under- to over-adaptation quickly, producing sharp performance cliffs.

**(ii) Mixing head / MP-like bulk.** If $\mathcal{A}^{(h)}Z$ behaves like a high-rank random feature matrix, then $K(\mathcal{H})$ can have an MP-like bulk. Let $d_\mathcal{H}$ denote the effective feature dimension contributed by selected heads, and $\gamma_\mathcal{H} := n/d_\mathcal{H}$. Then the deterministic proxy (43) suggests $\mu_\infty(\gamma_\mathcal{H}) = \frac{(1+\gamma_\mathcal{H})+\lambda_\star}{1+\lambda_\star}$, and $t_{\mu_\infty(\gamma_\mathcal{H})} = \frac{1}{\mu_\infty(\gamma_\mathcal{H})} \log\left(1 + \frac{\mu_\infty(\gamma_\mathcal{H})}{\lambda_\star}\right)$.

**Intuition.** Adding heads expands the effective dimension $d_\mathcal{H}$, thereby reducing the aspect ratio $\gamma_\mathcal{H}$. This adjustment reshapes the spectral distribution of $K(\mathcal{H})$ and, consequently, shifts the predicted near-oracle horizon, reflecting a higher adaptation capacity.

---

> **Summary: spectrum couples how far and which heads to adapt**
>
> - **Spiky heads:** large eigenvalues ⇒ faster diffusion ⇒ smaller near-oracle times.
> - **Mixing heads:** bulk spectra ⇒ horizons are well-approximated by deterministic equivalents.
>
> Which heads to adapt and how far to adapt are spectrally coupled through the prompt geometry.

## H. Transformer-specific derivations

### H.1. Technical details for Transformer Jacobian features

#### H.1.1. OUTER-PRODUCT JACOBIAN FOR LINEAR LAYERS

**Lemma H.1** (Outer-product Jacobian for linear layers (exact at $w_0$)). *Let $y = Wx$ with $W \in \mathbb{R}^{d_{\mathrm{out}} \times d_{\mathrm{in}}}$ be a linear layer in the computation graph, and let $s_i(w)$ be any scalar output. Evaluate all forward activations and backpropagated gradients at $w_0$, and define $a_i := x\big|_{w_0} \in \mathbb{R}^{d_{\mathrm{in}}}$ and $b_i := \nabla_y s_i(w)\big|_{w_0} \in \mathbb{R}^{d_{\mathrm{out}}}$. Then, using the column-major convention for $\mathrm{vec}(\cdot)$,*

$$\nabla_{\mathrm{vec}(W)} s_i(w)\big|_{w_0} = a_i \otimes b_i.$$

*Consequently, the kernel contribution induced by a full update of $W$ satisfies*

$$K_{ij}^{(W)} = \left\langle \nabla_{\mathrm{vec}(W)} s_i, \, \nabla_{\mathrm{vec}(W)} s_j \right\rangle = (a_i^\top a_j)\,(b_i^\top b_j).$$

*Proof.* By the chain rule for $y = Wx$, holding $x$ fixed when differentiating w.r.t. $W$, $\nabla_W s_i(w)\big|_{w_0} = \left(\nabla_y s_i(w)\big|_{w_0}\right)\left(x\big|_{w_0}\right)^\top = b_i a_i^\top$. Under the column-major convention, $\mathrm{vec}(ba^\top) = a \otimes b$, hence $\nabla_{\mathrm{vec}(W)} s_i(w)\big|_{w_0} = \mathrm{vec}(b_i a_i^\top) = a_i \otimes b_i$. Finally, $\langle a_i \otimes b_i, \, a_j \otimes b_j \rangle = (a_i^\top a_j)(b_i^\top b_j)$. □

#### H.1.2. LOCAL LINEARIZATION REGIME: FIXED FORWARD/BACKWARD FEATURES

**Assumption H.2** (Fixed forward/backward features in the local regime). When forming Jacobian features for selected parameter blocks and the induced geometry $K$, we evaluate all forward activations (e.g. residual streams $z_s$ and attention weights $\alpha_{ts}^{(h)}$) and all upstream gradients (e.g. $g_i$) at $w_0$ and treat them as constants under the local linearization. Equivalently, we ignore second-order feedback effects whereby changing one block perturbs intermediate activations/gradients used to define Jacobian features of another block.

**Remark (why attention can be treated as fixed for $W_V/W_O$ features).** For standard attention, $\alpha_{ts}^{(h)}$ depends on $(W_Q^{(h)}, W_K^{(h)})$ and the input activations, but not directly on $W_V^{(h)}$ or $W_O^{(h)}$. Thus, when differentiating w.r.t. $W_V^{(h)}$ or $W_O^{(h)}$ at $w_0$, holding $\alpha^{(h)}$ fixed is exact at the layer level; Assumption H.2 only rules out indirect, higher-order changes of $\alpha^{(h)}$ through upstream activation shifts when multiple blocks are updated. Similar "frozen-attention" simplifications also appear in theoretical analyses of attention in random-feature regimes (Fu et al., 2023).

### H.1.3. DERIVATION: VALUE-MATRIX UPDATES $W_V$

Fix a coordinate $i$ that supervises token $k = k(i)$ and let $g_i := \nabla_{o_k^{(h)}} s_i(w)\big|_{w_0}$ be the upstream gradient into $o_k^{(h)}$ at $w_0$. Under Assumption H.2, the first-order dependence on $W_V^{(h)}$ is

$$\delta s_i = \left\langle g_i,\ W_O^{(h)} \sum_{s \le k} \alpha_{ks}^{(h)} \delta W_V^{(h)} z_s \right\rangle + o(\|\delta W_V^{(h)}\|).$$

Define $\beta_i^{(h)} := (W_O^{(h)})^\top g_i \in \mathbb{R}^{d_h}$, and $m_i^{(h)} := \sum_{s \le k(i)} \alpha_{k(i)s}^{(h)} z_s \in \mathbb{R}^{d_{\text{model}}}$. Then $\delta s_i = \langle \beta_i^{(h)},\ \delta W_V^{(h)} m_i^{(h)} \rangle + o(\|\delta W_V^{(h)}\|)$, so by Lem. H.1 applied at $w_0$, $\nabla_{\text{vec}(W_V^{(h)})} s_i(w)\big|_{w_0} = m_i^{(h)} \otimes \beta_i^{(h)}$, $K_{ij}^{(h,V)} = (m_i^{(h)\top} m_j^{(h)})(\beta_i^{(h)\top} \beta_j^{(h)})$.

**Matrix form and PSD.** Under tokenwise supervision ($i \equiv k$), stacking $m_k^{(h)}$ and $\beta_k^{(h)}$ gives the Hadamard form (3). Since $M^{(h)} M^{(h)\top} \succeq 0$ and $\mathbf{B}^{(h)} \mathbf{B}^{(h)\top} \succeq 0$, the Schur product theorem implies $K^{(h,V)} \succeq 0$. The global-upstream specialization yields the attention-sandwiched Gram (4).

### H.1.4. DERIVATION: OUTPUT-MATRIX UPDATES $W_O$

If we update $W_O^{(h)}$ only, then (at $w_0$) $\delta s_i = \langle g_i,\ \delta W_O^{(h)} u_{k(i)}^{(h)} \rangle + o(\|\delta W_O^{(h)}\|)$, so Lem. H.1 yields $\nabla_{\text{vec}(W_O^{(h)})} s_i(w)\big|_{w_0} = u_{k(i)}^{(h)} \otimes g_i$, $K_{ij}^{(h,O)} = (u_{k(i)}^{(h)\top} u_{k(j)}^{(h)})(g_i^\top g_j)$. Since $u^{(h)} = \mathcal{A}^{(h)} V^{(h)}$ with $V^{(h)} := Z W_V^{(h)\top}$, this again exhibits attention-induced structure.

### H.1.5. DERIVATION: MLP BLOCK UPDATES

For an MLP block $h_k = \sigma(W_1 z_k)$ and $\text{mlp}_k = W_2 h_k$, updating $W_2$ yields Jacobian features $h_k \otimes g_k$ (Gram of hidden activations). Updating $W_1$ yields features $z_k \otimes (\sigma'(W_1 z_k) \odot (W_2^\top g_k))$, again an outer product (input × backpropagated gate).

### H.1.6. ADDITIVITY ACROSS UPDATED BLOCKS

If $\mathcal{B}$ is a set of disjoint parameter blocks that are updated, stacking Jacobians blockwise yields $\Phi = [\Phi^{(b)}]_{b \in \mathcal{B}}$ and hence $K = \Phi \Phi^\top = \sum_{b \in \mathcal{B}} K^{(b)}$.

## H.2. A simple stability bound for the step size

In the global-upstream regime (4) for a head $h$,

$$K^{(h,V)} = \|\beta^{(h)}\|_2^2 M^{(h)} M^{(h)\top}, \quad M^{(h)} = \mathcal{A}^{(h)} Z.$$

Therefore, $\lambda_{\max}(K^{(h,V)}) = \|\beta^{(h)}\|_2^2 \|M^{(h)}\|_{\text{op}}^2 \le \|\beta^{(h)}\|_2^2 \|\mathcal{A}^{(h)}\|_{\text{op}}^2 \|Z\|_{\text{op}}^2$. Since $K = \sum_{b \in \mathcal{B}} K^{(b)}$ is PSD, $\lambda_{\max}(K) \le \sum_{b \in \mathcal{B}} \lambda_{\max}(K^{(b)})$, so enlarging the updated block set tightens the admissible step-size range $0 < \rho < 1/\lambda_{\max}(K)$.

## H.3. Representable prompt subspace for value updates

**Lemma H.3** (Representable prompt subspace for $W_V$ updates (global-$\beta$ case))**.** *Assume tokenwise supervision and a global upstream direction $g_k \equiv g$, so that $\beta_k^{(h)} \equiv \beta^{(h)}$ with $\|\beta^{(h)}\| > 0$ for each head $h$. For a head $h$ with $W_V$ updates, $K^{(h,V)} = \|\beta^{(h)}\|_2^2 M^{(h)} M^{(h)\top}$ where $M^{(h)} = \mathcal{A}^{(h)} Z$. Hence $\text{range}(K^{(h,V)}) = \text{range}(M^{(h)}) = \text{range}(\mathcal{A}^{(h)} Z)$. If we update a head set $\mathcal{H}$ (and sum their kernel components), then*

$$\text{range}\left(\sum_{h \in \mathcal{H}} K^{(h,V)}\right) = \sum_{h \in \mathcal{H}} \text{range}(M^{(h)})$$
$$= \text{range}\left([M^{(h)}]_{h \in \mathcal{H}}\right),$$

*where $[M^{(h)}]_{h \in \mathcal{H}}$ denotes column-wise concatenation.*

*Proof.* In the global-$\beta$ case, $K^{(h,V)} = \|\beta^{(h)}\|_2^2 M^{(h)} M^{(h)\top}$, hence $\text{range}(K^{(h,V)}) = \text{range}(M^{(h)} M^{(h)\top}) = \text{range}(M^{(h)})$. For the sum, note that for symmetric PSD matrices $\{S_\ell\}$, $\text{null}(\sum_\ell S_\ell) = \cap_\ell \text{null}(S_\ell)$ because $x^\top (\sum_\ell S_\ell) x = 0$ implies $x^\top S_\ell x = 0$ for every $\ell$; by PSD-ness this gives $S_\ell x = 0$. Taking orthogonal complements yields $\text{range}(\sum_\ell S_\ell) = \sum_\ell \text{range}(S_\ell)$. Finally, $\sum_{h \in \mathcal{H}} \text{range}(M^{(h)}) = \text{range}([M^{(h)}]_{h \in \mathcal{H}})$ is standard. $\square$

## H.4. Computational cost, memory, and scalable approximations

This appendix complements Sections 7 and 8 by (i) making the computational overhead explicit (time, memory, and additional forward/backward passes), and (ii) describing practical approximations for large Jacobians. We emphasize that the expressions $(J_w \Sigma J_w^\top + \sigma^2 I)^{-1}$ and $\mathcal{I}$ can look expensive, but the core linear algebra lives in *prompt space*

(dimension $n$) and can be implemented via Jacobian–vector products (JVP/VJP), without materializing $J_w \in \mathbb{R}^{n \times p}$.

### H.4.1. BOOKKEEPING AND PRIMITIVE COSTS

We use the following dimensions (consistent with the main text):

- $p$: number of model parameters.

- $n$: number of prompt positions / residual terms (Sec. 3).

- $d$: dimension of the update subspace $\mathcal{U} = \mathrm{range}(A)$ (Assumption 3.1).

- $\mathcal{B}_{\mathrm{all}}$: partition of parameters into blocks (e.g., by layer/head/matrix type) used in Sec. 8.

To keep the discussion model-agnostic, we express runtime in terms of primitive passes:

- $C_{\mathrm{fwd}}$: cost of one forward pass on the prompt objective (producing the supervision outputs $s_w(P_n) \in \mathbb{R}^n$ and the scalar loss).

- $C_{\mathrm{bwd}}(\mathcal{S})$: cost of one backward pass producing gradients restricted to a parameter set $\mathcal{S}$ (e.g., the updated parameters, or a single block $b$). In practice, $C_{\mathrm{bwd}}(\mathcal{S})$ is typically of the same order as a full backward pass unless specialized kernels or partial-backprop are used.

Throughout, the *memory footprint* at test time is dominated by storing activations needed for a backward pass. Importantly, memory does *not* scale with the number of adaptation steps $T$ when updates are performed sequentially (activations are freed after each step), but time does scale with $T$.

### H.4.2. BASELINE: NO-UPDATE INFERENCE VS. STANDARD TTT

Let $T$ be the number of test-time gradient steps on the prompt loss. A coarse but useful accounting is:

- **No-update inference (ICL / standard inference):** one forward pass on the prompt+query, and no backward pass.

- **Standard TTT (few-step adaptation):** each step requires (at least) one forward pass to form the prompt loss and one backward pass to compute gradients, followed by a final forward pass for the query prediction. Thus,

$$\text{TTT runtime} \approx T \left( C_{\mathrm{fwd}} + C_{\mathrm{bwd}}(\mathcal{U}) \right) + C_{\mathrm{fwd,query}}.$$

This matches common lightweight TTT baselines (e.g., updating only a small parameter subset / LoRA state) where the dominant cost is the backward pass, not the parameter update itself.

### H.4.3. CONSTRUCTING THE PROMPT GEOMETRY $K$ WITHOUT FORMING $J_w$

Many quantities in Sections 4–7 depend on $K = \Phi\Phi^\top = J_w \Pi_{\mathcal{U}} J_w^\top \in \mathbb{R}^{n \times n}$. Crucially, $K$ is an $n \times n$ matrix: when $n$ is modest (few-shot prompts or a small set of tokens), forming and factorizing $K$ is cheap, and the real bottleneck is accessing Jacobian information.

**Explicit feature construction ($\Phi$).** Recall $\Phi \in \mathbb{R}^{n \times d}$ stacks the Jacobian features $\phi(x_i; P_n^{(-i)})^\top = \nabla_\theta s_i(w(\theta))|_{\theta=0}$. If one explicitly builds $\Phi$, the simplest approach is $n$ reverse-mode calls (one per $s_i$) restricted to the updated parameters:

$$\text{cost to build } \Phi \approx n \cdot C_{\mathrm{bwd}}(\mathcal{U}),$$

$$\text{memory} \approx \text{one backward-pass footprint.}$$

Once $\Phi$ is available, $K = \Phi\Phi^\top$ costs $O(n^2 d)$ flops and stores $O(n^2)$ numbers.

**Implicit matrix–vector products (recommended for large $p$ and/or larger $n$).** Many downstream procedures do *not* require $\Phi$ itself, only products of the form $Kv$ or $(K + \lambda I)^{-1}v$. These can be computed without forming $J_w$:

$$Kv = J_w \Pi_{\mathcal{U}} J_w^\top v. \tag{44}$$

A single $Kv$ can be computed with one VJP and one JVP (or two reverse-mode passes via Pearlmutter's trick):

1. compute $g := J_w^\top v$      (one VJP / backward pass),

2. project $g_{\mathcal{U}} := \Pi_{\mathcal{U}} g$      (cheap: masking or low-rank projection),

3. compute $Kv := J_w g_{\mathcal{U}}$    (one JVP / forward-mode, or equivalent).

Thus,

$$\text{cost of one } Kv \approx C_{\mathrm{bwd}}(\text{params}) + C_{\mathrm{jvp}}(\text{params}),$$

where $C_{\mathrm{jvp}}$ depends on the autodiff implementation. This is the standard way to scale kernel methods and Gauss–Newton/Fisher computations in large models.

**Solves with $K$ for larger $n$.** When $n$ is small, one can form $K$ explicitly and use a Cholesky factorization at $O(n^3)$ time. When $n$ is larger, one can solve linear systems with conjugate gradients (CG) using the matvec in (44). Each CG iteration costs one $Kv$ (plus $O(n)$ vector ops), and the number of iterations depends on the conditioning of $K + \lambda I$.

### H.4.4. COST OF PAC-BAYES HORIZON SELECTION (SEC. 7)

Horizon selection evaluates (for $T \in \mathcal{T}$) the evidence score

$$\ell_T(r) = \frac{1}{2n}\left\{\log\det(\Sigma_T) + r^\top \Sigma_T^{-1} r\right\},$$

where $\Sigma_T = \sigma^2(I - \rho K)^{-T}$. Given an eigendecomposition $K = U\mathrm{diag}(\lambda_i)U^\top$, we have

$$\log\det(\Sigma_T) = n\log\sigma^2 - \sum_{i=1}^{n} T\log(1-\rho\lambda_i),$$

$$r^\top \Sigma_T^{-1} r = \frac{1}{\sigma^2}\sum_{i=1}^{n}(1-\rho\lambda_i)^T(u_i^\top r)^2,$$

so evaluating $\ell_T$ for *all* $T \in \mathcal{T}$ costs $O(|\mathcal{T}|\,n)$ once $\{\lambda_i, u_i^\top r\}$ are known. Therefore, the total overhead of evidence-based selection is:

$$\underbrace{\text{cost to obtain } K}_{\substack{\text{dominant (Jacobian access)}}} + \underbrace{O(n^3)}_{\substack{\text{eig/Cholesky (cheap if } n \text{ is small)}}} + \underbrace{O(|\mathcal{T}|\,n)}_{\text{negligible}}.$$

In typical few-shot regimes (small $n$), the extra prompt-space linear algebra is negligible relative to even a single backward pass.

**Net compute can decrease.** A practical point is that horizon selection can *reduce* total test-time compute by choosing small $T$ (or $T = 0$) when adaptation is predicted to be unhelpful, potentially offsetting the cost of estimating $K$.

### H.4.5. COST OF BAYES-OPTIMAL SUBSPACE DESIGN (SEC. 8)

Thm. 8.2 involves the information-capture matrix

$$\mathcal{I} = \Sigma J_w^\top (J_w \Sigma J_w^\top + \sigma^2 I_n)^{-1} J_w \Sigma \in \mathbb{R}^{p \times p},$$

whose *full* formation is infeasible for modern $p$. In practice, we advocate two implementation regimes.

**(A) Offline global design.** In many deployments, the update mechanism (which blocks/heads are trainable at test time) is fixed *ahead of time*. One can estimate a *global* design from a development set of prompts (Cor. E.23) and then keep the selected subspace fixed at test time. In this regime, **there is no additional test-time cost** beyond the chosen TTT updates.

**(B) Prompt-adaptive scoring via prompt-space formulas.** When restricting to block on/off selection and (optionally) an isotropic correction prior $\Sigma = I_p$, Sec. 8.2 shows that the block score admits a purely prompt-space form:

$$s_b := \mathrm{tr}(\mathcal{I}_{bb}) = \mathrm{tr}\left(K_{\text{full}}^{(b)}(K_{\text{full}} + \sigma^2 I_n)^{-1}\right), \quad (45)$$

where $K_{\text{full}} = \sum_b K_{\text{full}}^{(b)}$ and $K_{\text{full}}^{(b)} = J^{(b)}J^{(b)\top}$. Here the only matrix inverse is $(K_{\text{full}} + \sigma^2 I_n)^{-1}$ in $\mathbb{R}^{n \times n}$. Thus, the *linear algebra* is again in prompt space.

The remaining challenge is to obtain $K_{\text{full}}^{(b)}$ (or its action on vectors) without forming $J^{(b)}$. This can be done with implicit Jacobian products restricted to block $b$:

$$K_{\text{full}}^{(b)} v = J^{(b)} J^{(b)\top} v$$

via one VJP and one JVP restricted to block $b$.

**Hutchinson estimator for large block sets.** If scoring every block exactly is expensive, one can approximate (45) with $R$ Hutchinson probes $z_r \sim \{\pm 1\}^n$:

$$s_b = \mathbb{E}_z\left[z^\top K_{\text{full}}^{(b)}(K_{\text{full}} + \sigma^2 I)^{-1} z\right] \approx \frac{1}{R}\sum_{r=1}^{R} z_r^\top K_{\text{full}}^{(b)} u_r,$$

where $u_r := (K_{\text{full}} + \sigma^2 I)^{-1} z_r$. When $n$ is small, each $u_r$ can be computed by a direct $n \times n$ solve; when $n$ is large, CG with the matvec $K_{\text{full}} v$ applies. The expensive part per probe is applying $K_{\text{full}}^{(b)}$ to $u_r$, which can be done implicitly without storing gradients.

**Top-eigenspace computation without forming $\mathcal{I}$.** If one truly requires the top-$d$ eigenspace of $\mathcal{I}$ (e.g., within a block for low-rank adaptation), one can use randomized SVD/Lanczos with only $\mathcal{I}$–vector products. A matvec $\mathcal{I}v$ can be implemented as:

$$\mathcal{I}v = \Sigma J_w^\top \underbrace{(J_w \Sigma J_w^\top + \sigma^2 I_n)^{-1}}_{n \times n \text{ solve}} J_w \Sigma v,$$

which requires (i) one JVP to compute $J_w \Sigma v$, (ii) one prompt-space solve, and (iii) one VJP to apply $J_w^\top$, plus multiplications by $\Sigma$ (cheap if $\Sigma$ is diagonal/structured). This yields a scalable route to approximate eigenspaces without explicit $p \times p$ matrices.

### H.4.6. FORWARD/BACKWARD PASS ACCOUNTING (SUMMARY)

Table 2 summarizes the dominant pass counts at test time. Here $n_J$ denotes the number of additional Jacobian evaluations needed to build or approximate $K$ (typically $n_J = n$ for explicit $\Phi$, and $n_J = O(R)$ for randomized trace/eigensolvers).

### H.4.7. PRACTICAL GUIDANCE AND REGIMES OF TRACTABILITY

**When is the matrix inverse cheap?** The inverse that appears in our design objectives, $(J_w \Sigma J_w^\top + \sigma^2 I_n)^{-1}$, is an $n \times n$ matrix. Thus it is cheap whenever $n$ is modest. If $n$ is large (e.g., token-level supervision on long prompts), one should not form this inverse explicitly; instead use CG/Lanczos with implicit matvecs.

| Method | # forward passes | # backward/Jacobian passes | Remarks |
|---|---|---|---|
| No-update inference (ICL) | 1 | 0 | baseline inference only |
| Standard TTT ($T$ steps) | $T+1$ | $T$ | one prompt step = fwd+bwd |
| TTT + horizon selection (ours) | $T+1$ | $T+n_J$ | $n_J$ to estimate $K$ / spectrum |
| TTT + offline subspace design (ours) | $T+1$ | $T$ | no extra test-time overhead |

*Table 2.* Test-time pass counts (coarse). Horizon selection adds Jacobian access to estimate $K$, while offline subspace design adds no test-time cost.

**Where is the true bottleneck?** The dominant cost is accessing Jacobian information, not the $n \times n$ linear algebra. This is why we emphasize (i) restricting test-time updates to a small parameter subset, and (ii) using implicit Jacobian products instead of explicit Jacobian materialization.

**Amortization via offline design.** The subspace-design step can be performed offline once (global design) and reused at test time, eliminating any additional inference-time overhead while still benefiting from a principled update mechanism.

**Compute–robustness tradeoff.** Our framework explicitly trades compute for robustness: step selection can avoid harmful over-adaptation (and may reduce $T$), while subspace design can restrict updates to identifiable, redundancy-aware directions (reducing the effective dimension of adaptation). Both tradeoffs are consistent with the empirical observation that "more test-time optimization" can be worse when the prompt signal is weak or misspecified.

# I. Experimental Details

## I.1. Task and Data Generation

We evaluate our framework using `distilgpt2` on a synthetic integer regression task.

We use the pretrained GPT-2 family model and, in particular, the `distilgpt2` checkpoint released via the Hugging Face Transformers library (Radford et al., 2019; Wolf et al., 2020; Sanh et al., 2019).

**Task Distribution.** Each task is defined by a hidden shift $s \in \{0, \ldots, 9\}$. Data pairs $(x, y)$ are generated such that the clean label is $y = (x + s) \pmod{10}$. At test time, we sample $n = 10$ input digits and one query digit from $0, \ldots, 9$.

**Noise and Templates.** We introduce label noise by replacing the target $y$ with a uniformly random digit sampled from $\{0, \ldots, 9\}$ with probability $p_{\text{noise}}$. We simulate two regimes: a *clean* regime where $p_{\text{noise}} = 0.0$, and a *noisy* regime where $p_{\text{noise}} = 0.55$. These values create a bimodal distribution over prompt quality, allowing us to test adaptive $T$ selection under varying signal-to-noise ratios.

The model sees the data formatted via two templates, selected randomly per task: `"x->y"` (Template 0) or `"x:y"` (Template 1).

## I.2. Model and TTT Optimization

**Architecture.** We use the pretrained `distilgpt2` model (6 layers, 12 heads, 768 embedding dim).

**Update Subspace.** We restrict optimization to the attention **Value** projection weights. For a selected block $(l, h)$ (layer $l$, head $h$), we construct a binary mask $M_{l,h}$ targeting the indices corresponding to the Value matrix in the fused `c_attn` linear layer (indices $2\,d_{\text{model}}$ to $3\,d_{\text{model}}$). This corresponds to the $W_V^{(h)}$ parameters in the notation of Section 3.3.

**Optimization Objective.** We map the digit label $y \in \{0, \ldots, 9\}$ to $\tilde{y} := y/9 \in [0, 1]$ and predict $\widehat{y}_{w(\theta)}(x; P_n) := \frac{1}{9} \sum_{j=0}^{9} j\, p_d$, where $p_d$ are the softmax probabilities of the digit tokens. For each prompt position $i$, we use leave-one-out supervision and define the residual $r_i(\theta) := \tilde{y}_i - \widehat{y}_{w(\theta)}(x_i; P_n^{(-i)})$. The prompt loss is $L(\theta) = \frac{1}{2\sigma^2} \sum_{i=1}^{n} r_i(\theta)^2$. In the few-step regime, linearizing $\widehat{y}_{w(\theta)}$ at $\theta = 0$ yields $r(\theta) \approx r - \Phi\theta$, recovering the quadratic surrogate $L(\theta) = \frac{1}{2\sigma^2} \|r - \Phi\theta\|_2^2$ as in Section 3.2.

Updates are performed via the stochastic gradient descent on the leave-one-out prompt residuals for $T$ steps with step size $\eta = \rho\sigma^2$ (equivalently, scaled step size $\rho$).

## I.3. Evidence-Based $T$ Selection

We select the horizon $T$ by minimizing the negative log-evidence of the prompt residuals $r$. We assume the residuals follow a zero-mean Gaussian induced by the TTT filter: $r \sim \mathcal{N}(0, \Sigma_T)$.

Using the eigen-decomposition of the kernel $K = U\Lambda U^\top$ with eigenvalues $\lambda_i$, the covariance is $\Sigma_T = \sigma^2(I - \rho K)^{-T}$. We implement two selection criteria:

**1. Fixed-$\sigma$ Evidence.** We estimate a constant noise variance $\hat{\sigma}^2 = \frac{1}{n}\|r\|^2$ from the unadapted residuals. We then select

$T$ to minimize:

$$\mathcal{L}(T) = -\frac{T}{n}\sum_{i=1}^{n}\log(1-\rho\lambda_i) + \frac{1}{n\hat{\sigma}^2}\sum_{i=1}^{n}(1-\rho\lambda_i)^T(\tilde{r}_i)^2,$$

where $\tilde{r} = U^\top r$ are the rotated residuals. This corresponds to the fixed-$\sigma^2$ variant discussed in Section 7.

**2. MLE-$\sigma$ Evidence.** We treat $\sigma^2$ as a free parameter to be profiled out (Murphy & van der Vaart, 2000). We minimize the profile log-likelihood: $\mathcal{L}_{\mathrm{MLE}}(T) = \log\left(\frac{1}{n}\sum_{i=1}^{n}(1-\rho\lambda_i)^T(\tilde{r}_i)^2\right) - \frac{T}{n}\sum_{i=1}^{n}\log(1-\rho\lambda_i)$. Both methods select $T_{\mathrm{MAP}} \in \arg\min_{T\in\mathcal{T}}\mathcal{L}(T)$ where $\mathcal{T} = \{0,1,2,\ldots,30\}$.

### I.4. Subspace Selection Algorithms

In Experiment 2, we select a subset of heads from the last 4 layers (48 candidate blocks). We vary $k_{\mathrm{budget}} \in \{1,2,4,8,16,32\}$ to assess scaling behavior.

**Trace-TopK.** We compute the trace score for each block $b$ as $\mathrm{Tr}(K^{(b)}(K_{\mathrm{sum}}+\sigma^2 I)^{-1})$, where $K^{(b)}$ is the kernel for block $b$ and $K_{\mathrm{sum}}$ is the sum of all candidate kernels.

**Query-Aware.** Consistent with Theorem 8.2, which identifies the subspace maximizing posterior variance reduction, we prioritize update directions with significant query-aligned gradients. In practice, this corresponds to iteratively selecting the block $b$ that maximizes the squared magnitude of the predicted correction $h^2$, where $h = k_x^\top Q_T(K_{\mathrm{curr}} + K^{(b)})r$. Here, $k_x$ is the prompt-query coupling vector, $K_{\mathrm{curr}}$ is the current kernel sum, and $Q_T(\lambda) = \frac{1-(1-\rho\lambda)^T}{\lambda}$ is the TTT filter response.

**Trace-BottomK.** We use the same trace score as TRACE-TOPK, but select the $k_{\mathrm{budget}}$ blocks with the smallest scores $s_b$. We include TRACE-BOTTOMK as a negative-control baseline for query-agnostic selection.

**Random (Uniform).** As a baseline, we sample $k_{\mathrm{budget}}$ blocks uniformly at random without replacement from the candidate set (e.g., the 48 (layer, head) blocks), and update only the selected blocks.

### I.5. Hyperparameters

We selected the prior scale $c$ via a pilot study on 20 tasks distinct from the test set, sweeping $c \in \{0.05, 0.1, 0.2\}$ with fixed $T = 8$. For the main experiments, we used a step size scale $\rho \approx c/\lambda_{\max}$ to ensure stability ($\rho < 1/\lambda_{\max}(K)$ as required by the monotone-filter regime in Section 4).

**Evaluation Protocol.** We evaluate all methods on 2,000 independent test tasks for Experiment 1, and 500 tasks for Experiment 2 (due to the additional computational cost of multiple block configurations).

## J. Supplementary Real-Data Experiment on SST-2

We include a small real-data experiment on SST-2 to check whether the two design principles from the main text also appear outside the synthetic digit-shift task. This experiment is intended as supplementary mechanism-level evidence, not as a definitive benchmark.

**Dataset and task construction.** SST-2 is a binary sentence-level sentiment classification dataset (Socher et al., 2013). We sample tasks from the validation split. Each task consists of 10 labeled prompt sentences and one held-out query sentence, sampled without replacement within the task. The query label is used only for evaluation. We encode the negative and positive labels as 0 and 1, respectively, and use a fixed text-label template: each prompt example is written as `Text: {sentence} Label: {0/1}`, followed by the query prefix `Text: {query} Label:`. To keep the setting close to the squared-loss theory, we cast the binary classification problem as scalar regression on labels in $\{0, 1\}$ and evaluate query mean-squared error.

**Prediction model.** We use `distilgpt2` as a causal language model. The prediction at the query is computed from the next-token distribution restricted to the two single-token label strings for 0 and 1: if $p_0, p_1$ are the restricted softmax probabilities, the scalar prediction is $\hat{y} = p_1$ (equivalently, the expected normalized label). For computational consistency across tasks, input texts are truncated before constructing the prompt, and all model weights are reset to the pretrained checkpoint before each TTT run.

**TTT protocol.** We use the same few-step TTT protocol as in the main experiments. Prompt residuals are computed in a leave-one-out manner to avoid label leakage: when forming the residual for a prompt example, its own label is not included in the prompt used by the base predictor. The prompt loss is the squared error between these residuals and the linearized prediction shifts. Adaptation updates only attention Value weights, selected at the attention-head level; for GPT-2's fused `c_attn` parameter, this corresponds to the Value slice from $2d_{\mathrm{model}}$ to $3d_{\mathrm{model}}$. Gradients outside the selected Value-head slices are masked to zero. We use 50 sampled tasks in this supplementary run.

**Kernel and step-size construction.** For each task, we compute the prompt kernel and prompt–query coupling from Jacobian features with respect to the candidate Value-head parameters. The effective noise level is estimated by the plug-in value $\hat{\sigma}^2 = \|r\|_2^2/n$ from the base leave-one-out residuals. The scaled step size is chosen by the same stable normalization as in the synthetic experiment,

$\rho = c/\lambda_{\max}(K)$ with $c = 0.1$ and clipped so that $\rho < 1/\lambda_{\max}(K)$.

**Horizon selection.** For the update horizon experiment, we compare a fixed-horizon baseline with $T = 8$ against evidence-based per-prompt selection. The evidence methods search over the same grid as in the synthetic experiment, $T \in \{0, 1, \ldots, 30\}$, using either a fixed plug-in estimate of $\sigma$ or the profiled MLE-$\sigma$ variant. The evidence score is evaluated from the prompt-kernel eigenspectrum and the rotated residuals, so the candidate horizons are not evaluated by rerunning TTT from scratch.

**Subspace selection.** For the block/head selection experiment, we consider 48 candidate attention heads from the last four layers of `distilgpt2`. We evaluate budgets $k_{\text{budget}} \in \{1, 2, 4, 8, 16, 32\}$ and compare QUERY-AWARE, TRACE-TOPK, TRACE-BOTTOMK, and Random selection. QUERY-AWARE uses the prompt–query coupling to greedily select heads under the fixed horizon $T = 8$; TRACE-TOPK ranks heads by the query-agnostic trace score; TRACE-BOTTOMK is the corresponding negative-control baseline. The Random baseline is averaged over five independent random head selections for each task and budget. As in the main text, subspace results are reported as query-MSE improvement relative to Random, so positive values indicate lower query error than Random.

| Method | Query MSE |
|---|---|
| Fixed-$T$ TTT | 0.328 |
| Evidence, fixed-$\sigma$ | 0.325 |
| Evidence, MLE-$\sigma$ | 0.324 |

*Table 3.* **SST-2 horizon selection.** Evidence-based horizon selection slightly improves over the fixed-$T$ baseline in this small real-data study.

| Budget | QUERY-AWARE | TRACE-TOPK | TRACE-BOTTOMK |
|---|---|---|---|
| 1 | +0.0010 | +0.0005 | −0.0001 |
| 2 | +0.0019 | +0.0011 | −0.0004 |
| 4 | +0.0031 | +0.0017 | −0.0007 |
| 8 | +0.0039 | +0.0031 | −0.0022 |
| 16 | +0.0059 | +0.0060 | −0.0030 |
| 32 | +0.0029 | +0.0029 | −0.0047 |

*Table 4.* **SST-2 subspace selection.** Mean query-MSE improvement relative to Random selection; positive values are better. QUERY-AWARE improves over Random across budgets and is stronger than TRACE-TOPK for small-to-moderate budgets, while TRACE-BOTTOMK consistently hurts.

Overall, the results show the same qualitative pattern as the synthetic experiment: evidence-based selection gives a modest improvement over a fixed horizon, and query-aware update directions tend to improve query prediction more reliably than query-agnostic negative controls. Because

this study uses only 50 tasks and a small distilled language model, we treat it as preliminary support for the proposed mechanisms rather than evidence of broad practical superiority.

