# OpenReview forum: "A Decision-Theoretic View of Test-Time Training: When, How Far, and Which Directions to Adapt"
_ICML.cc/2026/Conference — ICML 2026 regular_

### Official Review · Reviewer_a3Vk · 2026-03-12

**Soundness:** 4
**Presentation:** 2
**Significance:** 3
**Originality:** 3
**Overall Recommendation:** 5
**Confidence:** 4

**Summary:**

The paper proposes an analyzes of test-time training (TTT) through the lens of Bayesian inference. This new perspective sheds light on the limitations of current TTT frameworks, showing that fixed hyperparameters cannot guarantee good performance across different prompts. The paper further proposes principled methods to select the number of optimization steps and which parameters to update for each given prompt and query.

**Compliance With Llm Reviewing Policy:**

Affirmed.

**Final Justification:**

The paper presents a novel and insightful lens for analyzing TTT, with strong conceptual originality and potential significance. My initial reservations concerned the experimental evaluation, which limited my assessment of soundness; these concerns have been satisfactorily addressed through the authors’ rebuttal and additional results. While the core ideas are compelling and technically sound, the presentation would benefit from improved flow. Overall, despite this weakness, I believe the paper merits acceptance.

**Key Questions For Authors:**

1. The proposed techniques to find optimal number of optimization steps and update directions are to my mind the most interesting part of the paper, but their usefulness hinge on how costly it would be to implement them in very large models at test time. Could the authors comment on the scalability and easy-of-use of these methods?
2. In Table 1, we see that evidence-based selection leads to larger number of optimization steps $T$ than the baseline. Is that a fair comparison? Could one not argue that the improvement is simply due to larger values of $T$?

**Limitations:**

yes

**Strengths And Weaknesses:**

### Strengths
- Test-time training (TTT) is a relevant and promising topic, and the paper addresses directly one of its main limitations, which is the its sensitivity to hyperparameters.
- The analysis of TTT through the lens of Bayesian inference is well motivated and developed. The main insights it provides on how to choose the number of optimization steps and which parameters to update are also interesting and potentially very useful.

### Weaknesses
- Unless I am missing something, some of the empirical results are underwhelming. Are the results in Table 1 even statistically significant?
- I did not take this is consideration in my score because this is somewhat subjective, but the paper is not a very pleasant read. It is entirely subdivided into small sections, so that it reads as a collection of notes rather than a cohesive text.

---

> ### Author Rebuttal · Authors · 2026-03-31
>
> We thank you for your positive feedback on our perspective and its practical implications.
>
> **Question 1. scalability of the methods**
>
> We appreciate this question, as it raises an important practical point.
>
> **Step selection.**
>
> Our step-selection method is relatively lightweight once the prompt-kernel information is available. As described in App. I.3, for each candidate horizon $T \in \\{0,\dots,30\\}$, we evaluate a closed-form evidence score based on the prompt-kernel eigenspectrum and the rotated residuals. Thus, we only sweep over a small set of candidate $T$ values, rather than rerunning TTT from scratch for every candidate. In practice, the main overhead is computing the Jacobian information once per prompt; after that, selecting $T$ is cheap.
>
> **Subspace selection.**
> We do not intend exhaustive per-parameter search over the full model. Instead, we focus on the more practical block/head-level TTT setting: Sec. 8.2 restricts the update projector to a Transformer block partition, and Exp. 2 uses a modest candidate pool of 48 attention heads from the last 4 layers under a fixed budget. In addition, as discussed in Sec. 8, replacing $Q = j_x j_x^\top$ by the query-averaged $Q = \mathbb{E}_X[j_X j_X^\top]$ yields a global update mechanism (line 355, right column), and a population-level mechanism can be obtained by averaging $I_Q$ across prompts  (line 359, left column). This suggests an efficient amortized variant: the global averaged design can be computed once and then reused across queries, rather than recomputed from scratch for each test example.
>
> We will clarify these computational points in the revised manuscript.
>
> **Question 2. fairness of comparing to $T=8$**
>
> We acknowledge that this point was not made explicit: **T=8 is the best fixed horizon over the same search grid $T\in\\{0,\dots,30\\}$**, and hence, this is a fair comparison.
> So the gain of evidence-based selection is not due to comparing against a weak fixed-$T$ baseline. More importantly, our claim is not that “larger $T$ is better,” but that the best horizon is prompt-dependent. Figure 1(a) shows substantial variation in the selected $T_{\mathrm{MAP}}$, and the selected $T$ shifts with prompt signal-to-noise ratio: it is typically smaller in noisy prompts and larger in clean prompts. This is also consistent with our theory, which emphasizes that a fixed $T$ can be Bayes-suboptimal (Prop. 6.1) and that over-adapting can hurt (Prop. 6.2).
>
>
> **Weakness 1. underwhelming empirical results.**
>
> > some of the empirical results are underwhelming.
>
> We added a preliminary SST-2 experiment with distilgpt2 in the same few-step TTT setting as in the simulation experiment. Although this study is limited in scale and we therefore view it as supplementary mechanism-level evidence rather than a definitive benchmark, it shows the same qualitative trends: evidence-based horizon selection slightly improves over fixed $T$, and query-aware subspace selection is stronger than query-agnostic alternatives across a broad range of budgets. We provide details in our response to Reviewer EiDU (Question 2).
>
> > Are the results in Table 1 even statistically significant?
>
> To address your concern, we performed a paired analysis on the same 2,000 tasks. Specifically, for each task $i$, we computed the paired improvement $\Delta_i =$ “baseline method’s query MSE on task i” − “method’s query MSE on task i”
> so that a positive value means the method achieves lower query MSE than the baseline. This paired setup is important because it compares the two methods on exactly the same tasks.
>
> We report the 95% bootstrap confidence interval for the mean paired improvement and the paired permutation p-values below.
>
> | method | 95% bootstrap CI | Paired permutation p-value |
> |---|---:|---:|
> | TTT ($T=8$) vs ICL | [0.000648, 0.001796] | 0.00005 |
> | TTT (evidence, fixed-$\sigma$) vs  TTT ($T=8$) | [0.000545, 0.002894] | 0.00325 |
> | TTT (evidence, MLE-$\sigma$) vs  TTT ($T=8$) | [0.000605, 0.003079] | 0.00250 |
>
> In all three comparisons, the 95% bootstrap CI excludes 0, and the paired permutation tests are significant. Thus, although the effect sizes are modest, the gains are statistically significant under a task-matched paired analysis: best fixed-$T$ TTT significantly improves over ICL, and both evidence-based $T$-selection variants significantly improve over the best fixed-$T$ baseline.
>
>
> **Weakness 2. presentation.**
>
> We appreciate your constructive feedback regarding the paper's structure. In the revision, we will merge some of the very short subsections, expand the roadmap at the beginning of each section, and improve paragraph transitions to ensure the paper reads as a more cohesive narrative.
>
> ---
>
> We appreciate these helpful questions and comments, and we would be happy to elaborate on any of these points during the discussion period.

---

> > ### Author Rebuttal · Reviewer_a3Vk · 2026-04-03
> >
> > I thank the authors for addressing my concerns, which are now solved. I am happy to increase my score provided the authors improve the readability of the final version as promised and include the new results.

---

> > > ### Author Response · Authors · 2026-04-06
> > >
> > > We thank you for the follow-up and updated assessment. We will ensure that the final version improves readability and includes the new results.

---

### Official Review · Reviewer_EiDU · 2026-03-13

**Soundness:** 3
**Presentation:** 3
**Significance:** 3
**Originality:** 3
**Overall Recommendation:** 4
**Confidence:** 2

**Summary:**

TTT is known to suffer from extreme hyperparameter sensitivity. This manuscript models TTT as implicit Bayesian inference within a local linear-quadratic kernel regime and gives three key results. In their Gaussian process setting, the authors show that TTT reduces prediction error only when spectral updates match the prompt's signal-to-noise ratio and geometrically align with query-relevant eigen-directions. They further derive a PAC-Bayes bound for selecting update steps via prompt evidence to avoid overfitting. Moreover, they characterize the Bayes-optimal update subspace under a linear-Gaussian correction prior, yielding query-aware scoring rule to select Transformer attention blocks and heads for adaptation.

**Compliance With Llm Reviewing Policy:**

Affirmed.

**Key Questions For Authors:**

- In Section 8.2, the query-aware block scoring rule requires $j_x$, the Jacobian evaluated at test input $x$, which requires an extra backward pass for every single test query before the actual TTT adaptation steps can even start. This effectively doubles the backward-pass latency at inference time. How do the authors propose mitigating this without falling back on query-agnostic heuristics?

- The evaluation is confined to a synthetic digit-shift task. Can the authors provide supplemental results for an established reasoning benchmark (e.g., ARC ) or vision adaptation benchmark to empirically show the superiority of their method?

**Limitations:**

yes

**Strengths And Weaknesses:**

- By formalizing TTT as prompt-induced Bayesian inference, the paper makes a significant contribution to the understanding of inference-time optimization.

- The theoretical foundation is very solid. The primary weakness is empirical validation. The claims are validated exclusively on a simplified, synthetic digit-shift task using a distilled language model. Robust validation on benchmarks, such as ARC and BIG-Bench Hard, or via tabular foundation models like TabPFN should be included.

- Another weakness of the query-aware scoring rule is the expensive backward pass that is required to compute the Jacobian (see questions.)

---

> ### Author Rebuttal · Authors · 2026-03-31
>
> We appreciate your positive assessment of the paper’s theory and its contribution.
>
>
> **Question 1. on the extra backward pass**
>
> We agree that the exact query-specific instantiation $Q = j_x j_x^\top$ requires query-specific sensitivity information, so there is no free exact query-aware score. That said, two clarifications are important.
>
> First, the extra cost is **additive rather than per-block**: one backward pass on the query output yields $j_x$ for all blocks, after which the block restrictions $j_x^{(b)}$ are obtained by slicing. Relative to standard $T$-step TTT, this is roughly a **$+1$ backward-pass overhead**, rather than a factor-of-two except in the degenerate $T=1$ case.
>
> Second, Sec. 8 already provides a **principled amortized alternative**: replacing
> $Q = j_x j_x^\top$ by the query-averaged $Q = \mathbb{E}_X[j_X j_X^\top]$ yields a global update mechanism (line 355, right column), and a population-level mechanism can be obtained by averaging $I_Q$ across prompts  (line 359, left column). This removes any per-query backward-pass overhead and is not an ad hoc heuristic; it is the Bayes-optimal solution for the averaged query objective.
>
> We will revise Sec. 8.2 to make the distinction between (a) the ideal query-specific score and (b) the efficient amortized global design explicit.
>
> **Question 2. on empirical validation.**
>
> To partially address the concern about validation beyond the synthetic digit-shift task, we added a real-data experiment on SST-2, a binary sentence-level sentiment benchmark [1]. We used distilgpt2 in the same few-step TTT setting as in the main text with 10 prompt examples and 1 held-out query per task. This supplementary study currently covers 50 tasks.
>
> - For $T$ selection (like Table 1 in our paper), we used the same search grid $T \in \\{0, …, 30\\}$ with fixed-$T$ baseline $T = 8$.
>
>
> - For subspace selection (like Figure 1(b) in our paper), we considered 48 candidate attention heads from the last 4 layers and budgets $\\{1,2,4,8,16,32\\}$; the Random baseline was averaged over multiple random selections.
>
> We note that we view it as supplementary mechanism-level evidence rather than a definitive benchmark.
>
> | SST-2: query MSE  | value |
> |---|---:|
> | fixed $T=8$ | 0.328 |
> | evidence-based $T$ (fixed-$\sigma$) | 0.325 |
> | evidence-based $T$ (MLE-$\sigma$) | 0.324 |
>
> | SST-2: $\Delta$ query MSE vs. Random  | QUERY-AWARE | TRACE-TOPK | TRACE-BOTTOMK |
> |---|---:|---:|---:|
> | budget 1                                   |     +0.0010 |    +0.0005 |       -0.0001 |
> | budget 2                                   |     +0.0019 |    +0.0011 |       -0.0004 |
> | budget 4                                   |     +0.0031 |    +0.0017 |       -0.0007 |
> | budget 8                                   |     +0.0039 |    +0.0031 |       -0.0022 |
> | budget 16                                  |     +0.0059 |    +0.0060 |       -0.0030 |
> | budget 32                                  |     +0.0029 |    +0.0029 |       -0.0047 |
>
> On this study, we observe the same qualitative trends as in our synthetic setting.
>
> - The first table shows that evidence-based horizon selection slightly improves query MSE relative to the fixed-$T$ baseline (0.325/0.324 vs. 0.328).
>
> - The second table shows that QUERY-AWARE improves over Random across budgets and is generally stronger than TRACE-TOPK for small-to-moderate budgets, while TRACE-BOTTOMK consistently hurts.
>
> We therefore view these results as preliminary real-data support for the paper’s two main design principles: selecting the adaptation horizon by evidence, and selecting update directions by query alignment.
>
> —
>
> Thank you for these constructive comments, and we would be very happy to address any follow-up questions that arise in the discussion period.
>
> ### References
>
> [1] Socher, et al. “Recursive Deep Models for Semantic Compositionality Over a Sentiment Treebank.” In Proceedings of the 2013 Conference on Empirical Methods in Natural Language Processing. 2013. https://nlp.stanford.edu/sentiment/index.html.

---

> > ### Author Rebuttal · Reviewer_EiDU · 2026-04-04
> >
> > The authors successfully address my theoretical and computational concern regarding the backward-pass latency. However, my primary concern regarding the empirical validation remains unresolved.The authors only provide a preliminary experiment on SST-2. This is a simplistic, saturated binary sentiment classification task, and their evaluation is restricted to only 50 tasks using a heavily distilled language model (distilgpt2). Contemporary TTT research validates on complex reasoning tasks like the ARC and BIG-Bench Hard, or utilizes robust tabular foundation models like TabPFN.
> >
> > While the paper's framework of Bayesian optimal experimental design remains highly original and sound, the empirical evidence provided does not sufficiently prove that the proposed query-aware selection rule yields meaningful performance improvements in real-world models under complex distribution shifts.
> >
> > Therefore, I will maintain my already positive rating. I thank the authors for their detailed response.

---

> > > ### Author Response · Authors · 2026-04-06
> > >
> > > Thank you for the thoughtful follow-up and for maintaining your positive rating.
> > >
> > > We appreciate this perspective and will clarify the empirical scope and limitations more explicitly in the revision. We will also make clear that this paper is intended as a theoretically grounded starting point for query-aware TTT design, with broader validation on more challenging benchmarks as an important next step.

---

### Official Review · Reviewer_5Cfk · 2026-03-13

**Soundness:** 3
**Presentation:** 3
**Significance:** 3
**Originality:** 3
**Overall Recommendation:** 4
**Confidence:** 4

**Summary:**

This paper aims to provide a decision-theoretic and Bayesian view of test-time training. The main idea is to reinterpret few-step TTT as learning a correction term from prompt residuals, and then analyze this process through Bayes risk decomposition, prompt-induced geometry, spectral filtering, and implicit Bayesian inference. Based on this framework, the paper further discusses when TTT can help, why the update horizon T and update subspace should be prompt-dependent, and how evidence-based step selection and query-aware block selection can be derived from the theory.

**Compliance With Llm Reviewing Policy:**

Affirmed.

**Key Questions For Authors:**

Please refer to weakness.

**Limitations:**

Mostly yes for technical limitations, but less so for broader deployment considerations. The paper does mention several theoretical limitations, though I would still encourage a clearer discussion of practical failure modes and possible risks of incorrect test-time adaptation.

**Strengths And Weaknesses:**

Overall, I think the paper offers an interesting and coherent theoretical perspective on TTT, although the current theory still remains somewhat idealized and the empirical validation is relatively limited.

Strength:
1. The paper offers a nice and fairly unified theoretical perspective on TTT. It brings together several different viewpoints into one coherent framework, which makes the paper conceptually richer than many works that only provide more isolated observations.
2. The paper identifies several useful mechanisms for thinking about when TTT may help. In particular, the distinction between fitting the prompt and improving the query is quite helpful.
3. The paper makes a reasonable effort to translate the theory into concrete design principles.

Weakness:
1. The paper offers a nice unifying perspective, but it is not always easy to tell which parts are genuinely new technical contributions and which parts are mainly reinterpretations of existing ideas. As a result, the novelty boundary with prior work still feels somewhat unclear.
2. There remains a noticeable gap between the theory and practical TTT. The analysis mainly relies on a local linear quadratic surrogate, whereas practical foundation-model TTT is usually based on token-level cross-entropy or self-supervised objectives. It is therefore unclear how directly the current theory transfers to realistic settings. It would be meaningful if the authors could discuss this gap more explicitly or provide some evidence on whether the framework can be extended beyond the squared-loss setting.
3. The justification for evidence-based step selection still feels somewhat incomplete. The paper gives a reasonable argument for selecting T from prompt evidence, and Theorem 7.2 provides a PAC-Bayes guarantee for out-of-sample negative log-evidence. However, the theory still seems more directly tied to prompt residual modeling than to downstream query prediction, so it remains somewhat unclear why the same criterion should be expected to select the best T for the final prediction objective.
4. All experiments are conducted on a synthetic digit-shift task with distilgpt2, and even within this toy setup, I still do not find the empirical evidence fully convincing. The gains are relatively small or somewhat marginal, and the baselines are limited. Right now, the experiments show that the method can work in this toy setting, but they do not yet establish strong practical significance.
5. The most interesting aspect of the paper, in my view, is the set of interpretable theoretical mechanisms it introduces. Although the experiments already provide some useful mechanism-level evidence, the empirical validation still feels partial, since the connection to the paper's main theoretical quantities is not yet made in a sufficiently direct quantitative way.

---

> ### Author Rebuttal · Authors · 2026-03-31
>
> We thank you for the careful reading and detailed evaluation. We address the main concerns below.
>
> **Weakness 1. novelty boundary**
>
> Thank you for this helpful comment. We agree that the current presentation may not sufficiently separate conceptual framing from technical novelty. The Bayesian interpretation of few-step TTT in the kernel regime is intended primarily as a unifying lens; taken in isolation, this part can indeed be read as a reinterpretation of known connections between spectral filtering / early stopping and Gaussian inference.
>
> Our main novelty is what this lens enables in the TTT setting. Specifically, the paper derives: (1) a query-level conditional Bayes-gap identity showing that TTT helps when the update filter matches the prompt SNR and the updated eigendirections align with the prompt-query coupling, together with formal failure results for fixed step counts and fixed update subspaces; (2) a PAC-Bayes guarantee for evidence-based per-prompt selection of the number of update steps; and (3) a characterization of the Bayes-optimal update subspace via the information-capture matrix, which leads to a practical block/head scoring rule for Transformers.
>
> We will clarify this point by revising the Introduction section and adding the following sentence to the Discussion section :
>
> > The implicit-Bayesian view is a conceptual lens rather than our sole technical contribution. Our technical novelty lies in the TTT-specific consequences of this lens: query-level Bayes-gap identities, PAC-Bayes step selection, and optimal update-subspace design.
>
>
>
> **Weakness 2. gap to practical TTT**
>
> We agree that the current theory is fundamentally local: few-step restricted updates, linearization, and a quadratic prompt surrogate. We already discuss this scope and provide a finite-step deviation bound from the nonlinearity in App. F.1. For non-squared losses, App. E.2 already gives the one-step analogue via the pseudo-residual $\tilde r = -\sigma^2 \nabla_s \ell$, and explains that extending the exact multi-step filter identities requires a local curvature approximation (e.g., Gauss-Newton/Hessian freezing). We will make this scope more explicit in the main text. Thank you for the helpful suggestion.
>
> **Weakness 3. why evidence for $T$ if the final objective is query prediction?**
>
> Thank you for the comment. We agree that our current presentation may have suggested a stronger claim than what is actually proved. We do not prove a single end-to-end theorem that evidence-optimal $T$ is automatically query-risk-optimal. Instead, the logic is indirect: Section 7 provides a principled criterion for choosing $T$ within the TTT-induced filter family, Thm. 5.2 shows that query prediction depends on that same filter via prompt–query coupling, and Section 8 separately handles the query-dependent choice of update directions. We will revise the paper to render this reasoning explicit and to avoid overstating the scope of Thm. 7.2.
>
>
> **Weakness 4. experimental scope**
>
> Thank you for raising these issues.
>
> > The gains are relatively small or somewhat marginal
>
> We believe this comment refers to the results in Table 1 (page 8). To address this issue, we confirmed that, on the same 2,000 tasks, paired bootstrap confidence intervals exclude $0$ and paired permutation p-values are significant in the comparisons. For further details, please refer to our response to Reviewer a3Vk's Weakness 1.
>
> > Right now, the experiments show that the method can work in this toy setting
>
> We added a preliminary SST-2 experiment with distilgpt2 in the same few-step TTT setting as in the simulation experiment. Although this study is limited in scale and we therefore view it as supplementary mechanism-level evidence rather than a definitive benchmark; it shows the same qualitative trends: evidence-based horizon selection slightly improves upon fixed $T$, and query-aware subspace selection is stronger than query-agnostic alternatives across a broad range of budgets.  For details, please refer to our response to Reviewer EiDU (Question 2).
>
>
> **Weakness 5. connection between theory and empirical evidence**
>
> We agree that this can be made more direct. Figure 1 already isolates two of the paper’s key quantities/mechanisms: prompt-dependent horizon selection and the distinction between prompt fit and query-aligned improvement. We will revise the text to connect these plots more explicitly to the filter-match / alignment quantities in Thm. 5.2 and Sec. 8.
>
> ---
>
> Thank you again for the careful review. If any aspect remains unclear, we would be happy to discuss it further during the discussion phase.

---

> > ### Author Rebuttal · Reviewer_5Cfk · 2026-04-04
> >
> > Thanks for the detailed responses, which address most of my concerns. I keep my positive score.
> >
> > Yet, the authors acknowledge the gap to practical TTT and the included new experiments remain somewhat toy.

---

> > > ### Author Response · Authors · 2026-04-06
> > >
> > > Thank you for the helpful follow-up and for keeping the positive score.
> > >
> > > We agree that the current paper should be positioned more clearly as a theoretical starting point for understanding principled TTT design, rather than as a comprehensive empirical study of practical large-scale TTT. We will make this positioning explicit in the revision and clarify the empirical scope and limitations more carefully.

---

### Official Review · Reviewer_v6Qe · 2026-03-21

**Soundness:** 4
**Presentation:** 2
**Significance:** 3
**Originality:** 4
**Overall Recommendation:** 4
**Confidence:** 1

**Summary:**

This paper explores the theoretical understanding on when and how TTT improves predictions in target domain, specifically by framing TTT as a Bayesian inference in a kernel regime. It shows that TTT succeeds when its updates align with prompt’s signal-to-noise and query-relevant eigen-directions, which gives insights on how the adaptation hyperparameters, such as update steps interact with it.

**Compliance With Llm Reviewing Policy:**

Affirmed.

**Key Questions For Authors:**

See weakness.

**Limitations:**

No.

**Strengths And Weaknesses:**

**Strength**:
- It provides rigorous mathematical framework that explains TTT in terms of Bayesian inference, which allows theoretical explanations regarding empirical behaviors such as overfitting.
- Based on theoretical analysis, this paper proposes empirical setups that optimizes the hyperparameters for TTT, such as update steps.

**Weakness**:
- It is not easy to follow the paper, which might need to provide some theoretical backgrounds.
- Overall scope is limited to simple (local) linear assumption, which is acceptable in theory-based paper like this, but it would be helpful if it can relate to more practical, real-world models’ behaviors.

---

> ### Author Rebuttal · Authors · 2026-03-31
>
> We thank you for appreciating our framework for TTT and its originality. We address your concerns below.
>
> **Weakness 1. readability and presentation**
>
> Thank you for the comment. The appendices already contain detailed explanations of technical concepts and a short primer (App. C), but we agree that too much of this material is currently left to the appendices. We will move a concise roadmap and some of the technical introduction currently in App. C into the main text, and add intuitive explanations. We expect this to substantially improve accessibility.
>
> **Weakness 2-1. scope of the theory**
>
> Our goal is **not** to claim that practical TTT is exactly linear. Rather, we study the practically relevant **few-step, restricted-update regime**, where many TTT methods operate. The local model is intended as a controlled approximation in that regime (similar to the neural tangent kernel [1]), and App. F.1 already provides a finite-step deviation bound between the nonlinear and linearized prompt corrections for a few steps. We will make this scope clearer in the main text and state more explicitly that fully nonlinear prompt dynamics are important future directions.
>
> **Weakness 2-2. gap to practice**
>
> We agree that this bridge should be made more explicit. Our current theory is already specialized to Transformers: Sec. 3.3 derives the prompt geometry for attention value updates and Sec. 8.2 derives query-aware block/head scores. In addition, we partially bridge this gap through an additional experiment on a real-world dataset (please refer to our response to Reviewer EiDU (Question 2)).
>
> ---
>
> We appreciate this helpful feedback and would be glad to clarify any remaining points or respond to further questions during the discussion period.
>
> ### Reference
>
> [1] Jacot, A., Gabriel, F., and Hongler, C. Neural tangent kernel: Convergence and generalization in neural networks. In Advances in Neural Information Processing Systems (NeurIPS), 2018.

---

> > ### Author Rebuttal · Reviewer_v6Qe · 2026-04-05
> >
> > Authors have clearly addressed my concerns, and please make sure to add explanation / discussion how the submission mainly focused on simple, theoretical perspective would affect the real-world complicated TTT framework.

---

> > > ### Author Response · Authors · 2026-04-06
> > >
> > > We thank you for the confirmation and for this helpful suggestion. We will make the connection to more realistic TTT settings more explicit in three parts:
> > >
> > > 1. At the end of the Introduction, we will state explicitly that this paper provides a principled starting point for analyzing few-step, restricted-update TTT, rather than a complete theory of fully nonlinear TTT.
> > >
> > > 2. At the beginning of Section 3, we will strengthen the motivation for the local linearization by emphasizing its role as a controlled approximation in the few-step regime and by giving a more prominent pointer to the finite-step deviation bound (App. F.1 / Thm. F.5).
> > >
> > > 3. In the Discussion, we will clearly separate what is expected to transfer to practice from what remains out of scope. Transferable insights include prompt-dependent horizon selection and query-aligned update directions; out-of-scope aspects include large updates, changing features during adaptation, full-parameter adaptation, and self-supervised dynamics.
> > >
> > > We will also highlight the Transformer specializations in Secs. 3.3 and 8.2 as a concrete bridge between theory and practice.
> > >
> > > Thank you again for the constructive feedback.

---

### Decision · Program_Chairs · 2026-04-30

**Decision:**

Accept (regular)

**Comment:**

This paper provides a rigorous Bayesian framework for test-time training (TTT), offering a unified theoretical perspective that explains empirical phenomena such as overfitting and yields concrete design principles for hyperparameter selection. Reviewers found the theoretical contribution solid and well-motivated. The primary weakness is empirical validation: all experiments are conducted on a synthetic digit-shift task with distilgpt2, and a preliminary SST-2 experiment does not sufficiently close this gap. Contemporary TTT research validates on complex reasoning benchmarks such as ARC and BIG-Bench Hard, and the current evidence does not convincingly demonstrate that the proposed framework transfers to realistic settings. Reviewers also noted that the novelty boundary with prior work could be made clearer, and the gap between the local linear-quadratic surrogate and practical token-level objectives warrants more explicit discussion. The paper is recommended for acceptance contingent on the author's promise to place significant focus on the readability of the final version of the paper and the inclusion of the experimental results provided during the rebuttal.